# Interactive Coupling Relaxation of Dipoles and Wagner Charges in the Amorphous State of Polymers Induced by Thermal and Electrical Stimulations: A Dual-Phase Open Dissipative System Perspective

**DOI:** 10.3390/polym17020239

**Published:** 2025-01-19

**Authors:** Jean Pierre Ibar

**Affiliations:** Rheology Department, Polymat Institute, University of the Basque Country, 20018 Donostia-San Sebastian, Euskadi, Spain; jpibar@alum.mit.edu

**Keywords:** interactive coupling, dual-phase model, grain-field statistics, amorphous state, thermal-electrical stimulation, thermal depolarization kinetics, wagner charges, TSD, TWD, compensations, super-compensations

## Abstract

This paper addresses the author’s current understanding of the physics of interactions in polymers under a voltage field excitation. The effect of a voltage field coupled with temperature to induce space charges and dipolar activity in dielectric materials can be measured by very sensitive electrometers. The resulting characterization methods, thermally stimulated depolarization (TSD) and thermal-windowing deconvolution (TWD), provide a powerful way to study local and cooperative relaxations in the amorphous state of matter that are, arguably, essential to understanding the glass transition, molecular motions in the rubbery and molten states and even the processes leading to crystallization. Specifically, this paper describes and tries to explain ‘interactive coupling’ between molecular motions in polymers by their dielectric relaxation characteristics when polymeric samples have been submitted to thermally induced polarization by a voltage field followed by depolarization at a constant heating rate. Interactive coupling results from the modulation of the local interactions by the collective aspect of those interactions, a recursive process pursuant to the dynamics of the interplay between the free volume and the conformation of dual-conformers, two fundamental basic units of the macromolecules introduced by this author in the “dual-phase” model of interactions. This model reconsiders the fundamentals of the TSD and TWD results in a different way: the origin of the dipoles formation, induced or permanent dipoles; the origin of the Wagner space charges and the T_g,ρ_ transition; the origin of the T_LL_ manifestation; the origin of the Debye elementary relaxations’ compensation or parallelism in a relaxation map; and finally, the dual-phase origin of their super-compensations. In other words, this paper is an attempt to link the fundamentals of TSD and TWD activation and deactivation of dipoles that produce a current signal with the statistical parameters of the “dual-phase” model of interactions underlying the Grain-Field Statistics.


**The Objectives of This Review: Table of Content**


This review is written to present our views on the fundamental aspects of the interactions involved in amorphous polymer matter. It is not an easy thing to do for two reasons: First, we are using a more or less unfamiliar thermal analysis technique to characterize the amorphous matter, namely the thermally stimulated depolarization procedures called TSD and TWD that many polymer scientists have never heard of; and second, we discuss the TSD/TWD results in terms of a new approach to the physics of polymer interactions, which is in its first steps of broad dissemination and thus is even more unfamiliar to readers than TSD/TWD. This new theoretical approach, the dual-phase and cross-dual phase models of the interactions in polymers, uses a new language and defines new terms to describe the interactive coupling between the macromolecules and new statistics to assess the local and the collective behavior of a set of macromolecules occupying the interactive space. The new statistics emerges from classical considerations yet adds a new dissipative term to the system energy that appears to make all the difference to explain inexplicable experimental features that are observed by TSD/TWD. To be fair, the TSD methodology is covered and discussed in over a thousand publications but is often not admitted among the other classical methods of thermal analysis characterization such as DSC (differential scanning calorimetry), DMA (dynamic mechanical analysis), etc. The reasons for not being admitted to the highest podium step include the complexity of the origin of the discharge current signals (dipoles interactions, ionic and Wagner charges occupying the free volume spaces) and the lack of a clear consensus in the understanding of the peaks observed, namely T_g_, T_g,ρ_ or T_LL_.

The Table of Content for this review is as follows:
1. Introduction  1.1. The TSD/TWD experimental thermo-kinetic features attributed to the “interactive coupling” characteristics;  1.2. The main assumptions and main dynamics features of the dual-phase model;   1.3. Examples of simulation of the Dual-Split equations.2. Development  2.1. The electrical or internal motion nature of the TSD peaks;  2.2. The controversial issues raised in trying to understand “classically” the presence of the TSD peaks of T_g_, T_g,ρ_ and of the T_LL_ manifestations;  2.3. The controversial issues related to the positive and negative compensations in relaxation maps and the origin of the super-compensations observed in the amorphous state of specially cooled polystyrene samples.3. Discussion  3.1. Challenging results for the conventional models of polymer physics;  3.2. Dual-phase understanding of T_g_, T_g,ρ_ and the T_LL_ manifestations;  3.3. Dual-phase understanding of compensations, multi-compensations, and super-compensations;  3.4. Effect of the voltage field;    3.4.1. Effect of the voltage field on the dielectric medium;    3.4.2. Effect of the voltage field on the dual-split statistics of interactive coupling;    3.4.3. New interpretation of thermal-windowing and compensations;    3.4.4. The nature of what TWD is actually deconvoluting;    3.4.5. On the potentiality of the TWD methodology to find the fundamental parameters of the dual-phase model.4. Conclusions

## 1. Introduction

### 1.1. The Thermally Stimulated Depolarization (TSD) and the Thermal-Windowing Deconvolution (TWD) Characterization Techniques: A Brief Introduction

Originally, thermally stimulated current depolarization techniques were used to measure charge detrapping in low-molecular-weight organic and inorganic non-conductive compounds. Ever since 1967, they have been applied to the study of structural transitions in polymers, another class of non-conductive materials. The credit for the initial development must be given to C. Lacabanne at the University of Toulouse, France [1]; J. Vanderschueren at the University of Brussels, Belgium [2]; and J. Van Turnout of the University of Handoven in Holland [3], who applied thermally stimulated depolarization methods to the investigation of the microstructure and properties of polymers. The result of their 20 years of dedicated research has led to the publication of hundreds of articles in the leading scientific journals. Additionally, in 1974, Lacabanne pioneered the use of the polarization–depolarization procedural technique she called “thermal-windowing spectroscopy”, successively renamed over the years “relaxation map analysis (RMA)” in 1993 [4] and more recently “thermal-windowing deconvolution” (TWD) in 2022 [5], and applied it to the study of a wide variety of macromolecular materials, synthetic and organic [6]. The use of “thermal-windowing” rendered possible the deconvolution of the thermal stimulated depolarization peaks and the decoupling of the relaxation modes responsible for internal motion permitting, for the 1st time, a better understanding of their coupling characteristics which relate to the state of the material itself, in particular its thermodynamic sensitivity to “internal stress”.

The methods of thermally stimulated depolarization became very popular in the early 1990s as a result of the introduction by Solomat Instruments (Stamford, CT, USA) of the automated TSD/TWD spectrometer on the thermal analysis market [4]. At Solomat, where this author was the Director of Research, these dielectric spectrometers were called TSC/RMA, and a book was published as a guide for the use of these spectrometers and the analysis of the results to promote the sales of TSC/RMA spectrometers [4]. This new technology could “measure up” the amorphous state in such diverse applications as bonding and cohesion between matrices and fibers for composites, between metals and paints, between the crystalline phases in semi-crystalline polymers, or even between the phases in a blend or a block copolymer. In 2022, a new book on the same TSD/TWD subject was published [5] with a different objective in mind: explain the depolarization results using our new dual-phase model approach of describing the interactions in polymers by application of the Grain-Field Statistics. This new book provides a different interpretation of the results obtained by TSD and TWD, while the book published in 1993 [4] provided the classical formulation of Lacabanne [1], Vanderschueren [2], and Van Turnout [3]. In particular, in [7], in response to a question raised in [8], we develop a new understanding of the amorphous state of matter submitted to an electrical field using the variables that enter the dynamic equations of the dual-split statistics when it is brought out of equilibrium by such thermal–mechanical processing procedures as to induce dissipative structures (rheomolding [9], rheo-fluidification and sustained orientation) [10].

Our general objective is to explain all the properties of polymers using a new statistical formulation of their interactions: “the Grain-Field Statistics of dissipative systems”. The application to the rheology of polymer melts has been introduced [11,12] and is the subject of a specific publication [13]. In the case of the dielectric properties, in [5] and in this article, the subject of the polarization and depolarization of the dipoles and the creation of space charges is studied from a different angle than the traditional views [1,2,3]: Is it possible (or not) to correlate the parameters of the Grain-Field Statistics to the thermo-kinetic features observed in a relaxation map of interactive dipoles, namely compensations and super-compensation lines, and their comprehension in terms of the dual-phase model [7]?

Sophisticated thermal analytical instruments are available on the market, such as differential scanning calorimeters (DSCs), dynamic mechanical analyzers (DMAs), and dielectric spectrum analyzers (DEAs), but none can characterize the amorphous state to determine the interactive coupling molecular basis for its non-equilibrium characteristics, either in the solid or in the molten temperature regions.

The first book, published in 1993, narrated how industry engineers and scientists would welcome the arrival of an instrument to tag and measure the internal stress in injection-molded, extruded, or blow-molded parts, or capable of characterizing the segregation in a blend or a block copolymer, or determining the bonding strength of paints and glues. The book explained that Lacabanne, Vanderschuren, and Van Turnout developed the basic technology for such an instrument. Solomat Instruments LLP sought and obtained a license to develop, manufacture, and commercialize the new technology. The result was the first automated TSD/TWD spectrometer that started to sell in late 1988.

Several techniques exist to analyze the molecular response of materials to physical or chemical inputs in order to determine their specific performance. Differential scanning calorimetry (DSC) and differential thermal analysis (DTA) are among the most popular in laboratories and on production sites. Other techniques use equipment such as thermal–mechanical analyzers (TMAs), dynamic mechanical analyzers (DMAs), stress relaxation or creep analyzers, thermal expansion coefficient devices, and dielectric analyzers (DEAs). The method of thermally stimulated depolarization (TSD) consists of rapidly subjecting the specimen to a high temperature (above the transition temperature at which the relaxation phenomena is expected), orienting the dipoles at that temperature, and freezing in the orientation thus produced by quenching at low temperature (Figure 1).

The voltage field applied is then removed, and the temperature is ramped linearly back up to reveal the polarization induced at high temperatures. **TSD is therefore a thermally stimulated recovery experiment**. An electrometer is connected to the sample to record the short-circuit current while heating. A current is created when the material depolarizes. This thermally stimulated depolarization current reveals the molecular mobility of the material’s structure. The rate of depolarization is related to the relaxation times of the internal motions providing a new opportunity to study the physical and morphological structure of materials.

The depolarization current, J, flowing through the external circuit is measured by a very sensitive electrometer (capable of measuring currents 10 million times smaller than those measured by a tunneling microscope) and allows for the determination of the “dipole conductivity”.

The current peaks recorded this way (Figure 2a,b) are found to correlate well with the transition temperatures measured by mechanical relaxation (DMA), by DSC, or by conventional (a.c.) dielectric spectroscopy (DETA). A TSD output looks like a tan δ versus temperature plot, showing maxima at the transitions occurring inside the material. In fact, TSD provides very similar results to those obtained from other analytical instruments operating at the same low-frequency equivalent (10^−4^ Hz), with the addition of an accrued sensitivity, and a separating power unseen in other technologies.

The concept of “thermal-windowing” gives the TSD another dimension. It consists of polarizing only a fragment of the full spectrum of relaxation and depolarizing it partially to isolate or “window” a single relaxation process. There are two types of possible windowing techniques. The first method, which can be called “partial isothermal recovery” or “isothermal-windowing”, consists of the following: First, the sample is polarized at temperature T_p_ for a time t_p_ adjusted to allow for orientation only of a certain fragment of the dipoles. At the same temperature T_p_, the polarizing voltage is cut off and maintained at T_p_ for a time t_d_. This allows for the depolarization of a fragment of the oriented dipoles. Finally, the sample is quenched to obtain T_o_ << T_p_. The sample is reheated at a constant rate and the current of depolarization is measured. Δt = (t_p_ − t_d_) is the “time window” and can vary between 1 min and about 1 h.

The second commonly used thermal deconvolution method is the “thermal-windowing deconvolution experiment (Figure 3), which we designate TWD in this review. TWD essentially gives identical results as the first deconvolution method, yet it is faster to practice. In this option, a constant voltage is applied at T_p_ for a time t_p_, commonly of the order of 2 min. The temperature is then lowered to T_d_ at which the voltage is removed, and the specimen is allowed to recover partially for a time t_d_, usually equal to t_p_. ΔT = (T_p_ − T_d_) is the temperature window and can vary between 1° and about 10 °C. The specimen is then quenched by 50° to 100 °C to a sub-temperature (T_o_) at which the amount of polarization induced in the material is frozen. A linear heating-up is then performed, and the variation in current due to thermally induced depolarization or other current discharges is observed as a function of time (i.e., temperature). Since the current, J(t), is the derivative of polarization, the ratio P(t) divided by J(t) is a quantity with the dimension of time and represents, according to Bucci et al. [14,15], the elementary relaxation time τ_i_ typical of the relaxing system. Figure 4 shows the result of thermal-windowing on the TSD output.

When t_p_, t_d_, and (T_p_ − T_d_) are conveniently chosen, the depolarization current is supposed to represent the relaxation of a single Debye relaxation mode isolated from the spectrum of relaxation modes. By varying the value of the temperature of polarization T_p_ and repeating the above thermal-windowing process, one can isolate the elementary modes one by one (Figure 5).

The computer in the automated TSD/TWD spectrometer of Solomat integrates the current vs. temperature peak for each temperature and calculates the value of the relaxation time at each temperature. According to Bucci’s equation ([14,15]; also p. 7 and p. 34 of [5]), the analysis of each resolved Debye peak obtained at various polarization temperatures gives a temperature-dependent retardation time τ_i_(T), which often follows an Arrhenius dependence (Figure 6).

According to Lacabanne [1], the relaxation time in Figure 6 is the inverse of the frequency of jump between two activated states of the depolarization process, the intercept of the Arrhenius equation is proportional to the entropy of activation for the activated process involved, and the slope is proportional to the enthalpy of activation. If a structure is “loose”, the contrary of “ordered” or “compact”, i.e., when molecular mobility is less hindered by the interactive intra–intermolecular surrounding, the entropy of activation will be “larger”. Conversely, any parameter that acts to “organize” the structure and create a tighter environment for the bonds will cause a decrease in the entropy of activation. So, the activated entropy calculated from the intercept of Figure 6 gives an indication of “the degree of disorder” (DOD) of the structure [16].

A relaxation map (Figure 7) is obtained from a TWD experiment: this involves the collection of the relaxation lines obtained for each deconvoluted Debye peak and is analyzed according to Bucci’s equation. While the techniques based on thermally stimulated depolarization, even when named using various other designations than TSD, have been popular to characterize molecular motions in all kinds of non-conductive materials [1,2,3,4,5,6], the TWD technology leading to a relaxation map appears to be more specifically suited to determine the degree of cooperativeness between the relaxation modes responsible for internal motions at the main transitions, revealing the state of their structure and their morphology [5,16]. Relaxation maps can be looked at as “fingerprints” of the material, as they are representative of its chemical structure, morphology, and non-equilibrium structure (Figure 8).

The analysis of the relaxation map determines the elementary enthalpies of activation and the pre-exponential factors (related to the entropy of activation) for all the relaxation modes obtained by varying the temperature of polarization T_p_.

In summary, the relaxation observed during the recovery stage of TSD reveals the kinetics and the powerful method of “thermal-windowing deconvolution” (TWD) used to deconvolute the individual relaxation modes. This allows for the study of their coupling characteristics, reflecting the structure and the physical state of the material. Constitutive equations can be used thereafter to reconstruct the material dielectric behavior (Figure 9) by calculating the fundamental physical parameters from the spectrum of relaxation (dielectric permittivity, etc.).

The various Arrhenius lines obtained by thermal-windowing at different polarization temperatures (T_p_) often converge to a common point, namely the compensation point (Figure 10).

The spectral lines in Figure 10 apply to the thermo-electrical activation of the dipoles below T_g_ (T_p_ < T_g_), which results in a “positive” compensation: this means that the temperature of the compensation point is located above T_g_, and the convergence points downwardly. When T_p_ > T_g_ but below T_LL_, the spectral lines converge upwardly and backward to a compensation point located below T_g_: the compensation is designated “negative”. An example is shown in Figure 11. When T_p_ > T_LL_, the spectral lines are parallel to each other and thus no longer converge (for an amorphous state at or near equilibrium), and they may no longer display an Arrhenius behavior (their spectral line is curved).

The coordinates of the compensation points, either positive or negative, can be found by a “compensation search”, a plot of the intercept versus the slope of the Arrhenius spectral lines. The compensation is validated when the plot in the compensation search (called a compensation line) is linear. In such a case, the dipoles are not independent in their relaxation; they are “interactively coupled”, the meaning of which is defined and specified in this review. The coordinates of the compensation points are calculated from the slope and the intercept of the compensation line. When the dipoles are **not** interactively coupling, i.e., when their motions are independent, their spectral lines are parallel, and there is no compensation point in the compensation search. This lack of interactive coupling occurs above the T_LL_ transition under equilibrium conditions.

It should be noted that there are several types of representation of the state of interactive coupling of an amorphous phase, with the relaxation map of log τ_i_ vs. 1/T in Figure 11 considered one of them. For instance, when the relaxation time τ_i_ of each mode is converted to its Eyring form, τ_ι (Eyring)_, we obtain an “(Eyring relaxation map) pursuant to the following equation (Eq. 2.5 of ref. [5]):τ _(Eyring)_ = (h/kT) exp(−ΔS_p_/k) exp(ΔH_p_/kT)
where h and k are the Planck and Boltzmann constants, respectively, and ΔH_p_ and ΔS_p_ are the enthalpy and entropy of activation of the Debye elementary relaxation modes at T = T_p_. The relaxation data can also be presented in the ΔG vs. T plane, where all the Eyring relaxation times are converted into ΔG_i_ relaxation spectral lines using ΔG(T) = ΔH_p_ − T ΔS_p_ where T, the x-axis, is the temperature during the depolarization stage (see Figure 2.7 of ref. [5], p. 71). The compensation search in the ΔG plane consists of a plot of ΔS_p_ vs. ΔH_p_, as illustrated in Figure 12 for an amorphous polymer exhibiting a single T_g_ transition, “a one-phase system”.

In Figure 12, the intersection of the positive compensation line (top) and the negative compensation line (bottom) occurs at T_g_, offering a very precise and sensitive way to characterize the T_g_ of an amorphous phase in a polymer. For polymer blends of two largely immiscible components, the variation in the coordinates of the intersection of the two compensations allows us to quantify the amount of dissolution of one component in each phase.

In conclusion, the interest in thermally stimulated process outcomes revolves around this phenomenon of compensation, the determination of the coordinates of the compensation points, the interpretation of its origin, its practical use to characterize the degree of coupling in the amorphous phase of polymeric matter, and its relationship with the state of (non)-equilibrium [16]. In our opinion, a new type of thermal analysis was born with the introduction of the “thermal-windowing deconvolution” experimental procedure (TWD), which apparently allows us to isolate, i.e., filter out one by one, the single elementary Debye peaks that constitute the global depolarization peaks during the heating stage (Figure 5).

The purpose of this review is to clarify the general description of TWD stated above, i.e., discuss and challenge the consensual understanding that a spectrum of elementary relaxation modes coexist in global TSD peaks and that the technique of TWD can deconvolute single Debye peaks from them. The challenge is to understand the compensation of the elementary relaxations, either the positive or negative compensations, and determine the meaning of the “interactive coupling” between these elementary relaxation motions extracted from global peaks. In other words, to simplify the true fundamental issue behind this research, is the deconvolution of global depolarization peaks into relaxation maps of compensating single Debye relaxations a sophisticated curve-fitting procedure or is it fundamentally revealing the dual-phase and dissipative nature of the interactions in polymers? In order to better position this issue, we need to briefly review some of the assumptions of the model of dual-split kinetics (EKNETICS) that serve as the foundation of our development of the dual-phase open dissipative system perspective ([7,10,11,12,13]).

### 1.2. Introduction to the Dual-Split Kinetic Model (EKNETICS)

It is not our intention to present here the details of our model of the physics of interactions in polymers. The interested reader can refer to an introduction of the dual-phase and cross-dual-phase models of polymer physics interactions in references [10,12,13,17,18]. We will limit ourselves in this text to presenting the general principles and fundamentals of the theory in order to define the statistical parameters dealt with in the EKNETICS set of equations. In our view, “dual-conformers”, the constituents of macromolecules, gather into statistical systems that go beyond belonging to individual macromolecules. A “conformer” is shown in Figure 13, duplicated from reference [19]. The macromolecules themselves represent a chain of “covalent conformers” put together as an entity. The problem is to determine whether the chain properties, derived from its statistics, entirely control the dynamics of the collection of chains making up a polymer. This is what has been assumed by all the other theories, and this is what the dual-split kinetics and the Grain-Field Statistics challenge.

In our view, the free energy of the collection of chains assembled as a polymer is not equal to the scaled-up free energy of a macromolecule embedded in a mean field created by the influence of the other macromolecules, at least not below the temperature T_LL_ (see below), which itself is a function of the dynamics of the experiment and the chain characteristics. In fact, for many experimental conditions, depending on temperature and other factors, our model of polymer interactions does not require, in its hypotheses and derivations, a description of the changes that occur to the individual macromolecules. The dynamic statistical systems dealt with to determine the free energy and its structure (enthalpy and entropy) are not macromolecules in our approach. However, the fact that macromolecules compose the basic structure is essential, for instance, to understand the basis of our new dual-phase statistics and to explain “entanglements”, for which our model provides a completely different interpretation than the ones offered by the conventional spaghetti bowl or tube models [10]. A “covalent conformer” is not the same as a “free conformer” (Figure 13), such as a small molecule used as the monomer in the polymerization process. Its interaction with other conformers by covalent bonding modifies the conformational potential energy of a free conformer, and this governs the statistical properties of a “free chain”. Here, we are still “classical”, yet when dealing with a collection of chains put together, our approach differs from the classical one. Conformers belong to two types of sets: they belong to macromolecules, which link them via covalent forces, as we just said, or they belong to the grand ensemble of conformers, which are linked by inter–intra molecular forces; van der Waals forces; dipole–dipole forces; and electrostatic interactions, which affect and define the viscous medium. That duality is intrinsic to conformers, which we call “dual-conformers” to mark this specificity. The potential energy of a dual conformer is different from the potential energy of a conformer part of a free chain. To simplify, one could view the difference between our statistical model and the classical model to describe the properties of polymers as follows: according to classical views, statistical systems involve macromolecules, i.e., a network of chains; the properties of the chains are disturbed by the presence of other chains and by external conditions (temperature, stress tensor, electric field, etc.). The classical definition of a statistical system contrasts with our approach in which statistical systems involve “dual-conformers”, not macromolecules. The interactive coupling between dual-conformers is defined by a new field of statistics, the Grain-Field Statistics, which explores the correlation between the local conformational property of dual-conformers and their collective behavior as a dissipative network. In what follows, for simplicity, we will just call the dual-conformers “conformers”. Again to simplify, the statistics systems that are used by classical models and our model indicate that the rotational isomeric state (RIS) of the conformers are fundamentally different: The classical molecular dynamic statistics is the Boltzmann statistics, famous for its kinetic formulation of the properties of gases. The dual-split or dual-phase statistics, leading to the Grain-Field Statistics, is inspired by the classical Boltzmann concept but departs from it by defining a dissipative term in the equations and assuming that the free energy remains always equal to its minimum value, that of the equilibrium state, even for transient states. The kinetics created by such changes in the fundamental equations result in the formation of free energy structures, which we once called “the energetic kinetic dissipative network of conformers (EKNET)” ([20,21,22,23,24,25]), and more recently, while dealing with rheology, we changed it to “the elastic dissipative network” ([10,17]). In our analytical formulation of the dynamics of these “open dissipative systems of interactions” generated by our two modifications of the classical formula, we realized that, essentially, two mechanisms of structuration of the free energy prevail and compete, namely a “vertical structuring” and a “horizontal structuring”, each specifically applying its own version of the basic equations. This distinction increases the complexity of the analytical solution but is, in our opinion, a fundamental aspect of the way interactions work. Vertical structuring refers to a split of the units (collectively interacting in the system) into two compensating sub-systems having each a different statistical partition. The horizontal structuring offers a different split of the collective set via the generation of N_s_ identical sub-systems, each with the same statistical partition. Each split mechanism generates a dissipative function. The total dissipative function ought to be minimized (it is 0 at stable equilibrium), a condition that creates their compensation, i.e., whether they work independently, in sequence, or together. The set of equations used to simulate the dynamics of a given dissipative system belongs to the general solution designated “the Grain-Field theory”. By applying these general principles to polymers, specifically, we can now summarize our model in one sentence: We define a polymer as a set of “dual-conformers”, i.e., a set of three-bond elements belonging or not to the same macromolecule, submitted to intra–intermolecular forces described by an energy potential to which the equations of the Grain-Field Statistics theory apply. In this review, we only focus on the simplest simulation model, the vertical dual-split kinetics. References [18,25] provide examples of more complex dynamic simulations that we suggest can be applied to simulate many specific problems of polymer physics, including the action of a mechanical stress field on the interactions between the dual-conformer networks [13]. In this paper, we present the dual-split kinetic model in a way that makes its application to TSD and TWD results more transparent. Working by analogy, one will recognize some identical patterns of behavior between computer simulations and the real response of polymers using TSD and TWD stimulation techniques. The simulation is presented in a shortened and simplified way to concentrate on the meaning of the concepts that could be useful to the study of interactive coupling in the amorphous state, the subject of this review. However, further details of dual-split simulations (the effect of cooling/heating or annealing to simulate other viscous flow or thermal analysis results) can be found elsewhere ([10,11,12,13,18]). As we will show below, the dual-split dynamics depend on the value of three fundamental parameters, namely υ_m_ and Δ_m_, which are independent of external actions such as stress or voltage, and Δ_e_, which can vary with the application of external actions. Within the scope of this review, we limit ourselves to dynamic situations that simulate simple experiments after the system’s initial conditions have been imposed; these simulations imply that the value of Δ_e_ is predetermined by the effect of an electrical or mechanical field applied prior to the dynamic thermal stimulation. In the Section 2, we discuss how a thermal–electrical history procedure, such as the TWD characterization studied in this article, can be simulated by modifying Δ_e_ with the temperature of polarization, T_p_, and the intensity of a voltage field. Once the value of Δ_e_ is set by these preconditions, the depolarization step of TWD can be simulated by returning to the equilibrium induced by heating at a constant rate in the equations of the dual-split kinetics.

However, first, let us describe the assumptions that led to the dual-split kinetic model, also called the EKTOR model (from the fusion of energetic and kinetic considerations) or the EKNETICS equations since the result is the formation of a self-induced network, the dissipative network.

#### 1.2.1. Conventional Kinetics

The study of kinetics is a discipline that describes the evolution of the units of a population of, say, chemical molecules, which participate in chemical reactions. Another example would be to describe the evolution of units of a population that could occupy different “states”. Many other terms have been used to describe the same objective, for instance, “statistics”, or “dynamics”, as shown in the following definitions: the population partition that evolves with time can be studied with the tools of “statistics”, a transient statistics in fact, a field also regarded as “dynamics”. All these definitions are used in our presentation. The important thing here is to define the terms quantitatively.

Consider a simple dynamic process such as a first-order reversible kinetic equation between two states, t ↔ cg, controlled by an activated process with direct and reverse kinetic constants k_1_ and k_2_ and activation energy Δ_1_ and Δ_2_, respectively. One can write the following elementary set of kinetic and thermodynamic equations to describe the evolution of the system:B_o_= (n_t_ + n_cg_)(1)

Here, B_o_ is the total number of units in state energy levels t, cg.ΔG = RT Ln(n_t_/n_cg_) is the Free energy of the system withΔG_e_ = RT Ln(n_te_/n_cge_) = RT Ln (k_2_/k_1_) the equilibrium value(2)
(the sub-index “e” corresponds to the equilibrium value). R is the gas constant.dn_t_/dt = −k_1_ n_t_ + k_2_ (B_o_ − n_t_)(3)dn_cg_/dt = −k_2_ n_cg_ + k_1_ (B_o_ − n_cg_)
withk_1_ = υ_m_ exp(−Δ_1_/RT) and k_2_ = υ_m_ exp(−Δ_2_/RT)
where υ_m_ is the frequency front factor.

In the following, we callΔ_e_ = (Δ_1_ − Δ_2_)/2 and Δ_m_ = (Δ_1_ + Δ_2_)/2(4)

When the total number of units in the two levels, B_o_, is constant, the statistics apply to a **closed** thermodynamic system. If the system is cooled at a constant cooling rate q = dT/dt, the now non-linear system of differential equations above can easily be solved (for instance, using a Runje–Kutta fifth-order algorithm) to produce a set of n_t_ and n_cg_ values at each temperature, which can be compared to those of equilibrium at the same temperature. This will determine the effective departure from equilibrium due to non-isothermal cooling. Notice in the equation that gives the free energy that the term Ln(n_t_/n_cg_) reflects the departure of the free energy from its equilibrium value, and n_t_, n_cg_ values are determined by solving the kinetic set. Therefore, under non-isothermal conditions, the free energy plays a very subordinate role, and its magnitude is driven by the kinetic aspect.

For a classical set of kinetic equations, it is clear that the transient free energy is not at its minimum value, since the value of the minimum free energy at any given temperature is known to be the equilibrium value at that temperature: ΔG_e_ = RT Ln(k_2_/k_1_).

When the system is brought out of equilibrium and then allowed to relax, the kinetic equations drive the system back to the equilibrium state, which implies that the value of the equilibrium free energy is implicit in the formulation of the kinetic constants. In fact, the ratio of the two kinetic constants is equal to the thermodynamic constant, a quantity that gives the partition function for the two energy levels at equilibrium: ΔG_e_ = RT Ln(n_te_/n_cge_).

In conclusion, for classical kinetic equations under non-isothermal conditions for which the solution is driven by kinetic considerations only, the free energy of the system is not equal to its minimum equilibrium value.

#### 1.2.2. Dual-Split Kinetics (EKNETICS)

Can we modify the set of equations driving the kinetics so the system’s free energy stays at its minimum value at all times? We call the solutions to this challenge “the EKNETICS equations”. The dual-split kinetic model, also called the EKTOR model, is the simplest EKNETICS set of rate equations fulfilling these conditions.

It is assumed that we can split each state cg and t into an f and b category, giving four states cgf, cgb, tf, and tb; the result is a **dual partition**, between cg and t units on the one hand and between the f and b units on the other hand. The two partition functions are coupled.

In what follows, we present the assumptions driving the new EKNETICS and study the difference between its results and the results obtained classically. The new equations converge to traditional kinetic equations at “long times”, which we will learn corresponds to either T > T_LL_ or under “true” equilibrium conditions. Under non-isothermal conditions, the system becomes **self-dissipative**, i.e., for closed systems, its free energy remains equal to its equilibrium value but its Eknetics cross-dual partition evolves in time.

##### Structuring between the b and F Dual-Conformers

Suppose the total B_o_ population is split into two sets of units, N_b_ and N_f_, with N_b_(t) and N_f_(t) transient or steady state. The total population (N_b_ + N_f_) = B_o_ remains constant, and for this reason, the rate of change in N_b_ is equal to minus the rate of change in N_f_. The kinetics of n_tb_ and n_cgb_ in the N_b_ sub-set are different from the kinetics of n_tf_ and n_cg_ in the N_f_ sub-set. We can define a marker of the split between these two sub-sets, N_b_(t)/N_f_(t), to follow the split situation:B_o_ = N_b_ + N_f_(5)N_b_ = (n_tb_ + n_cgb_) N_f_ = (n_tf_ + n_cgf_)

It is clear that, under equilibrium conditions, N_be_ = N_fe_ = B_o_/2, and that the population of each level is n_tbe_, n_cgbe_, n_tfe_, and n_cgfe_, and of coursen_tbe_ = n_tfe_n_cgbe_ = n_cgfe_Ln(n_tbe_/n_cgbe_) = Ln(n_tfe_/n_cgfe_) = Ln(k_2_/k_1_)

The free energy is equal to:0.5 RT Ln(n_tbe_/n_cgbe_) + 0.5 RT Ln(n_tfe_/n_cgfe_) = RT Ln(k_2_/k_1_) = 2 Δ_e_
where R is the gas constant. In the equilibrium state, the two types of units become indistinguishable; only non-isothermal kinetics will populate them distinctively.

##### Dual-Split Kinetics (EKTOR EKNETICS)

Let us now consider the following modified system of equations, the “vertical dual-split statistics” system of equations, and comment on the assumptions made:N_b_ = n_tb_ + n_cgb_(6)N_f_ = n_tf_ + n_cgf_B_o_ = (N_b_ + N_f_)dn_tb_/dt = (dN_b_/dt)/2 − k_1_ n_tb_ + k_2_ (N_b_ − n_tb_)(7)dn_tf_/dt = (dN_f_/dt)/2 − k_1_ n_tf_ + k_2_ (N_f_ − n_tf_)k_1_ = υ_m_ exp(−Δ_1_/T)k_2_ = υ_m_ exp(−Δ_2_/T)
(the gas constant R is equated to 1 here and from now on.)Ln (n_tb_/n_cgb_) + Ln (n_tf_/n_cgf_) + Ln (N_b_/N_f_) = 2 Ln (k_2_/k_1_) = 4Δ_e_/T(8)q = dT/dt (under isothermal relaxation q = 0)

Note the presence of an additional term, Ln (N_b_/N_f_), in the expression of the free energy. This function is what we designate **the “dissipative term”.** Its introduction is fundamental in our work on interactions; it is the source of the originality of the new statistics and results in the study of a new generation of dynamics open self-dissipative systems. At equilibrium, N_be_ = N_fe_ = (B_o_/2), and therefore Ln(N_b_/N_f_) is equal to zero.

Additionally, in Equation (7), note another important modification of the original kinetic equations, which now includes an extra term: (dN_b_/dt)/2 for b-conformers and (dN_f_/dt)/2 for F-conformers. The introduction of this term is to make the different rates consistent with the belonging of the units into one single closed system. Hence, the presence of this term is not an assumption: it is a requirement.

When B_o_ is constant in Equation (7), for closed systems, (dN_f_/dt)/2 = −(dN_b_/dt)/2.

There is no assumption in Equations (6)–(8) regarding the variation in dN_b_/dt and its temperature dependence: the solutions come directly from Equations (6)–(8), as illustrated in Figure 15, which is discussed in the next section. The coupling between energetic and kinetic constraints driving the new kinetic model is now apparent: the solutions for n_tb_, n_cgb_, n_tf,_ n_cgf_, and N_b_ are not obtained from a simple kinetic assumption (the expression of the kinetic constants and the proportionality between rate and population concentration). There is a real coupling between the free energy constraint and the kinetic constraints. The condition regarding energy is nothing other than a minimal principle since it is assumed that the free energy remains that of the equilibrium state at the corresponding temperature. In conventional kinetics, a system always evolves toward its minimum free energy, and therefore the minimum value is given by the equilibrium value. In dual-split kinetics, the free energy remains equal to its equilibrium value when it evolves, which is a fundamental new assumption.

### 1.3. Examples of Simulation of the Dual-Split Kinetic Equations (6)–(8)

The solution of Equations (6)–(8) is obtained by solving the system of equations numerically, using a fifth-order Runje–Kutta algorithm.

#### 1.3.1. Cooling Simulation

Let us consider a system of B_o_ = 1000 units, with statistics corresponding to the following parameters: Δ_e_ = 250, Δ_m_ = 9250, and υ_m_ = 10^11^. Recall that Δ_m_ is equal to (Δ_1_ + Δ_2_)/2, and Δ_e_ is equal to (Δ_1_ − Δ_2_)/2. The system cools at a rate of q = −1 degree Kelvin per second from an initial temperature of T_o_= 400. The system is assumed to be initially at equilibrium at that temperature, so N_b_ = N_f_ = B_o_/2.

In Figure 14, the rate of change in N_b_ and dN_b_/dt is plotted against the cooling temperature. The rate does not remain equal to zero: the rate increases rapidly at first, goes through a maximum, and decreases back to zero at lower temperatures. Figure 15 shows the variation in the dissipative term, Ln (N_b_/N_f_), which rises from its equilibrium value (0) to a plateau value at a lower temperature. This figure clearly demonstrates that the minimal principle of the free energy implies a structuring of the b/F population as the non-equilibrium triggered by cooling proceeds: cooling favors the production of b-conformers. Conventional kinetics (Equations (1)–(3)) also creates non-equilibrium values for the cg and t populations, but the interactive coupling between the b/F kinetics and the (c,g,t) kinetics in Equations (6)–(8) generates significant large differences. The differences between the conventional kinetics (“classical”) and the dual-split kinetics (EKNETICS) are illustrated in Figure 16, Figure 17 and Figure 18.

Figure 16 is a plot of n_t_ and n_tb_ vs. Temperature, respectively, for a system of B_o_ = 1000 conformers either submitted to the conditions of Equations (1)–(3), for classical kinetics, or to Equations (6)–(8) for dual-split-kinetics (“DSK”=EKNETICS in Figure 16). The values of Δ_m_, υ_m_, and Δ_e_ are the same for both simulations, given in the captions of Figure 14 and Figure 15. The cooling rate is also the same; q = −1 K/s. n_t_ reaches a non-equilibrium plateau at the low temperature of 458.3, which is 22% lower than the plateau value of n_tb_ (Figure 16), or 45% lower than the total plateau number of trans-conformers (n_tb_ + n_tf_) shown in Figure 17. The modulated effect of dN_b_/dt on the dynamics of n_tb_, n_tf_, n_cgb_, and n_cgf_ is quite significant.

In Figure 17, both n_tb_ and n_tf_ are plotted vs. T (n_tb_ is the top curve). One sees that the increase in the trans-conformers is only favored in the b state but actually decreases in the free volume interfacial tissue, i.e., in the F state. The variation in n_tF_ (the lower curve of Figure 17) suggests that the flexed conformations, the cis-gauche, are favored by the cooling process for the free volume conformers. As a result, the total *trans*-population, (n_tb_ + n_tF_), although we said is much larger than what classical kinetics could produce, plateaus at a low temperature to a value that is only greater by 12% over the equilibrium value at T_o_ = 400.

The difference between the transient behavior for the n_tb_ and the n_cgb_ conformers during cooling is shown in Figure 18. From T = 400 down to 320, n_cgb_ decreases, while n_tb_ rises toward a plateau; then, as T continues to decrease, n_cgb_ rises slightly toward its own frozen plateau value. The sum (n_tb_ + n_cgb_), which is equal to the total of b-conformers, increases and plateaus off to 685, which is 49.6% greater than the corresponding value obtained assuming classical kinetics (n_t_ + n_cg_). Figure 17 and Figure 18 provide and compare the populations in various “levels’ and demonstrate that the kinetics of (cg ↔ t) is coupled and modulated by the kinetics of (b ↔ F) and vice versa.

In summary, the coupling of pure kinetic considerations with a minimal principle, resulting in the splitting of B_o_ into the b and F sub-sets, has a considerable effect on the kinetics of the transient dynamics and the non-equilibrium values obtained at low temperatures (by analogy with a glass, corresponding to the glassy state values). The b and F sub-systems are **conjugated open dissipative systems** because their individual total number of units is not a close set; it varies with time. Only the number of units of the overall system, B_o_, is closed. This dual-split kinetic mode, the vertical structuring mode, gives rise to self-dissipative effects, quantified by the dissipative parameter, Ln(N_b_/N_f_), which transcribes the departure from a pure equilibrium state. The minimum principle of free energy establishes the value of free energy at all times as that of equilibrium, yet pure equilibrium is obtained when the statistics of the populations are also at equilibrium.

Grain-Field Statistics is the study of all the possible ways that interactions could be organized to minimize the value of the dissipative factor. This gives rise to the concept of a dissipative wave and the finding of other compensating modes of structuration such as the horizontal structuring mode.

The energy equation, Equation (8), breaks down the total value at equilibrium, (4Δ_e_/T), into three components, namely I_b_, I_f_, and I_ds_, which are defined in Equation (9), and all are found to vary with time, as shown in Figure 19. I_b_ = Ln (n_tb_/n_cgb_)(9)I_f_ = Ln (n_tf_/n_cgf_)I_ds_ = Ln (N_b_/N_f_)

Figure 19, illustrating a plot of these functions vs. 1000/T, shows that, during cooling, I_f_ and I_b_ depart quickly from each other and the equilibrium condition value (at the left of the plot) when the dissipative term, I_ds_, sharply takes off after remaining zero until a certain temperature is reached (thus defining the equivalent of the T_LL_ temperature for the simulation—see the Section 3).

One also sees that, when plotted against 1/T, both I_b_ and I_ds_ seem to rapidly level off and stay constant at lower temperatures, whereas I_f_ takes most of the value of the total energy and varies almost linearly with 1/T, continuing the high-temperature equilibrium trend.

#### 1.3.2. Effect of the Cooling Rate on the Dynamics

The effect of the cooling rate on the kinetic rate of dN_b_/dt and the “glassy” value of N_b_(t) found at low T are shown for three cooling rates (q = −1, −10, −100 K/s) in Figure 20 and Figure 21.

In summary, the cooling rate, which is responsible for bringing the system out of equilibrium in the first place, modifies the kinetic and energy structuring, allowing dissipative effects to take place at higher temperatures. By analogy, the same situation is observed for the glassification process from the liquid state across the glass transition temperature of materials.

**Figure 20 polymers-17-00239-f020:**
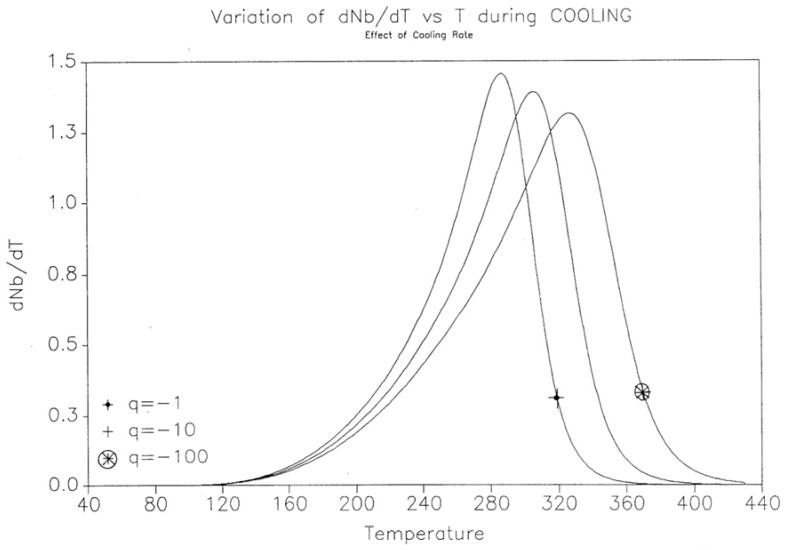
Variation in dN_b_/dT vs. T(K) during cooling. Effect of the cooling rate. Reproduced with permission from [4], SLP Press, 1993.

**Figure 21 polymers-17-00239-f021:**
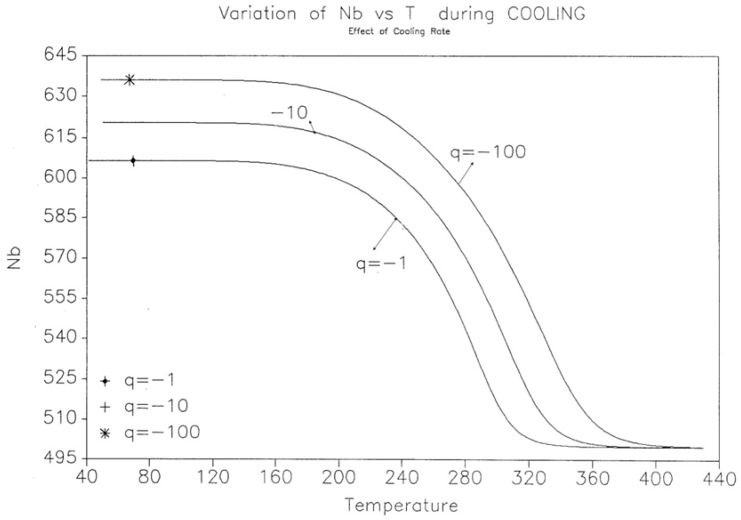
Variation in N_b_ vs. T(K) during cooling. Effect of the cooling rate. Reproduced with permission from [4], SLP Press, 1993.

#### 1.3.3. Heating at Constant Rate

Figure 22, Figure 23 and Figure 24 provide plots of non-isothermal returns to equilibrium and a comparison of the kinetic paths obtained during cooling and heating. In Figure 22, the system is first cooled at a constant rate (−1 K/s) until a temperature of 100 is reached. Taking the system variables at that temperature as initial conditions, the rate is reversed and one records the variation in N_b_ with temperature (Figure 22), n_tb_ (Figure 23), and their derivative (Figure 24). It is clear that N_b_ behaves like n_tb_, but is modulated, and one observes the classical undershoot on heating, which is common to dynamic kinetic systems studied by thermal analysis (DSC traces for instance [26]).

It is difficult to decide whether the kinetics of N_b_ “drags” the kinetics of n_tb_ or vice versa. It is probably more appropriate to say that the two are interactively coupled in some recursive fashion and that this coupling is a feature that describes the properties of a self-dissipative system (see the next section on annealing).

Figure 24 is a “phase plot” of dN_b_/dt vs. dn_tb_/dt. In this plot, we compare the cooling and heating cycles. Cooling corresponds to the loop on the right end side of the plot (designated by “1”). The small upward arrow follows the cooling pattern. In the first section of this loop, the plot is non-linear, due to the strong influence of the kinetic equation, namely Equation (7), since the kinetic constants are not frozen in this temperature range. Additionally, notice the value of dn_tb_/dt at time t = 0: it does not seem to be zero. The initial system is, however, at equilibrium (dn_tb_/dt = 0), but it instantaneously jumps to the value 0.25, and as a result, the other values seem to emerge from this state. Once the top of the loop is passed, the two derivatives are proportional to one another: the energetic constraint, Equation (8) now dominates the overall kinetics. Both rates converge towards zero as temperature decreases.

Let us now turn to the heating loop on the left of the plot (designated as “2”). The small downward arrow follows the heating path this time. The system starts to relax right away, with dN_b_/dt considered proportional to dn_tb_/dt. The system starts to imitate the process previously described for the cooling path but in reverse. Actually, there is a large difference with the cooling path: the loop is much larger, with almost double the magnitude for the maximum rates, as if the process was self-accelerating. The system finally undershoots, probably due to the excess of “speed” on its return to equilibrium, and closes in on a positive loop, which also exhibits a characteristic asymmetry with the cooling loop. We stipulate that this behavior is typical of a self-dissipative system.

#### 1.3.4. Isothermal Return to Equilibrium (Annealing)

This simulation consists of letting the system relax at a given temperature and studying the kinetics of return to equilibrium. The initial state is taken from the cooling path (cooling rate −1 K/s for all curves), and the system is allowed to relax according to the same set of Equations (5)–(7), except that q = dT/dt is now zero. Figure 25 shows the variation in N_b_(t) with time for two temperatures of isothermal relaxation, namely T_1_ = 313.90 K and T_2_ = 305.51 K. The figure demonstrates that N_b_ decreases back to the equilibrium value of B_o_/2 in a “kinetically controlled” manner: the rate of change in N_b_ is dependent on the temperature of isothermal relaxation. Since there is no specific kinetic assumption regarding the variation in N_b_, its temperature dependence should reveal the nature of the intercoupling between the kinetic and energetic constraints. Figure 26 and Figure 27 cast some light on this issue.

The decay of N_b_(t) in Figure 25 looks exponential, and therefore, the first equation we try to fit the data with is a first-order kinetic expression of the form, which is expressed as follows:dN_b_/dt = k_x_^(1)^ N_b_ – k_x_^(2)^ N_f_(10)
where k_x_^(1)^ and k_x_^(2)^ are kinetic constants. Equation (9) can be rewritten to graphically test its validity, so Figure 26 plots [(dN_b_)/dt]/N_b_ vs. (B_o_ – N_b_)/N_b_, since B_o_ = (N_b_ + N_f_); one sees that, for these two temperatures, Equation (10) is only valid as we approach equilibrium, i.e., as dN_b_/dt tends toward zero. In the near equilibrium zone (98% to 100%), we find that the first-order fit is excellent with k_x_^(1)^ = k_x_^(2)^ = 0.009737 for T = 305.90 and 0.0216275 for T = 313.90. The fit gets non-linear below 96%, as clearly observed in the figure and explained elsewhere [27].

**Figure 26 polymers-17-00239-f026:**
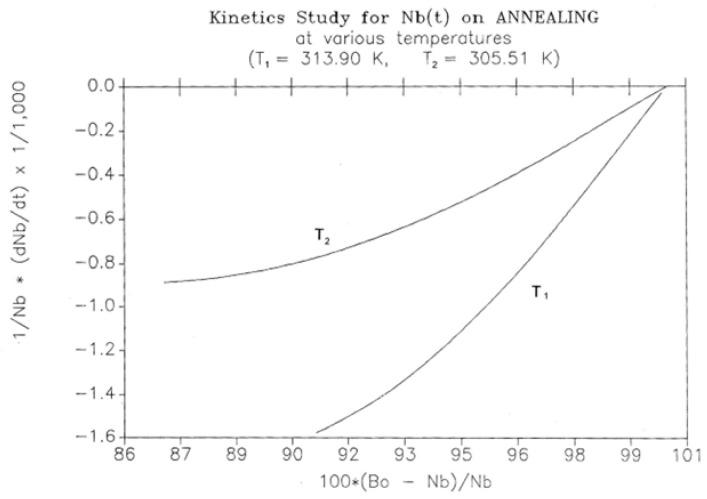
Kinetics study of N_b_(t) on annealing at two temperatures (T_1_ = 313.90 K, T_2_ = 305.51 K). Reproduced with permission from [4], SLP Press, 1993.

Figure 27 illustrates the return to equilibrium for n_tb_ at the temperature T_1_= 313.90. The graph is also not linear, showing non-linear kinetics. As we lower the temperature of relaxation, the non-linearity becomes predominant. However, let us turn our attention to the region where a first-order kinetic applies well to the description of the variation in N_b_(t). The situation is favorable as the temperature of relaxation is raised, or as we study systems with smaller Δ_e_ values.

**Figure 27 polymers-17-00239-f027:**
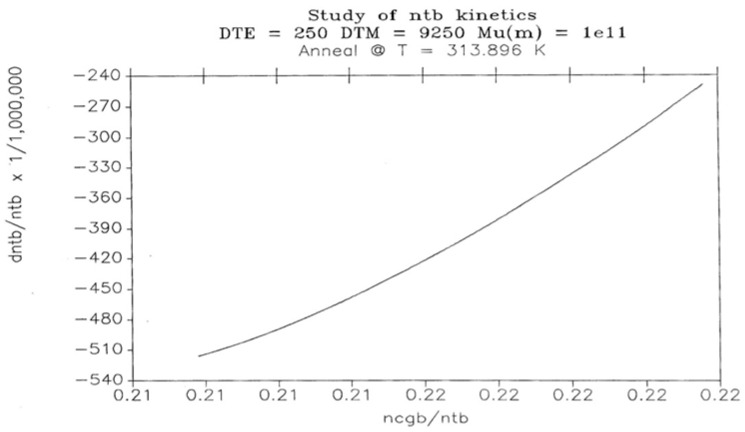
Study of n_tb_ kinetics Δ_e_ = 250, Δ_m_ = 9250, *υ*_m_ = 10^11^. Annealing at T = 313.896 K. Reproduced with permission from [4], SLP Press, 1993.

Let us call k_x_ the value of the kinetic constant found from a plot of N_b_ versus dN_b_/dt (Figure 28) when (dN_b_/dt) → 0. k_x_ is the dual-split kinetics constant, equal to k_x_^(1)^ and k_x_^(2)^ of Equation (10):(N_b_ – B_o_/2) = −[1/(2k_x_)] dN_b_/dt(11)

**Figure 28 polymers-17-00239-f028:**
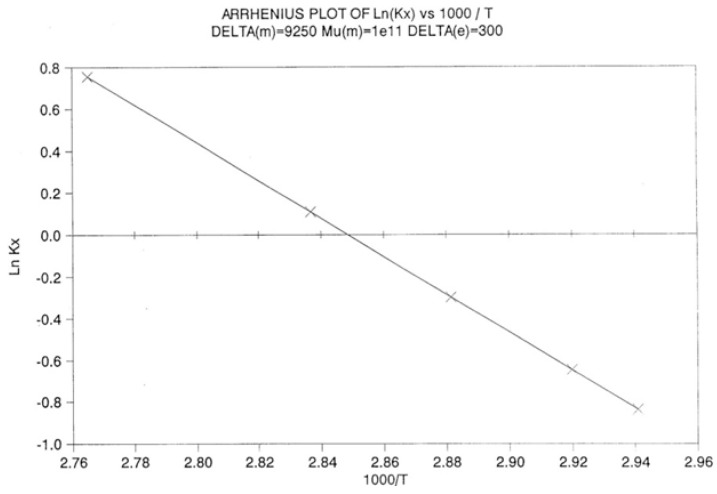
Arrhenius plot of Ln(k_x_) vs. 1000/T. Δ_m_ = 9250, *υ*_m_ = 10^11^, Δ_e_ = 300. Reproduced with permission from [4], SLP Press, 1993.

By varying the temperature of relaxation and curve-fitting the N_b_ vs. dN_b_/dt plots, as dN_b_/dt → 0, we calculate k_x_ at several temperatures. Figure 28 is a plot of Ln(k_x_) versus 1000/T to attempt to fit the temperature dependence of k_x_ by an Arrhenius equation. **The plot is linear** with an activation energy ∆_x_ and a frequency front factor υ_x_ respectively equal to 9048.2, and 7.815 10^10^ (r^2^ = 0.99999).

One can write the following:dN_b_/dt = k_x_ (B_o_ − 2N_b_) = υ_x_ exp (−Δ_x_/T) (B_o_ − 2N_b_)(12)

The (vertical) dual-split kinetic model produces the structuring of the total population of conformers which, as a result of non-equilibrium cooling, kinetically behaves like a rate process with activation energy and the frequency front factor easily derivable but different from the kinetic parameters of the main statistics. Note that υ_x_ and Δ_x_ are not equal to the parameters of the kinetic constants, either υ_m_ or Δ_m_. It is interesting that the frequency front factor is found different from the main kinetics’ υ_m_. The coupling between kinetic and energetic constraints literally generates a cross-dynamic partition: the “dissipative partition” between F and b varies in a kinetically controlled manner, just as the kinetics between cg and t, albeit with a different set of kinetic constants. The auto-catalytic effect observed for the kinetics is due to the self-generated split of the conformers into the b and F types. A thermally activated transition between a “t” and “cg” conformation, described by Δ_m_, υ_m_, and Δ_e_, cannot occur without a change in the F and b populations, and the statistics b/F is itself thermally activated with an activation energy (Δ_x_) and a frequency front factor (υ_x_) generated by and coupled to the activation energy (Δ_m_) and the frequency front factor (υ_m_) that characterize the t ↔ cg conformational transition. **This self-induced dissipative duality defines and describes the duality characteristics of dual-conformers submitted to coupled intra- and intermolecular interactions. This is the reason we have called our model of interaction “dual phase”: one “phase” is symbolized by Δ_m_ and υ_m_ and the other “phase” by Δ_x_ and υ_x_.**

When this (vertical) dual-split statistics is applied to real-time-dependent relaxation phenomena (say the relaxation of dipoles below or above the glass transition temperature of polymeric materials), the question to raise is whether the kinetic response observed corresponds to the “Δ_m_, υ_m_” kinetics, from which one can derive the dual-split kinetics parameters, or the “by-product” kinetics (Δ_x_ and υ_x_), which seems to be a self-created image of the inner mechanism, or to a combination of both. In other words, are we dealing with the F ↔ b kinetics, the t ↔ cg kinetics, or both? If what we measure is υ_x_ and Δ_x_, then we need to find a correlation between these variables to obtain the real kinetic constants υ_m_ and Δ_m_. The search for this correlation is essentially the objective of the Section 3 of this review. In the Section 3, we show that a change in the value of Δ_e_ while keeping υ_m_ and Δ_m_ constant results in the compensation between the corresponding variation in Δ_x_ and ln(υ_x_) (see Figure 63). This compensation means that the Ln(k_x_) vs. 1/T Arrhenius lines compensate when Δ_e_ varies. This situation is similar to the compensation of the −log τ_o_ vs. 1/T lines in the Arrhenius plane when T_p_ varies.

As explained in Section 3.2, this is a key result for the comprehension of interactive coupling between the conformers and the use of the compensation phenomenon to characterize it by TWD. Many classically incomprehensible features described in Section 2.2, such as the differences between the positive and negative compensations across T_g_ (Figure 11 and Figure 53) or the existence of a converging network of super-compensation lines in the relaxation maps of mechanically treated amorphous samples (PS), submitted to vibration during their cooling from the melt (Figure 45, Figure 46 and Figure 47), can be qualitatively understood from the normalized plots of (Ln(υ_x_) – Ln(υ_m_)) vs. (ΔH_x_ − ΔH_m_) obtained at various values of Δ_e_, as illustrated in Figure 66 and Figure 67, for instance. In summary, simple relationships between Lnυ_x_, Δ_x_, and Δ_e_ exist, which are revealed by varying Δ_e_ in Equations (6)–(8). The vertical splitting kinetics is, on its own, powerful enough to simulate the effect of activating the dipoles (permanent and/or induced) at the polarizing temperature T_p_ and observing its thermally activated depolarization as a Debye current.

## 2. Development

### 2.1. The Electrical or Internal Motion Nature of TSD Peaks

It should be clear from the Introduction section that the sample to be tested in a TSD instrument is a dielectric capacitor charged by the application of a voltage at a high temperature and then quenched to freeze the charges. The polarization stage consists, therefore, of producing a thermal electret. The depolarization stage, which is induced by thermal energy as temperature ramps up at a given rate, occurs by shorting the sample using an electrometer. The electrometer is nothing more than a current amplifier of exceptional characteristics capable of detecting infinitesimal current discharge. Recent advances in metal–oxide–silicon (MOS) field-effect transistors (FETs) and the development of bipolar FETs and MOS operational amplifiers have enabled the detection of ultralow current (down to 10^−17^ Amp). The electrometer in a TSD/TWD instrument is so sensitive that it measures currents one million times smaller than those measured in tunneling microscopes. Dipolar, ionic, and space charges create the thermally stimulated current output. As already described in Figure 1, the process involves two steps: the polarization stage and the depolarization stage. In the polarization stage, the power supply (battery) charges up the capacitor, creating a current of polarization I_p_. The thermally stimulated discharge of the charges results in current, I_d_, in the depolarization stage, flowing in the sensitive electrometer. The effect of the voltage field is to stretch and/or orient the dipoles (either molecular or ionic), distort the orbitals of the atoms, and create new charges either at interfaces between heterogeneous surfaces or at internal local places of micro-density differences, for instance, in the F spaces around the b-grain structures in the dual-phase model (Figure 22 of ref. [12] or Figure 1.8 in ref. [10]). The current measured in the electrometer is the total of all contributions; space charge, and dipolar relaxations combined (Figure 29). When there is no good contact between the electrode and the polymer, the accumulation of surface charges at their point of contact may occur, resulting in a negative current peak just after T_g_ or interfering with it (Figure 2.3 of [5]). The negative peak can be eliminated by applying a small pressure to the surface of contact, by changing the metal of the electrode of the same Figure above), by performing vacuum deposition of thin-film metallic layers, or simply by using silver lacquer to paint both surfaces of the sample and suspending it with thin copper wires between the electrodes.

The temperature of polarization is −40 °C, kept for 1.5 min under a field of 200 V/mm. The peak at −75 °C represents T_g_ for the butadiene phase. It is important to emphasize that the choice of the temperature of polarization T_p_ determines the focus of molecular or relaxational motion. For instance, the T_g_ of the styrene phase is around 80 °C; if we polarize at −40 °C, we will not be able to excite, orient, or polarize the styrene dipoles. However, that temperature of polarization activates the butadiene dipoles, and through this selective polarization, we are capable of focusing on that T_g_ transition. Polarizing closer to the peak maximum, say between −95 °C and −60 °C, would enhance the peak intensity even further. TSD allows the user to zoom in on a given transition “by putting his finger on it”, so to speak, i.e., by adjusting the temperature of polarization.

**Figure 29 polymers-17-00239-f029:**
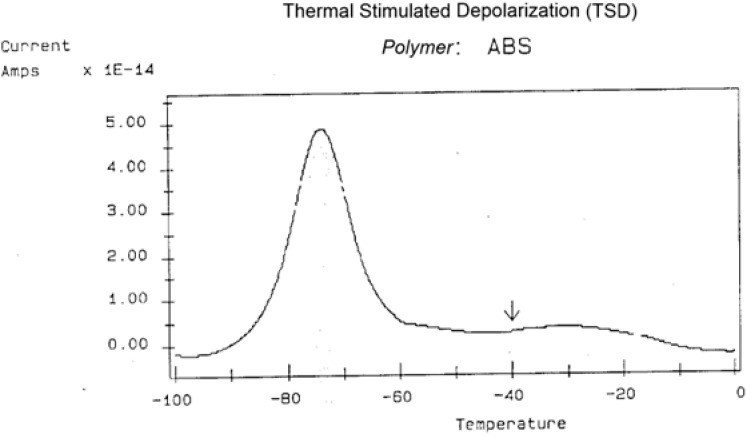
TSD depolarization curve obtained for ABS polarized at −40 °C (arrow) for 1.5 min under 200 V/mm of sample thickness. This illustration is the output curve of the TSC/RMA spectrometer by Solomat Instrument (1989). Reproduced with permission from [4], SLP Press, 1993.

### 2.2. The TSD Manifestation of T_g_, the T_g,ρ_ Peak, and of T_LL_

Some peaks of the TSD outputs are caused by the motion of the dipoles at characteristic transitions in the material, such as its T_g_. Some other peaks are due to the interactions between the fundamental morphological structure of the amorphous phase (and b-grains surrounded by F-conformers in the dual-phase model) and the electrical field, creating Wagner charges in F micro-voids. Figure 30 is a typical TSD curve for polyethylene terephthalate (PET). Two peaks are clearly visible, which we call T_g_ and T_g,ρ_. The two peaks are separated by about 50 °C in this figure, and the intensity of the higher peak, T_g,ρ_, is higher than the intensity of the first peak, T_g_. Figure 31 displays the TSD curve for polycarbonate (PC), showing that T_g,ρ_ is only 16 °C above the T_g_ peak.

The value of T_g,ρ_ observed by TSD is designated differently by various authors: Lacabanne and Boyer designate this as the T_LL_ peak [28] and associate its existence with the local order. Vanderschueren, Van Turnout, and other authors call it T_ρ_ [2,3,29,30] and speculate that it originates from the discharge of space charges delocalized in the structure. Extensive studies have shown that the relative intensity of the T_g_ and T_g,ρ_ peaks, as well as their respective position, depends on the thermal and mechanical history of the polymer, in particular physical aging. The magnitude of the T_g,ρ_ peak is also a function of the voltage field, increasing first and then leveling off as the voltage increases. The presence of the T_g,ρ_ peak is universal in the characterization of polymers by TSD. We found that peak just above T_g_ for all the polymers tested: from paints to thermoplastics, from thermosets to rubbers, and for non-polar plastics as well as polar ones. This peak is apparently absent in the characterization of the same polymers by dynamic mechanical analysis (DMA) or DEA. It is stipulated here that differential scanning calorimetry (DSC) experiments can show T_g,ρ_ under special cooling and processing conditions, which enhance its manifestation (see subsequent sections). The position of T_g,ρ_ with respect to T_g_ depends, to a large extent, on the choice of the polarization temperature T_p_, which enhances the respective magnitude and the resolution of the peaks by the effect of polarization selectivity mentioned in the previous section. Thus, it is not uncommon to observe the T_g_ and T_g,ρ_ peaks merge into a broad intense peak for certain polarization temperatures, complicating their fine analysis. The transition T_LL_ is also found by TSD, but it is located at a higher temperature, and it should not be confused with T_g,ρ_. In some instances, the mechanical history of the specimen and the temperature of polarization are such that T_g,ρ_ and T_LL_ merge or overlap, making the analysis even more complicated (Figure 32). Fortunately, the thermal-windowing process normally allows for a good separation of all the peaks, which can be isolated and deconvoluted individually. The only drawback with thermal-windowing at temperatures of polarization above T_g_ is that it modifies and partially erases the kinetic effects, which are responsible for the relaxations under investigation. In other words, the manifestation of both T_LL_ and T_g,ρ_ is favored by thermal history treatments, which bring the specimen far out of equilibrium, but this is quickly erased by annealing above T_g_, which occurs when the sample in the TSD cell is brought up to the polarization temperature. In Figure 32, the 2 mm thick polystyrene specimen is compression-molded while rapidly cooled to room temperature. A platen pressure of 226 bars is applied at 122 °C prior to and during cooling. The polarization temperature is 115 °C. Three peaks are clearly visible: T_g_ at 97 °C, T_g,ρ_ at 112 °C, and T_LL_ at 150 °C.

The intensity of the T_g,ρ_ peak is enhanced by fast cooling or any other thermo-mechanical process, which creates a state of non-equilibrium in the polymer. The intensity of T_g,ρ_ decreases with annealing time, as demonstrated in Figure 33. In this figure, PC is subjected to a compression molding treatment involving mechanical vibrations as it is being cooled. This type of treatment is called rheomolding and is described elsewhere [8,9]. In the TSD experiments shown in Figure 33, a rheomolded PC is rerun three times in a row without changing the specimen. The same polarization conditions are used each time. Each run partially erases the initial thermal history of the specimen. We focus on the intensity and position of the T_g_ and T_g,ρ_ peaks. The first trace (first polarization) displays a very intense T_g,ρ_ peak relative to the T_g_ peak, but the intensity of this peak rapidly decreases for traces 2 and 3, as the initial state of PC returns to more traditional non-equilibrium conditions (cooling in the TSD cell, although rather fast, is no match to the severe cooling conditions imposed by rheomolding). It is interesting to observe that as the intensity of T_g,ρ_ decreases, the intensity of the T_g_ peak increases, in a kind of complementary manner. The position of the T_g_ and T_g,ρ_ peaks also changes in a reverse way: as T_g_ maximum increases slightly, T_g,ρ_ decreases, giving the impression of fusion of the two peaks.

The understanding of the existence of T_g,ρ_ and T_LL_ is offered in the Discussion section (Section 3.1). As far as we are concerned, the T_g,ρ_ peak is real, with a high value, often stronger than T_g_, and located a few degrees after T_g_. In this section, which deals with the applications of TSD, it is important to introduce the existence of peaks T_g_, T_g,ρ_, and T_LL_ and suggest that their presence is universal in the context of the TSD characterization technique. It is not essential, however, to speculate about their origin. Let us just mention that Part I, Chapter 3, suggests a common kinetic origin to T_g_, T_g,ρ_, and T_LL_, and even to T_β_. The apparent complexity of the relaxation behavior arises for two reasons: Even if T_g_ and T_g,ρ_ are kinetically related, their dielectric origin is different because the effect of the voltage field on the kinetic units responsible for their respective relaxation is different. In the case of T_g_, the dipole moment’s relaxation is associated with a molecular dipole but probably not so for T_g,ρ_. The electric moment for T_g,ρ_ seems to be associated with ionic dipoles or perhaps space charges. There is much evidence to confirm that the T_g,ρ_ peak is related to the “free volume” in the sample, which suggests that either local unstable ionic dipoles or space charges get trapped in the free volume of the polymer under the influence of the voltage field and relax at T_g,ρ_. More details are provided in Section 3.1 of the Discussion where the “free volume” is defined in the morphology in terms of statistical units of the polymer chains. The difference between the characteristics of T_g,ρ_ and T_LL_ is definitive: Hydrostatic pressure lowers the value of T_g,ρ_ and increases the value of T_LL_. This is shown for T_g,ρ_ in Figure 8 of the Section 1. In this figure, polystyrene samples are prepared by compression molding under various conditions. In the top graph, a relaxed specimen is cooled under no mechanical stress at a very slow cooling rate. The Arrhenius transform of the T_g,ρ_ peak is shown at the extreme right of the graph; it is curved and can be fitted with a WLF type of equation, providing the free volume thermal expansion coefficient above T_g_, and the temperature of infinite viscosity (the values found match viscosity data well). In the middle graph, corresponding to the “pressurized” specimen, the WLF curve is shifted toward a lower temperature. Another thermo-mechanical treatment is shown in the bottom graph. Figure 34 displays the Arrhenius transforms of the T_g,ρ_ peak for several processing conditions: static pressure, rheomolding treatment, oriented sample, and relaxed polystyrene.

The relaxation mode obeys a free volume criterion for all conditions, but the content of the free volume and the mobility at a given temperature, given by the horizontal position of the WLF curve, are strong functions of the state of the polymer due to processing conditions. In some instances, the T_g,ρ_ peak is broad and contains a combined effect of molecular dipoles and free volume relaxations (Figure 35). In such cases, the thermal-windowing around the T_g,ρ_ peak produces several relaxation modes (Figure 8, bottom curve), and sometimes one or two additional compensations are observed in the (T > T_g_) region up to T_LL_.

We consider T_LL_ as the temperature, on heating, marking the end of a certain type of relaxation behavior due to cooperative kinetic interactions. Whether or not it actually corresponds to a TSD peak is irrelevant to our definition. For instance, we mentioned that T_LL_ shows up in Figure 32 as a peak at 150 °C. It might be more appropriate to categorize this peak as one of the kinetic manifestations resulting from the cooperative kinetic process already giving rise to T_β_, T_g_, and T_g,ρ_. In Part I, Chapter 3, as well as in references [25,31,32], it is suggested that the mechanism of relaxation is due to the coherence between the collective behavior defining the statistical ensemble and the local existence of dual phases (the b and F phases), which describes the interactive coupling between the conformers belonging to the macromolecules modulating both the sub-T_g_ and the (T > T_g_) kinetics. In that regard, T_LL_ is probably more at 175 °C than 150 °C in Figure 32, and the peak at 150 °C should be considered as a manifestation of the modulation by the network (collective) of b/F interactions occurring in the phase richer in free volume and responsible for the T_g,ρ_ relaxation manifestation. This point is further illustrated in Figure 36.

In this figure, we plot the variation in T_g,ρ_ with T_p_, T_g,ρ_ being considered the “first” peak observed above T_g_. When a second peak appears, which we call T′_g,ρ_, we indicate its temperature on this graph with square dots. One sees that T_g,ρ_ increases slowly and linearly with T_p_ (the slope is much less than 1), at least until T_p_ = 145 °C, a temperature at which T_g,ρ_ rises quickly to a maximum. The first split of T_g,ρ_ into two peaks occurs for T_p_ = 165 °C, and the second split occurs for 180 °C. In the case of a split, the temperatures of the two peaks are located on both sides of the “expected” T_g,ρ_ extrapolated from the lower values. The departure from the baseline at 145 °C reveals the kinetic presence of the network modulating collectively the interactions [7]. Beyond T_p_ = 170 °C, the modulation by the elastic dissipative network of the inter–intramolecular interactions between the various dipoles is over. T_LL_ would be 170 °C, according to our definition.

In Figure 37, the DSC trace for a rheomolded polystyrene showing cooperative relaxation above T_g_ is compared with a reference trace (obtained by cooling the same sample in the DSC cell after the first run). The difference between the heat flow of the two curves is significant just above the T_g_ peak, and until the T_LL_ temperature is reached. The thermal history of the rheomolded sample, revealed during the first run, creates a thermal activity above T_g_ clearly similar to what is observed by TSD (Figure 35). The thermal history is erased as the sample is heated in the DSC cell, and both the reference and rheomolded DSC traces are identical above T_LL_. We consider Figure 37 as an important illustration of the existence of T_g,ρ_ and T_LL_ using DSC. However, the sample needs to be severely out of equilibrium in order to reveal by DSC the T_g,ρ_ and T_LL_ transitions as clearly as in Figure 37. TSD, which works at a very low-frequency equivalent (Part II, Chapter 1, [5]), is capable of resolving the T_g_/T_g,ρ_ kinetic presence with far higher resolution and sensitivity (note how thermal history of the rheomolded PC of Figure 33 is still revealed after the successive runs). The thermal analyst should be gratified: the T_g,ρ_ peak gives us a new parameter to characterize polymers. It varies with the processing conditions and thus can be used to characterize the effect of processing variables (pressure, orientation, and thermal history); it is often stronger than Tg and exists for non-polar materials when the free volume can be trapped with charges. It is perhaps the only peak that one can observe to characterize thick non-polar polymers. In that case, the “T_g_” value obtained is possibly a T_g,ρ_ value several degrees higher than what would have been expected from a DSC comparison (the TSD T_g_ peak matches the T_g_ obtained in a DSC experiment at a 20 °C/min heating rate well).

### 2.3. Super-Compensations Observed in the Amorphous State of Single-Phase Amorphous Samples

The objective is to explore the complexity of the phenomenon of compensation to better understand its origin. In the case of a multi-phase system, such as a segregated blend or block copolymers, the relaxation map displays multi-compensations, each associated with the T_g_ of the corresponding phase. In the case of crystallizable polymers or liquid crystal polymers, multi-compensations are often visible both below and above the glass transition temperature of the polymer. All these apparent complexities are expected because of the presence of multiple phases in the material. Figure 38 and Figure 39 illustrate, respectively, what is expected for the compensation search to look like for, a two-phase amorphous polymer (blend or copolymer) and a single-phase amorphous homopolymer such as polystyrene. In the case of a homopolymer (Figure 39), only one T_g_ is observed. The compensation search leads to a single pair of compensation lines, one positive and one negative. The presence of two phases in the morphology is clearly revealed by two T_g_ s and thus two pairs of compensation lines (Figure 38). A comparison of the respective position of the compensation lines for the homopolymer and the block copolymer leads to a better understanding of the segregation characteristics between the phases and, in particular, the degree of interpenetration and local compatibility (Part II, Chapter 4, [5]). It is clear that, in this case, there is a direct relationship between the number of compensation lines and the number of phases in the morphology.

It will be shown in this section (e.g., Figure 55) that, for a true monophasic amorphous polymer (in this case, polystyrene), not only is it possible to observe several compensations but also to vary the number of compensations depending on the thermo-mechanical history: the processing variables, the cooling rate, the amount of pressure, vibration [33], orientation, etc. This unexpected complex behavior has remained controversial: Lacabanne and Bernes [34] have suggested that the several compensation points observed for polystyrene (and polycarbonate) were due to the presence of several amorphous phases, the indication of “local ordering”. Furthermore, these authors suggested that T_LL_ was the melting transition of these local micro-ordered structures [35,36,37,38]. In this Section 2.3, we describe the multi-compensations in monophasic amorphous systems giving rise to super-compensations, and in D3, we propose an alternative explanation for the existence of super-compensations and of T_LL_ based on the dual-phase model of the dissipative interactions.

Figure 40 is a relaxation map in the “Eyring plane” (defined in A1) for a polystyrene disk (T_g_~100 °C) very slowly cooled from 180 °C to 50 °C in a platen pressure mold with virtually no pressure applied on the specimen during the cooling stage. The mold is cooled very slowly by “Newtonian cooling”, i.e., by shutting off the power to the heater cartridges after thermal equilibrium is reached at 180 °C and letting it cool by itself, without the use of cooling fluids in the cooling channels of the mold halves. This procedure resulted in very slow cooling: it took approximately 7 h to cool the mold down to 50 °C. Table 1 gives the thermo-kinetic parameters as a function of T_p_, namely ΔH_p_(“enthalpy”), ΔS_p_(“entropy”), and ΔG_p_(“Gibbs”), as well as the value of the polarizing temperature, T_p_, and T_m_, the temperature at the maximum of the Debye depolarization peak. Figure 41 illustrates the compensation search in the EE plane (Entropy vs. Enthalpy). T_p_ varies between 60 °C and 125 °C (the window temperature is 10 °C). The numbers near the data in Figure 41 apply to the row numbers in Table 1, increasing with the polarization temperature T_p_. We clearly observe two negative compensation lines in Figure 41. The first compensation line comprises points 1, 2, 3, and 4, all with T_p_ < T_g_, and the second compensation line aligns points 6, 7, and 8, all with T_p_ > T_g_; point 5 (at T_p_ = 95 °C) seems to be “lost” between the two negative compensation lines. We will go back to the analysis of this PS_VA treatment in Figure 56 and Figure 57 and Table 5 after the study of the relaxation maps for other thermo-mechanical histories, which will have given us some insight into how to handle the analysis of these unusual and complex relaxation maps. Our first impression of the spectral lines in Figure 40 is that the first five lines, starting from the low T_p_ end, are more or less parallel. This is reflected in the compensation search by the closeness of their enthalpy values on the corresponding compensation line (from 31 to 25 Kcal/m). We also note that the entropy decreases from 24 cal/m-K to 0 (for point 4) and increases sharply to 111 for point 6.

**Figure 40 polymers-17-00239-f040:**
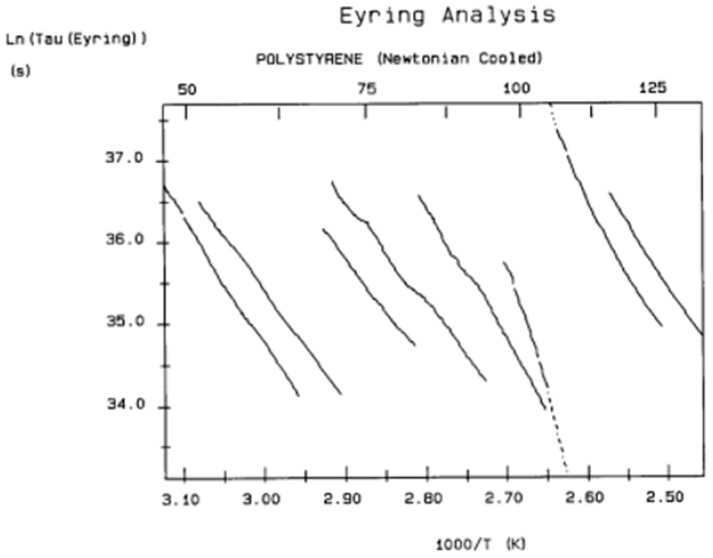
Relaxation map in the Eyring plane for a very slowly cooled compression-molded polystyrene sample with almost no pressure during cooling. The sample is called PS_VA in subsequent figures. The slope and intercept of the elementary deconvoluted relaxations provide the enthalpy and entropy values in Table 1. The data are analyzed in Figure 41, Figure 56, and Figure 57. Reproduced with permission from [4], SLP Press, 1993.

**Table 1 polymers-17-00239-t001:** Thermo-kinetics for sample PS_VA of Figure 40.

Row #	T_p_ (K)	T_m_ (K)	Enthalpy △H_p_	Entropy △S_p_	△G_p_
	(K)	(K)	(Kcal/m)	cal/m-K	Kcal/m
1	333.2	344	30.8906	23.6368	23.0148
2	338.2	349.8	27.1823	11.1923	23.3971
3	348.2	367.5	25.9793	4.211	24.5130
4	353.2	372.7	25.0293	0.0593	25.0084
5	368.2	380.5	32.4905	18.5915	25.6451
6	378.2	382	67.5262	111.1845	25.4762
7	388.2	411.9	38.8583	23.3365	29.7991
8	398.2	418.1	30.5001	5.8074	28.1876

**Figure 41 polymers-17-00239-f041:**
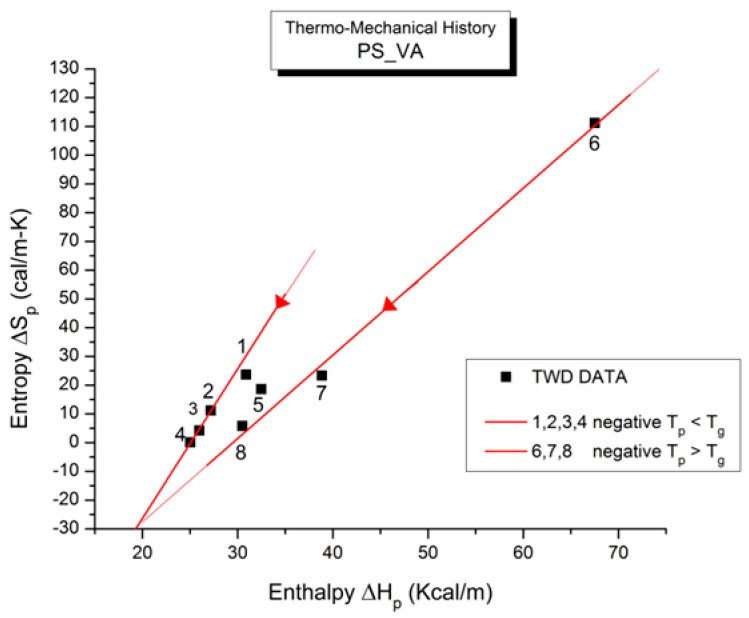
Compensation search in the EE plane for the data of Figure 40. The numbers near the data relate to the row position in Table 1. Reproduced with permission from [4], SLP Press, 1993.

Figure 42, Figure 43, Figure 44, Figure 45, Figure 46 and Figure 47 relate to the same polystyrene but mechanically pressurized during a fast cooling treatment. The mechanical treatment includes the application of vibrational positive pressure to induce specific thermal history patterns, a process known as “rheomolding” [39]. The mold is cooled in approximately 30 s. The rate of cooling when the glass transition temperature is crossed is around 7 °C/s, considerably faster than for the PS_VA treatment in Figure 40. Figure 42 is the relaxation map in the Eyring plane. Figure 43 is the compensation search in the EE plane based on the entropy and enthalpy data of Table 2. The thermo-mechanical treatment is called “PS_RL”. Figure 43 appears complex, unlike what is observed for stable amorphous states for which two compensation lines can be drawn passing through the data: one positive compensation line for T_p_ < T_g_ relaxations and one negative compensation line for T_p_ > T_g_. This cannot be the correct solution in Figure 43 because the two lines only solution would not join consecutive T_p_ data points, which is a sine qua non condition to define compensation. In fact, as Figure 44, Figure 45, Figure 46 and Figure 47 will demonstrate, the situation is only apparently more complex because we find several correlations and a network structure between the six compensations (three positive and three negative) that are necessary to comprehend the relaxation maps in Figure 42 and Figure 43. Figure 44, Figure 45, Figure 46 and Figure 47 illustrate our step-by-step method to clarify complex compensation searches, which we will only explain for PS_RL.

In the Eyring plane, the Debye relaxation times are normalized by the vibrational energy of the atoms (kT/h), where k is the Boltzmann constant, T is the absolute temperature, and h is the Planck constant. The slope and intercept of log τ _(Eyring)_ vs. 1/T directly provide the activation enthalpy, ΔH_p_ = slope*k, and the activation entropy, ΔS_p_= intercept*k, of the elementary Debye depolarization process: (k is replaced by R, the gas constant, equal to 1.987 cal/m-K, to express the activation energies per mole instead of per molecule).

**Figure 43 polymers-17-00239-f043:**
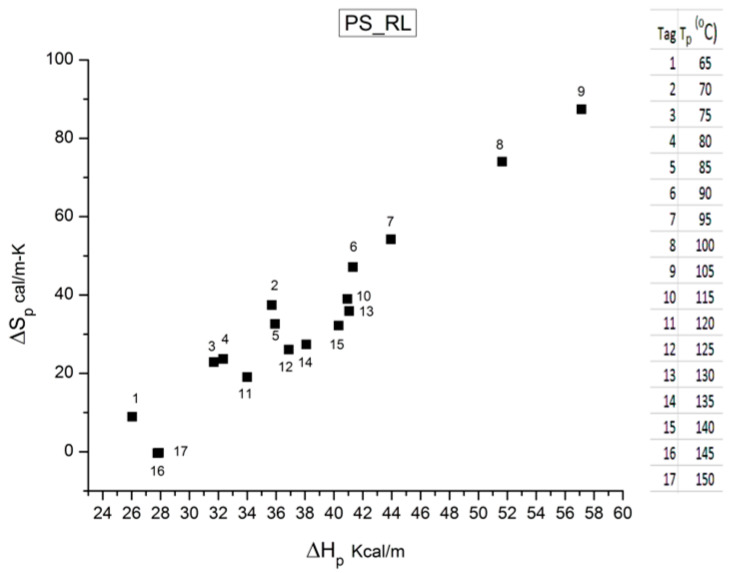
Compensation search in the EE plane for the PS_RL sample of Figure 42. The analysis appears complex. The numbers near the data (black squares) are the row numbers in Table 2 corresponding to the T_p_ value (shown in the inset) during TWD. For a simple amorphous state, such an EE plot has two straight lines going through the data, a positive and a negative compensation line intersecting at T_g_, but we cannot represent such a simple solution since compensation lines should join consecutive increasing or decreasing T_p_ points for positive or negative compensations, respectively. The finding of the true positive and negative compensation lines for this complex situation is shown in Figure 45 and Figure 46.

A compensation search looks for an affinity between successive interactive coupling relaxation modes activated at different polarization temperatures T_p_. In Figure 43, the Eyring relaxation map of Figure 42 is converted to the EE plane variables (entropy vs. enthalpy), and the affinity between the relaxation modes is determined by the alignment on a straight line (the “compensation”) of the thermo-kinetic coordinates (ΔH_p_ and ΔS_p_) for neighboring T_p_ values. In Table 2, we see that T_p_ varies every 5 degrees from 65 °C to 150 °C (17 values). Here, rows 1 and 2 for T_p_ = 65 and 70 °C correspond to the two first relaxation lines on the left of Figure 42. These two relaxation modes appear to cross in that figure at an Eyring relaxation time, τ_c_, and a temperature T = T_c_. In the EE plane (Figure 43), these two relaxation modes are points #1 and 2. The line (1,2) is the compensation line that allows for the calculation of the coordinates of the compensation points of these two elementary relaxations in Figure 42. This is illustrated in Figure 44, which shows a plot of ΔH_p_ vs. ΔS_p_ in which the x and y axes are swapped to allow for an easy determination of the compensation point in the ΔG plane: T_c_, ΔG_c_. In Figure 44, the value of ΔG_c_ is simply the intercept of line (1,2), and the T_c_ value (in Kelvin degrees) is equal to 1000 times the slope of line (1,2). In Figure 43, as T_p_ increases beyond point # 2, one sees that #3 is not located on line (1,2) but rather below it, although its T_p_ has increased; a clear reversal of the direction indicates the presence of a new compensation with a change in its “sign”, here between points #2 and #3. The first compensation (1,2) is called “1^+^”since it is positive, while (2,3) is called “1^−^” since it is the first negative compensation (we also use the appellations C_1_^+^ and C_1_^−^ to designate these compensations). As T_p_ continues to increase, points #3 to #9 are aligned on a single straight line forming the second positive compensation, “2^+^” (or C_2_^+^). The second negative compensation lines down points #9, #10, and #11 and is designated “2^−^” (or C_2_^−^), and so on and so forth for the remaining points sub-grouping in compensations “3^+^” and “3^−^” or even in “4^+^” and “4^−^” in some instances. The rule followed to define the various compensation lines is that only successive points can participate in a compensation line. Sometimes, we hesitate between two options because of the errors that occur during the determination of the coordinates of the points: this happens at larger T_p_ values where the relaxation of non-equilibrium samples occurs faster. The compensation search is illustrated in Figure 45 for positive compensations and Figure 46 for negative compensations.

**Table 2 polymers-17-00239-t002:** Thermo-kinetics for sample PS_RL.

Row #	T_p_	T_m_	Enthalpy ΔH_p_	Entropy ΔS_p_	ΔG_p_
	°C	°C	(Kcal/m)	cal/m-K	Kcal/m
1	65	69.7	26.0444	8.917	23.03045
2	70	70.2	35.7053	37.4304	22.86667
3	75	79.8	31.6927	22.8775	23.73133
4	80	84.6	32.343	23.6549	23.99282
5	85	88.8	35.9299	32.599	24.25946
6	90	90.2	41.3195	47.1144	24.21697
7	95	90.6	43.9499	54.2181	23.99764
8	100	94.6	51.6457	74.0286	24.03303
9	105	98.8	57.1351	87.381	24.10508
10	115	113.5	40.9335	38.9967	25.80278
11	120	120.4	34.0003	19.0242	26.52379
12	125	121.9	36.8776	26.0567	26.50703
13	130	125.4	41.0596	35.889	26.59633
14	135	129	38.0933	27.3621	26.92956
15	140	132.5	40.3364	32.172	27.04936
16	145	137.3	27.7921	−0.3743	27.94856
17	150	138.5	27.8984	−0.3003	28.02543

**Figure 44 polymers-17-00239-f044:**
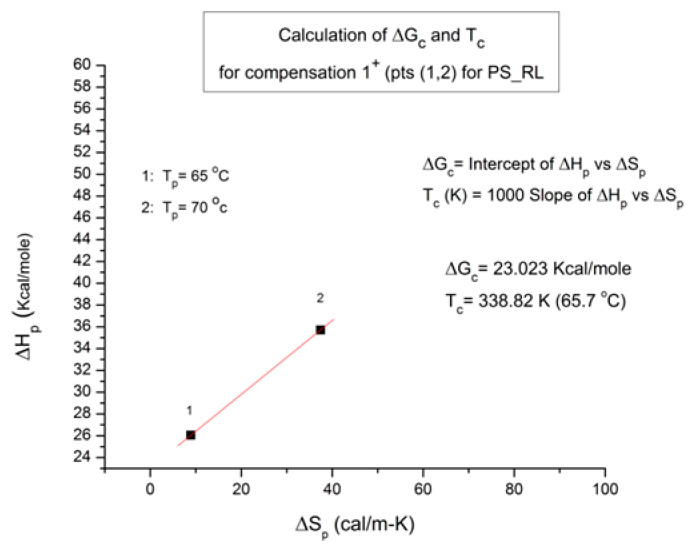
Compensation search **in the EE plane (Enthalpy vs. Entropy) plane** for PS_RL to find the compensation(s) in Figure 42. Several positive and negative compensations emerge from what initially looked complex (in Figure 43). The down arrows indicate positive compensations, whereas the up arrows indicate negative compensations.

Figure 45 reveals that the positive compensation lines (in red) appear to converge on the ΔS_p_ = 0 axis. The corresponding value of ΔH_p_ at the intercept point is 23 kcal/mole. This point is fully independent of the value of T_p_: it is a **super-compensation**, i.e., the compensation of all the positive compensation lines for sample PS_RL. When we say for “all” positive compensations, we should add that this is 100% sure for compensations C_1_+ and C_2_+, but that a different split of the points is possible for C_3_+, as shown in the inset of Figure 45: either C_3_+ = (11,12,14,15) or C_3_+ = (11,12,13) and C_4_+ = (14,15). Figure 45 shows the first choice, which has only three compensation lines.

**Figure 45 polymers-17-00239-f045:**
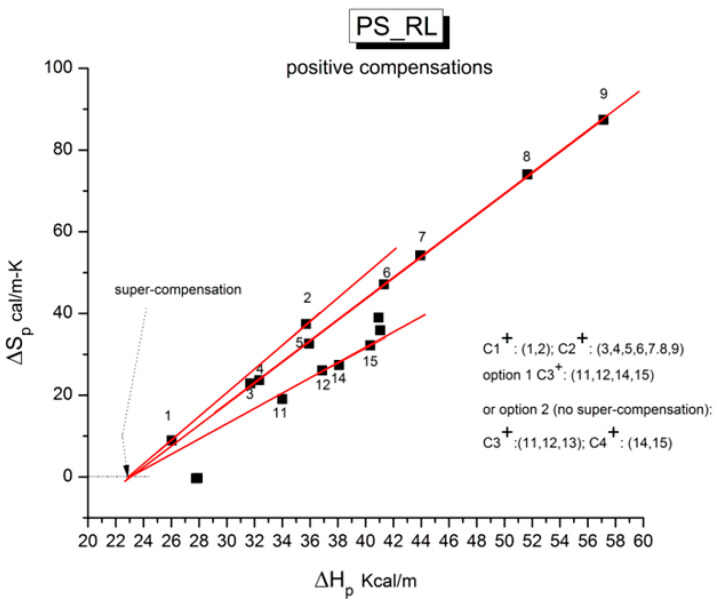
Compensation search in the EE plane for PS_RL showing the **positive compensation** results only (lines joining successive T_p_ points with increasing values of ΔS_p_ and ΔH_p_). The 3 positive compensation lines converge to a “super-compensation” point (whose y-coordinate is ΔS_sc(+)_ = 0) for option 1. For option 2, with 4 compensations, only 3 of them super-compensate. The analysis for the negative compensations is illustrated in Figure 46.

Figure 46 demonstrates the points interacting to generate negative compensation lines. This analysis is a little bit more difficult because there are fewer points to define the lines accurately. Nevertheless, it is possible to draw with confidence three negative compensation lines (option 1) or four compensation lines (option 2), and in the latter case, they themselves converge with confidence to a super-compensation (ΔH_c_ = 18.8 Kcal/m, ΔS_c_ = −23 cal/m-K).

**Figure 46 polymers-17-00239-f046:**
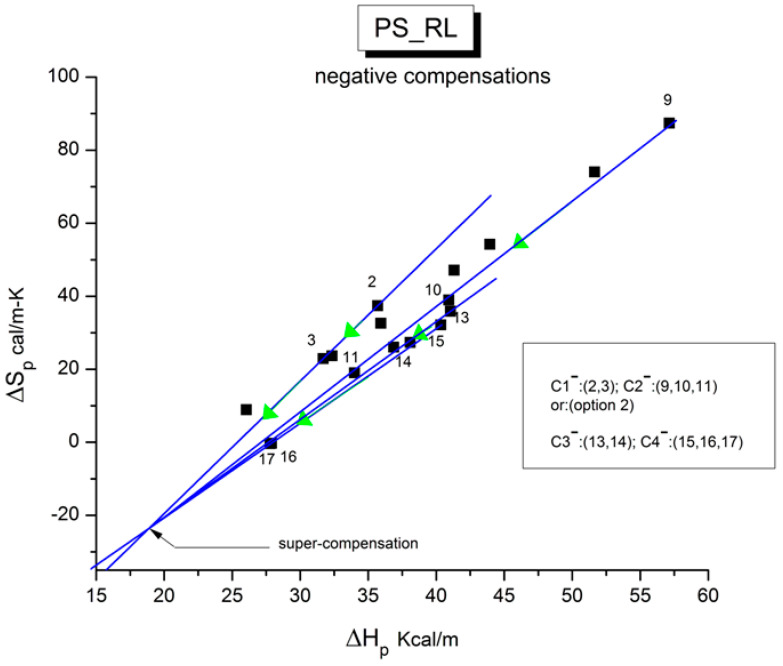
The compensation search in the EE plane for PS_RL showing only the **negative compensation** results of the analysis conducted in Figure 43. In option 2, the 4 negative compensation lines converge to a single point, a super-compensation point whose ΔH_pc(−)_ = 18.8 Kcal/m and Δ_Spc(−)_ = −23 cal/m-K. For option 1, with C_3_− = (14,15,16,17)), there are only 3 negative compensations and the super-compensation is defined with less confidence (r^2^ = 0.988). For option 2, with C_3_− and C_4_− defined in the inset, the convergence to a super-compensation is established with confidence (r^2^ = 0.9987).

In Table 3, we summarize the results of the regressions of the compensation lines (ΔS_p_ vs. ΔH_p_) shown in Figure 45 and Figure 46, with ΔS_p_ in cal/m-K and ΔH_p_ in kcal/m.

**Table 3 polymers-17-00239-t003:** Intercept and slope of the compensation lines in Figure 45 and Figure 46 (limited to the options studied in Figure 48, Figure 49, Figure 50, Figure 51, Figure 52, Figure 53, Figure 54 and Figure 55).

ΔSp (cal/m-K) vs. ΔHp (Kcal/m)							
PS_RL							
Positive Compensations		Intercept	Slope	Negative Compensations		Intercept	Slope
(1,2)	1(+)	−67.951	2.95142	(2,3)	1(−)	−92.0656	3.6268
(3–9)	2(+)	−59.1929	2.57205	(9,10,11)	2(−)	−81.8557	2.96034
(11,12,13)	3(+)	−62.0482	2.38628	(14,15,16,17)	3(−)	−73.74	2.6381

**Figure 47 polymers-17-00239-f047:**
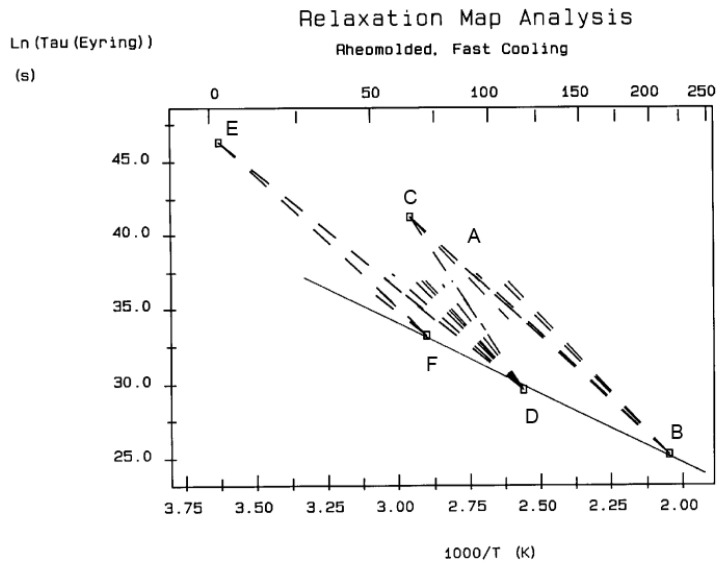
Schematic reconstruction for PS_RL of the compensations of the spectral lines in Figure 42 following the compensation search results of Figure 45 and Figure 46. This graph shows the full Debye spectral lines for the 3 positive compensations (with F, D, and B indicating the aligned compensation points); the spectral lines for the negative compensations are only visible for the lower T_p_ compensation points (E and C) and not for compensation point A.

Figure 47 represents a summary of our complex compensation search, displaying the interactive coupling map of the relaxations revealed at different T_ps_. The positive compensation points are shown at F, D, and B in order of increasing T_p_. The line passing through F, D, and B is the super-compensation line whose intercept and slope provide the coordinates of the super-compensation point of the positive compensations. Note that we did not draw a line through the negative E, C, and A compensation points because the spectral lines for A were not drawn. Yet, this negative super-compensation line can be imagined passing through E and C and would appear to be parallel to the positive super-compensation line FDB. In other words, all the compensation points and the super-compensation lines appear to create a macro-network where the local spectral lines collectively belong, showing their collective dependence. We will explain this concept in the next figures below (Figure 48, Figure 49, Figure 50, Figure 51, Figure 52, Figure 53, Figure 54 and Figure 55). For this, we will focus on the ΔG_p_(T) plane representation of the TWD relaxation map ([16], p. 79 of [5]), where ΔG_p_(T) is the free activation energy of depolarization at the T of the elementary Debye relaxation after polarization at T_p_ according to the TWD protocol, namely ΔG_p_(T) = ΔH_p_ − T ΔS_p_, in the ΔG plane. Table 3 provides the intercept and slope of the linear relationship between ΔS_p_ and ΔH_p_ for the compensation lines. The free energy at each of the compensation points, ΔG_c_, and the temperature of compensation, T_c_, can be calculated from the value of the slope (S_p_) and the intercept (I_p_) of the compensation lines when they are rewritten as ΔH_p_ = I_p_ + S_p_ ΔS_p_; with the conversion of the energy units for enthalpy and entropy to become coherent, we obtain the following: ΔG_c_ = I_p_ and T_c_(K) = 1000 S_p_. Their values are tabulated in Table 4 for each of the positive and negative compensations.

**Table 4 polymers-17-00239-t004:** Values of ΔG_c_ and T_c_ for the 3 positive (+) and the 3 negative (−) compensation points. These compensation point coordinates are the references that characterize the sub-groups of interactions, each comprising the relaxation modes belonging to the sub-group.

	△G_c_ (Kcal/m)	T_c_ (°C)
Comp(1)^+^	23.02	65.7
Comp(1)^−^	25.38	2.57
Comp(2)^+^	23.01	115.6
Comp(2)^−^	27.65	64.6
Comp(3)^+^	26	145.9
Comp(3)^−^	27.95	105.9

Figure 48 is a plot of ΔG_c_ vs. T_c_ for the six compensations in Table 4, using a color change to differentiate the positive (red, bottom) from the negative (blue, top) compensation concave curves. One sees that T_c_ increases as the T_p_ range of the points belonging to the sub-group increases. The trace going through the points is monotonous for both “polarities”, but their curvature is the opposite, concave downwards for the top curve and concave upwards for the bottom one. The two traces of Figure 49 link points with the same polarity, positive or negative, just like Figure 45 and Figure 47, separate the polarities, trying to determine correlations within a given polarity: we did find the super-compensation characteristic of the compensation lines within each polar group. In Figure 49, Figure 50, Figure 51, Figure 52 and Figure 53, we investigate the possible correlations between the compensation points across polarities. For instance, in Figure 49, we show that the lines joining the “cross-compensation” points, such as C_1_^+^ and C_3_^−^ or C_3_^+^ and C_1_^−^, cut each other at a point belonging to the line joining C_2_^+^ and C_2_^−^. In other words, the point designated C_o_ in Figure 49 is a super-compensation point for the cross-polarity network of compensation points, i.e., the vertex of a pencil of lines formed by the 3 cross-polarities lines C_1_^+^C_3_^−^, C_2_^+^C_2_^−^, and C_1_^−^C_3_^+^. This appears to be quite a remarkable property of the depolarization process of a rheomolded sample occurring over time as the temperature follows the TWD sophisticated protocol (Figure 3) and at the same time releases the internal stress induced by the thermal–mechanical treatment. The coordinates of C_o_, the super-compensation point, are ΔG_co_ = 25.66 Kcal/m and T_co_ = 87.43 °C, which is 12.6 °C below the T_g_ of PS.

**Figure 48 polymers-17-00239-f048:**
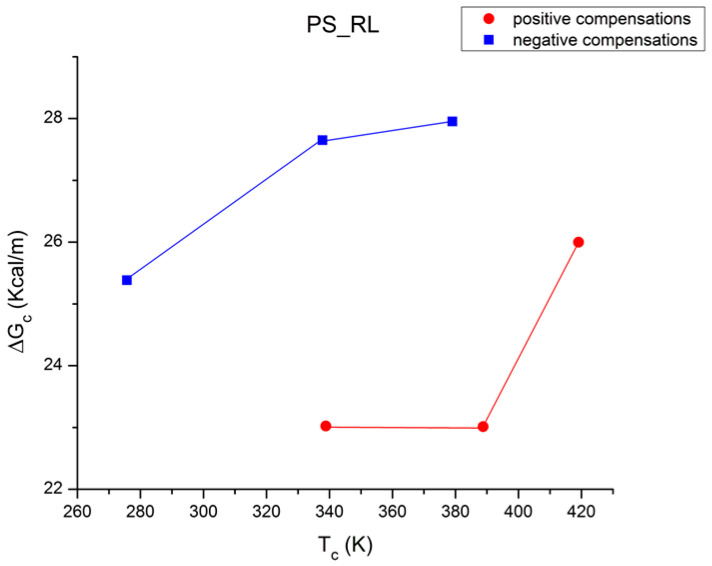
Plot of ΔG_c_ vs. T_c_ for the 6 compensations in Table 4. Red, bottom curve: positive compensation points; blue, top curve: negative compensation points.

**Figure 49 polymers-17-00239-f049:**
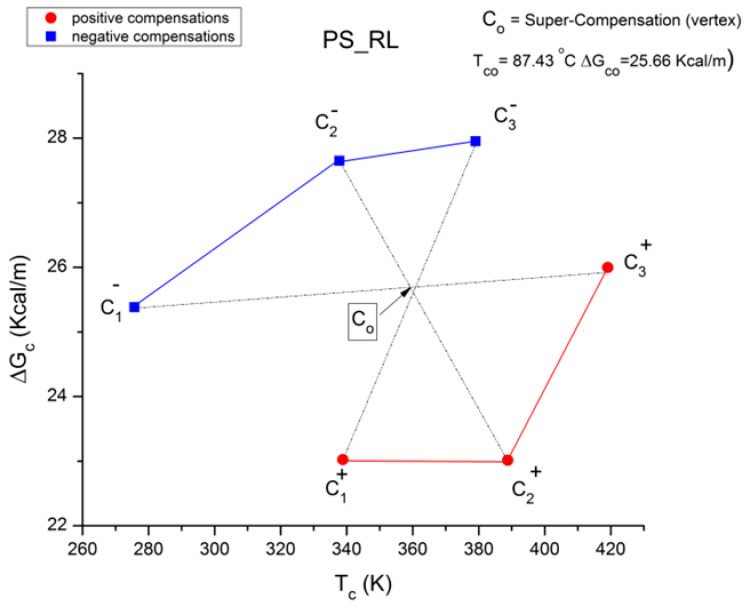
Possible correlations between the compensation points across polarities.

Figure 50 is an extension of our exploration in Figure 49 of the geometrical features characterizing the network of the compensation points. We expanded the number of links to include C_3_^+^C_3_^−^ and C_1_^+^C_1_^−^, in addition to C_2_^−^C_2_^+^, which was already included from the previous figure. We also extended these three links so that they could cut each other. C_1_^−^C_1_^+^ cuts C_2_^+^C_2_^−^ at point O_12_, and C_3_^−^C_3_^+^ cuts C_2_^+^C_2_^−^ at point O_32_. The new remarkable feature is that O_12_ and O_32_ are found vertically below and above C_3_^+^ and C_1_^−^, respectively. This may indicate that the position of the points in the geometrical network responds to symmetrical rules of dependence. We are aware of the possible errors in the determination of the coordinates of the compensation points, which is carried out at the beginning by regressions of the Debye elementary relaxation lines in the Eyring plane (Figure 42), far from being perfect straight lines even visually, followed by new regressions in the EE planes after the selection of the compensation sub-groups (Figure 45 and Figure 46). This makes the quasi-perfectness standing of the geometrical features in Figure 50 almost impossible to accept. At least some imperfection should exist. This imperfection may be actually present and could have resulted, in fact, in slightly distorting the network symmetry so that C_1_+ should align horizontally with C_2_^+^, C_2_^−^ should also align horizontally with C_3_^−^, and C_3_^−^ should align vertically with C_2_^+^. In Figure 50, these alignments are almost there; for instance, C_2_^−^ already aligns vertically with C_1_^−^, in addition to O_32_ with C_1_^−^ and O_12_ with C_3_^+^, as already mentioned above. This is as if we had discovered a crystal-like perfection of the alignment of the compensation points in the network of compensations. This finding is new and was not presented in [5], although the experimental evidence was already included (II.4.2 of [5]). Obviously, these new findings remain at the stage of speculation, and the standard scientific procedure should be applied to affirm or disapprove such new discoveries. Other geometrical particularities of the “structure of the compensation network” are noticeable in Figure 51.

-The line (black) joining the middle of the segment C_1_^−^C_1_^+^, M_1_ to the middle of C_3_^−^C_3_^+^, M_3_ passes through C_o_, the vertex of the pencil of lines C_1_^+^C_3_^−^, C_1_^−^C_3_^+^, and C_2_^+^C_2_^−^.-Line C_1_^−^C_2_^−^ is parallel to lines C_1_^+^C_3_^+^ and M_1_C_o_M_3_.-distance O_32_C_2_^−^ = distance C_o_O_12._


**Figure 50 polymers-17-00239-f050:**
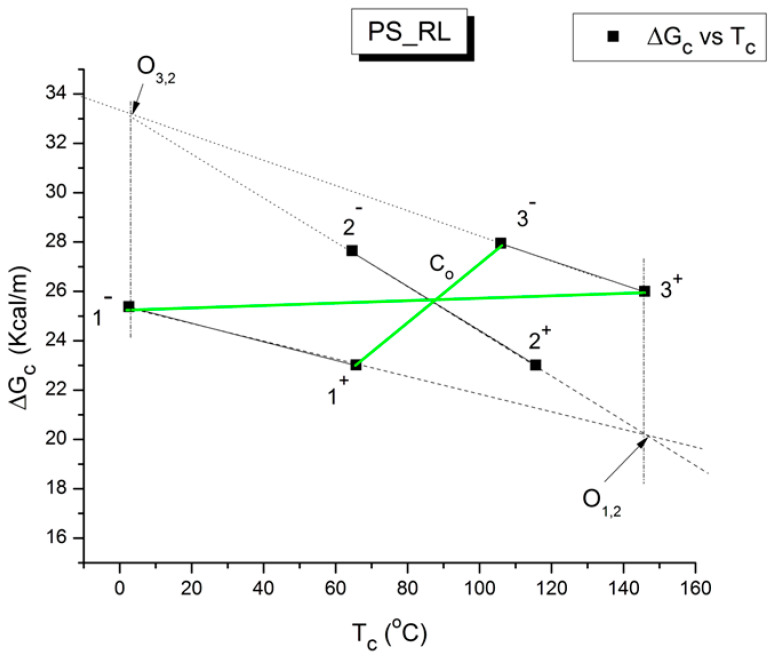
Other possible correlations between the compensation points across polarities. See the text.

**Figure 51 polymers-17-00239-f051:**
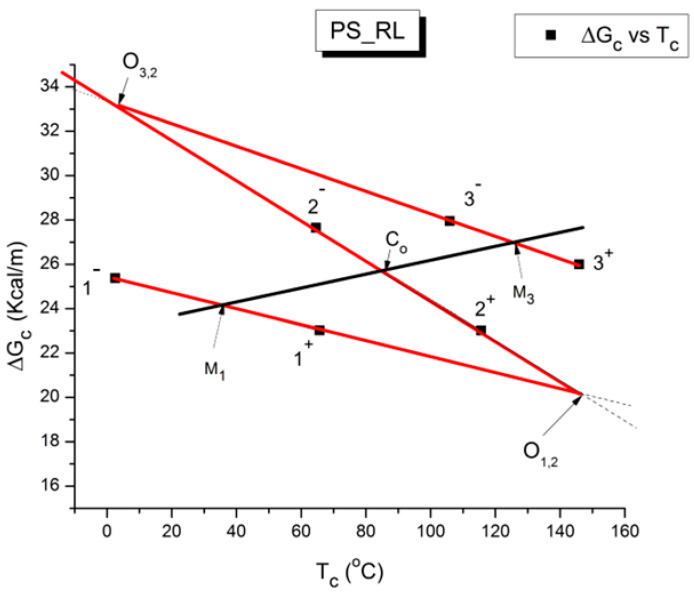
Z-structure-type correlations between the compensation points across polarities.

Another important particularity of the network of compensation points that is of great significance for the properties of the amorphous state of polymers is its ability to determine the T_g_ of the amorphous phases in polymers. We will refer to both Figure 52 and Figure 53 to explain this important finding. In Figure 52, we redrew the same graph that was used in a couple of previous figures but removed all the unnecessary details in the previous graph to reveal the presence of the so-called “Z structure” of the network of compensation points. The illustrated graph is actually a Z looked at in a mirror, which is the reason we called it the ZIM structure. A similar “Z structure” is already observed at the scale of the elementary Debye relaxations to characterize a T_g_ transition (Figure 11), but in Figure 52, we are working at a different scale, the scale of the compensation lines, not the elementary relaxations. Nevertheless, we observe a Z structure associated with the three transitions characterizing phase 1, phase 2, and say phase 3; in a way, Figure 52 is the second higher stage of a Russian doll assembly that repeats itself at different scales.

**Figure 52 polymers-17-00239-f052:**
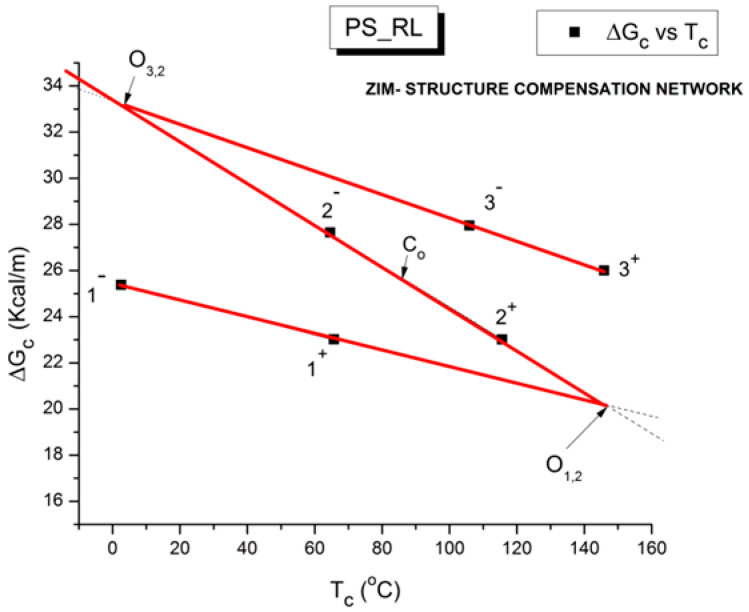
Z-structure (ZIM) compensation network for PS_RL.

Figure 53 is a plot of ΔS_p_ vs. ΔH_p_ limited to the positive and negative compensations C_2_^+^ and C_2_^−^, with red and blue lines passing through the positive or negative data points, respectively. It can be shown [16] that the intersection of these compensations corresponds to the T_g_ state of the amorphous state relaxing via the process of TWD. We visually see that the intersection occurs approximately at ΔH_p_ = 60 Kcal/m and ΔS_p_ = 90 cal/m-K. These values are the enthalpy and entropy of activation of the mode of relaxation triggered by t polarization at T_p_ = T_g_: we designate them ΔH_g_ and ΔS_g_. In previous communications (p. 73 of ref. [5] and ref. [16]), we have also explained how to find the value of T_g_ from the thermo-kinetics values of the amorphous phases of the positive and negative compensations. The first method is based on the assumption that at T_g_: ΔG_g_= (ΔG_c_^+^ + ΔG_c_^−^)/2 and since for all T_p_, then ΔG = ΔH_p_ − T ΔS_p_ for T_p_ = T_g_, we have ΔG_g_ = ΔH_g_ – T_g_ ΔS_g_. In other words, we have the following:


(13)
$$T_{g}=\frac{\Delta H_{g}-(\Delta G_{c}^{+}+\Delta G_{c}^{-})/2}{\Delta S_{g}}$$


Equation (13) summarizes the first method to find the value of Tg from TWD to characterize an amorphous phase. It can be used in conjunction with the values of ΔG_c_^+^ and ΔG_c_^−^ from Figure 52, applied to compensations 1, 2, and 3, and the values of ΔH_g_ and ΔS_g_, which are calculated similarly to the way explained for phase 2 in Figure 53 to find the T_g_ for phases 1 and 3 besides phase 2.

**Figure 53 polymers-17-00239-f053:**
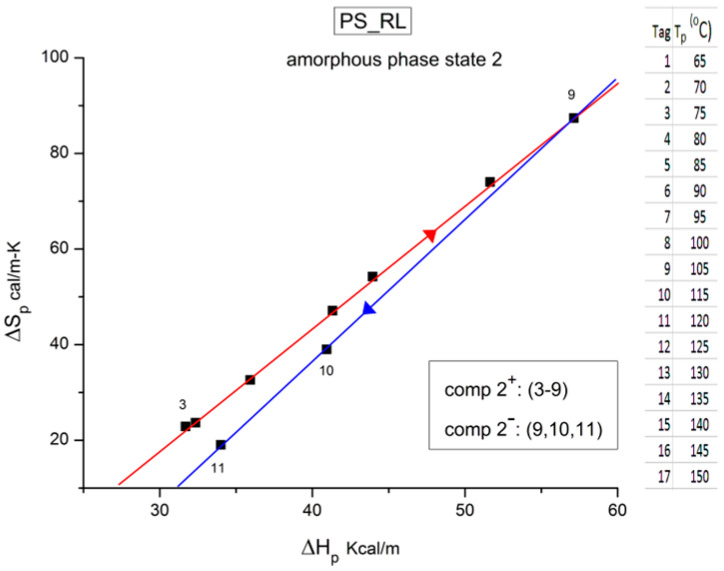
The two compensation lines, “positive” (points 3 to 9 at the top) and “negative” ((9,10,11) at the bottom) define the behavior of amorphous phase “2”. Their intersection occurs at T_g_, providing ΔH_g_ and ΔS_g_.

The second method to determine T_g_ from the results in a relaxation map is explained with regard to Figure 54 and Figure 55. Figure 54 is a plot of ΔG_p_ (Kcal/m) at various T_p_ values, where ΔG_p_ is simply the calculated value of ΔG = ΔH_p_ – T ΔS_p_ for T = T_p_. ΔG(T) is the description of the relaxation of a single Debye relaxation in the ΔG plane during the heating step in Figure 3. Notice the difference between ΔG_p_ and ΔG_c,_ the latter being the value of ΔG extrapolated at the point of compensation between several relaxations. For a polymer without any singular treatment to process it, the ΔG_p_ vs. T_p_ behavior is typically linear and almost invariably simple: ΔG_p_ ~ 0.07 T_p_ (K) in Kcal/m. This reference behavior, ΔG_pe_, is represented by the red dashed line in Figure 54; subscript “e” in ΔG_pe_ refers to the pure dielectric origin of the activation of the dipoles, resulting in a local dis-equilibrium, and of its return to equilibrium during the depolarization process. The PS_RL rheomolded sample in Figure 54 (the black squares) was purposely brought out of equilibrium to reveal the complex mechanisms of internal motions during the TWD process when the samples return to their equilibrium state during their heating or thermal annealing relaxation stages. The three stages of the relaxation process are indeed visible in Figure 54. It is clear that the sample is “sustaining” its initial non-equilibrium state to a persisting degree if we compare the evolution of the squares to the dashed reference line representing the equilibrium state. Yet, opening a short parenthesis, we should note that the highest temperature for T_p_ in Figure 54 does not reach the T_LL_ value of the sample, which increases beyond 160 °C (the classical value for PS) by the mechanical treatment during cooling [10,11,12].

**Figure 54 polymers-17-00239-f054:**
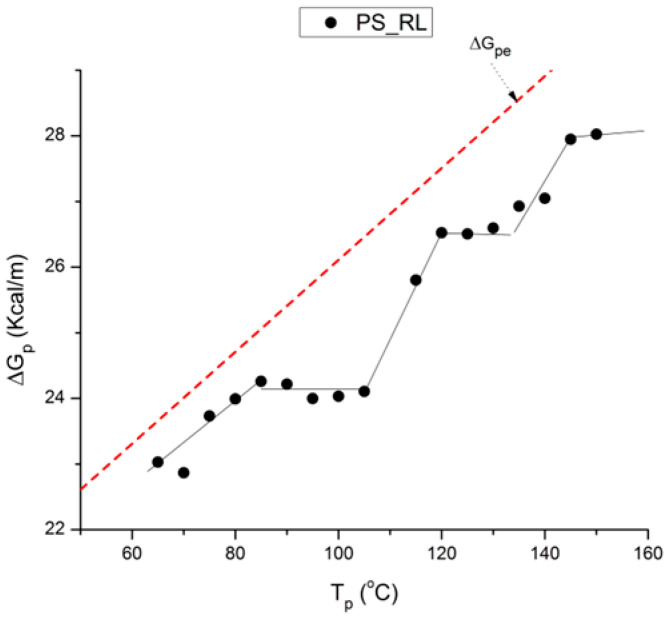
ΔG_p_ vs. T_p_ for PS_RL. The dashed red line is the expected behavior for a stable sample (with no internal stress before polarization). ΔG_p_ = ΔH_p_ – Tp(K)ΔS_p_.

Figure 55 shows the superposition of the graphs in Figure 52 and Figure 54. The ZIM structure of Figure 52 is here visible as red dashed lines. The blue line has the same slope as the dashed line of Figure 54 but is shifted by regression to fit the square data points. In other words, the blue line is obtained by linear regression of the squares forcing the value of the slope to equal 0.07 Kcal/m-K.

**Figure 55 polymers-17-00239-f055:**
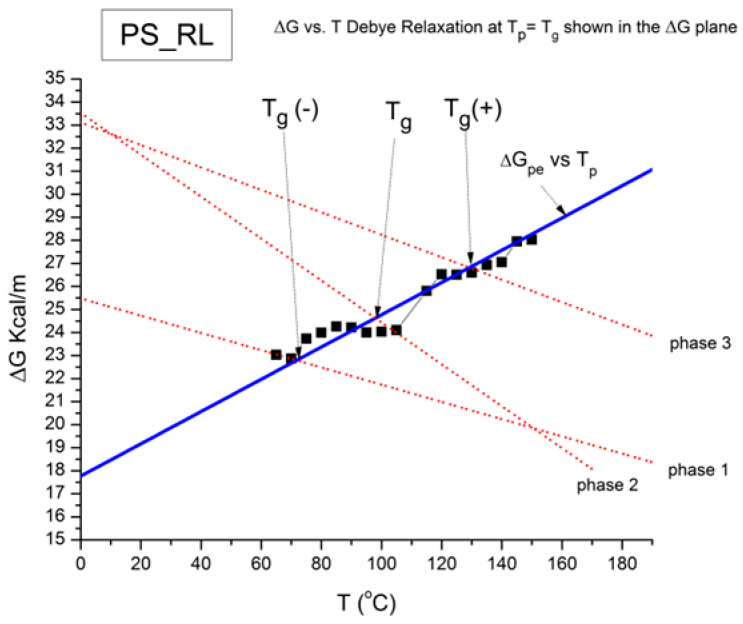
This figure illustrates the superposition of the graphs in Figure 52 and Figure 54. The ZIM structure of Figure 52 is here visible as red dashed lines. The blue line has the same slope as the dashed line of Figure 54 but is shifted by regression to fit the square data points. See the text.

According to the second method for the determination of a T_g_ transition from a TWD relaxation map [16], the value of T_g_ is at the cross-section of the compensation line of the Z structure pertinent to the relaxation process analyzed with the ΔG_p_ vs. T_p_ straight line. Figure 55 illustrates the method to determine T_g_ for compensation 1 (amorphous phase 1 with T_g_(−)), compensation 2 (amorphous phase 2 with T_g_), and compensation 3 (amorphous phase 3 with T_g_(+)). Note that, contrary to what we suggested in a previous communication (p. 495 of ref. [5]), we are affirming in this paper that T_g_(−) is not associated with T_β_ transition, nor is T_g_(+) associated with T_LL_: the visible breaks are parented to a true fragmentation of T_g_, or actually T_α_, the mechanical manifestation of T_g_, due to the initial conditions at the beginning of the thermal–mechanical treatment. In the Discussion (Section 3.2), we will show that simulations with the dual-phase model of the interactions succeed in generating positive and negative compensations, multi-compensations, super-compensations, and fragmented behavior at T_g_. The results below summarize our findings to describe T_g_(−), T_g_, and T_g_(+) in Figure 53, Equation (13), and Figure 55:

                           ΔS^−^_g_ = 37.42 cal/m-K, T^−^_g_ = 34.2 °C (method 1) or 72.9 °C (method 2);                           ΔH^−^_g_ = 35.70 Kcal/m;                           ΔS_g_ = 90.93, T_g_ = 90.2 °C (method 1) or 100.0 °C (method 2);                           ΔH_g_ = 58.37;                           ΔS^+^_g_ = 48.72, T^+^_g_ = 126.0 °C (method 1) or 129.0 °C (method 2);                           ΔH^+^_g_ = 46.42.

The only noticeable discrepancy between the two methods to calculate T_g_ is for T_g_(−), and we have no explanation as to why. The two methods are compatible at a 0.5-degree error for samples in a state of equilibrium [4], an empirical rule that does not appear to apply for samples out of equilibrium due to their processing history.

We propose that these “break transitions” of T_g,_ revealed by multi-compensations structuring as a Z network, correspond to different dissipative states of the interactions between the dual-conformers of the amorphous matter. The T_β_ transition in PS, and in other amorphous polymers, is indeed a local property of the dual-conformers, corresponding to the lower horizontal branch of the Z structure at the scale of the elementary relaxations (lower scale). The T_g_(−) break is different because it is the lower branch of a Z structure of compensations of the dual-conformers (higher scale), not of the dual-conformers themselves (lower scale). The same reasoning applies to the case of T_LL_ and T_g_(+): they are equivalent but not the same because working at different scales. The T_LL_ transition is observed in a relaxation map at the T_p_ temperature that corresponds to the upper horizontal branch of a Z structure involving a lower scale. The T_g_(+) break is the upper branch of a Z structure, apparently the equivalent of T_LL_ but working at the higher scale level of the compensation network. Section 3.2 and Section 4 will explain what we really mean by “a different scale” in a dual-phase dissipative context [7].

Finally, we present a plot of ΔS_p_ versus T_p_ for PS_RL in Figure 56. For a sample in a state of internal equilibrium, there is only one T_g_, and the positive and negative compensations that cross at T_g_ (like in Figure 53) are transposed, in a ΔS_p_ vs. T_p_ plot, to two hyperbolic functions on both sides of T_g_ sharing the same infinity axis at T_p_ = T_g_ [16]. Figure 56 applies to a sample in non-equilibrium, showing the fragmentation of T_g_ in three T_g_s values and the conversion of the hyperbolic functions to linear ones. There are no distinctions in behavior for either T_g_(−), T_g_, or T_g_(+). This confirms what was hinted before, namely the fragmented transitions perceived below and above T_g_ have the same relaxation origin as T_g_ itself (see Section 3.2 and Section 4). Another observation in Figure 56 is the sharp discontinuity observed between the fragmented transitions in a way rarely observed in physics except at critical transitions generated by chaotic behavior. This comment should be remembered when a dual-phase explanation of the fragmentation of T_g_ in the TWD experiments is provided in the Discussion (Figure 68).

Let us now go back to the analysis of the slowly cooled sample PS_VA (Figure 41). We said earlier that two negative compensations are clearly visible (Figure 41) and that point 5 (T_p_ = 95 °C) was “lost”. In view of what we learned from the analysis of PS_RL, we can now reexamine the situation (Figure 57). Knowing that positive and negative compensations alternate to define the state of an amorphous phase with a T_g_ transition at their crossing, point 5 is in fact located on a positive compensation that linearly lines up points 4, 5, and 6. The positive compensation line is not defined with certainty in this case since we only have one intermediate point between points 4 and 6 to determine its slope and intercept. Yet, the fact that point 5 is exactly located on the line joining points 4 and 6 gives a good probability that our assumption is correct.

We have learned how to find the values of the compensation points from the analysis of PS_RL above and will apply the same method below. We calculated the following parameters:Comp(1)^−^: ΔS_p_ = 3.990 ΔH_p_ − 99.03 for T_p_ [60–80 °C];
Comp(2)^+^: ΔS_p_ = 2.624 ΔH_p_ − 66.08 T_p_ [80–105 °C];Comp(3)^−^: ΔS_p_ = 2.857 ΔH_p_ − 81.93 T_p_ [105–125 °C].

The coordinates of the compensation points T_c_ and ΔG_c_, expressed in the ΔG plane, are found from the equation of the EE curves (above) and are compiled in Table 5.

Notice that, in the above description of PS_VA, we named the compensations comp(1)^−^, comp(2)^+^, and comp(3)^−^”, since the positive compensation was added to our initial count of 2 in Figure 41 after we realized that point 5 was not “lost” and that a new compensation should be added to the set of the two negative compensations. Additionally, to match the annotations used for PS_RL, comp(3)^−^ is actually the negative compensation line of comp(2), so its name should be changed to comp(2)^−^. Hence, In our new understanding of the TWD results of PS_VA, let us change the annotations to match the ones used that will reflect the differences between PS_VA and PS_RL appear more clearly:Comp(1)^−^ = (1,2,3,4) and the compensation point for this sub-group is C_1_^−^;Comp(2)^+^ = (4,5,6) and the compensation point for this sub-group is C_2_+;Comp(2)^−^ = (6,7,8) and the compensation point for this sub-group is C_2_^−^.

Figure 58 is a plot of ΔG_c_ vs. T_c_, using the data from Table 5 while considering the new understanding of the compensation points. The first difference observed between Figure 58 for PS_VA and Figure 48 for PS_RL is the reversal in the vertical position of the negative and positive compensation points: the negative compensations C_1_^−^ and C_2_^−^ are below C_2_^+^ for PS_VA, and the opposite is seen for PS_RL. This is a major difference apparently only due to their different thermo-mechanical histories. The comparison of the network of compensations for these two treatments makes it easier to realize that three additional compensation points are missing for PS_VA and would have probably become visible if we had extended the T_p_ range of the TWD Debye relaxations at both ends: polarizing at lower values than point 1 (T_p_ = 60 °C) to make C_1_^+^ visible and to higher values than point 8 (T_p_ = 125 °C) to make C_3_^+^ and C_3_^−^ visible. Using the three existing compensation points (Figure 58) and assuming that the geometrical rules established for PS_RL remain valid for PS_VA, we can construct what the compensation network could look like for PS_VA. This extrapolated network construction is shown in Figure 59. Amazingly, provided our assumptions are verified by new experiments with a broader range of T_p_ values, the network for the slowly cooled sample becomes a simple parallelogram tilted upwardly, with the three positive compensation points aligned (red line on top), the three negative compensation points also aligned (blue line at the bottom), and the parallelism of the lines passing through the positive and negative compensation points. When the three compensation points are aligned, it means that their corresponding compensation lines compensate at a single point: this is a super-compensation of the single Eyring relaxations. This same situation is observed in Figure 45 and Figure 56 for PS_RL. Yet, for PS_RL, the coordinates of the super-compensation points of the negative and positive super-compensations are different for both axes. The fact that the network of compensations is a parallelogram for PS_VA (Figure 58) and only a quadrilateral for PS_RL (Figure 52) is significant and represents a characteristic of the state of non-equilibrium of the amorphous phase that can be correlated to specific thermal–mechanical treatments.

The ΔH_g_ and ΔS_g_ values for the visible transition (T_g_) at the middle of C_2_^+^ and C_2_^−^ can be calculated from the intersection of the compensation lines (4,5,6) and (6,7,8): ΔH_g_ = 68.0 Kcal/m and ΔS_g_ = 112.4 cal/m-K. The value of T_g_ from Equation (13) is determined as 92 °C. The vertex of the pencil of the line is C_o_ (T_co_ = 91.78 °C, ΔG_co_ = 26.95 Kcal/m), and therefore T_co_ = T_g_, ΔG_co_ = ΔG_g_. For the PS_VA sample, with no internal stress, T_g_ is exactly located at the vertex of the network of compensations. This is not true for samples that are not at equilibrium and the analysis presented in this section allows for the characterization of the differences. Figure 60 below illustrates ΔS_p_ vs. T_p_ for PS_VA, which should be compared to Figure 56 for PS_RL.

The only certainty about Figure 60 is that the last three points on the left (points 6, 7, and 8 in Table 1) can possibly be fitted by a hyperbola since they belong to the negative branch of the T_g_ transition of an amorphous phase at equilibrium [5,16]. A hyperbolic fit is always feasible with three points, yet the interest is that, for the x-asymptote, the fit gives the vertical red line corresponding to T_p_ = 373 K, clearly what we expect to find for polystyrene. This is not, however, what we found for T_g_ using Equation (13): we found 92 °C. On the other hand, the two points on the left side of the red line belong to a hyperbola fitting the behavior of a positive branch of a T_g_ transition (the asymptote for that left branch is T_g_ = 373 K). Hence, points 3, 4, 5, 6, and 7 are correctly positioned with respect to a T_g_ transitional behavior for a sample at equilibrium. The other points (1–4) are not on that hyperbolic branch, and there are possibly two reasons for this deviation. First, these points may still not be at equilibrium despite the long cooling time because the cooling sample became a glass ever since it reached 100 °C, resulting in a drastic reduction in the rate to reach equilibrium. The other possibility is that points 1–4 belong to the negative branch of the T_g_(−) transition, which may exist at low T_p_. Figure 61 below, showing ΔG_p_ vs. T_p_ for PS_VA, may help solve this uncertainty when it is compared to Figure 54. The first four points of Figure 54 are shifted vertically to position themselves around the “equilibrium” red straight dashed line in Figure 61 and, therefore, these four points belong to the negative hyperbola of T_g_(−) in Figure 60, corresponding to the compensation point C1^−^ of Figure 59.

Another comparison of PS_RL and PS_VA relates to the super-compensation coordinates. For PS_VA in Figure 59, we saw that the red and blue lines joining the positive and the negative compensation points, respectively, were parallel to each other, separated by a distance (δΔG_c_ = 3.494 Kcal/m and δT_c_ = 31.1 °C (31.276). The equation of the positive super-compensation line is ΔG_c_ = 27.62 + 3.036 10^−3^ T_c_(K), and for the blue super-compensation line, it is ΔG_c_ = 24.12 + 3.036 10^−3^ T_c_(K). The intercept and slope of the super-compensation line provide the coordinates of the super-compensation points “of the lower scale correlation map”: ΔH_c_ = intercept and ΔS_c_ = −slope. Hence, the super-compensation coordinates for the positive and negative compensations of PS_VA can be compared to the ones shown in Figure 45 and Figure 46 for PS_RL, respectively.
PS_VAPS_RLΔH_sc_^+^ = 27.62 Kcal/mΔH_sc_^+^ = 23.0 Kcal/mΔS_sc_^+^ = −3.04 cal/m-KΔS_sc_^+^ = 0 cal/m-K

ΔH_sc_^−^ = 24.12 Kcal/mΔH_sc_^−^ = 17.5 Kcal/mΔS_sc_^−^ = −3.04 cal/m-KΔS_sc_^−^ = −23.0 cal/m-K

One sees that the thermo-mechanical treatment essentially lowers the enthalpy of the super-compensations, for both positive and negative compensations, with the effect being stronger for the negative compensations (6.67 vs. 4.62 Kcal/m). The network of compensations becomes asymmetrical pursuant to the thermo-mechanical treatment. When there was no mechanical treatment, and the sample was cooled very slowly in the mold to induce a near-equilibrium state for the molded sample, the network of compensations appeared symmetrical; yet, the structure of the polarity of the network of compensations was reversed (upside down). Note that the confirmation of these conclusions is pending a repeat of the TWD testing of the slowly cooled samples using a broader span of T_p_ values to validate the construction of the compensations’ network. Our analysis of the PS_VA results was performed with only three compensation points and using the empirical rules established from analyzing samples in non-equilibrium amorphous states. We have only presented the details of our analysis for one treated ample, PS_R, but have presented elsewhere (II.4 of [5]) the influence of several other thermal–mechanical treatments validating the generalization of the results presented in this review: the presence of three scales to consider the interactions, namely the scale of their Debye relaxation, the scale of their compensation, and the scale of the compensation of the compensations, i.e., of their super-compensation, with each scale interactively coupled to the next one forming a general network of compensations.

In summary of this Section 2.3, it should be concluded that the thermal-windowing depolarization (TWD) procedure applied to the rheomolded PS samples (compression-molded under vibration) allows for the quantification of the influence of the thermal–mechanical history on the “internal stress” incurred by the rheomolding variables. In this Section 2.3, we also showed the power of the TWD analysis to gain insights into the interactive coupling relaxation mechanisms between the local molecular “tags”, i.e., the dipoles, activated dielectrically. The relaxation map of these rheomolded samples indicates that the deconvoluted dipole relaxations belong to a highly ordered network of compensation lines themselves, compensating to a single point of the relaxation map. This “super-compensation” behavior appears to indicate a unique simple kinetic origin to the various electrical relaxation mechanisms found below T_g_, at T_g_, and above T_g_. The phenomenon of compensation of the compensation lines of the Arrhenius relaxations generated at various T_p_ values seems to be a key revelator of the fundamental mechanism responsible for this apparently diversified and complex behavior [5,16]. As we said in our objectives of this review, we will attempt to understand, in the Section 3 below, the dual-phase gears capable of generating such apparent simplicity and how the interactions in the amorphous phase are modulated by the presence of compensations at different scales.

## 3. Discussion

### 3.1. Challenging Results for the Conventional Models of Polymer Physics

In many ways, the results obtained by TSD or TWD represent a challenge to the classical views on macromolecules:-The T_g,ρ_ peak (Figure 30 and Figure 31) is in dispute: some scientists claim it is T_LL_ [34], but we believe that T_g,ρ_ and T_LL_ have different origins, and we propose a different dual-phase explanation for both transitions.-The existence of the T_LL_ relaxation transition, which, in the context of TSD/TWD outputs, can be observed by TSD (Figure 36) or can be derived from relaxation maps (Figure 11) as the value of T_p_ that corresponds to ΔS_p_ = 0 in the negative branch of a Z structure, is still a well-known controversial issue because its presence is incomprehensible using the current models of polymer physics [11,12,40,41].-The existence of a negative compensation, systematically found for T_p_ just above the T_g_ peak (Figure 12), is essentially ignored in the literature on the thermally stimulated depolarization current, probably because it remains without interpretation from the classical views of polymer physics: for instance, Lacabanne and collaborators [6], who have extensively contributed to exposing and documenting the benefits of TSD/TWD in thermal analysis, always search for positive compensations in a relaxation map (Figure 10) but never consider the duality positive/negative compensations, which we call the Z structure of the transition at the scale of the relaxation map (Figure 11).-Certain authors [42] have expressed their doubt that the compensations of Debye relaxations obtained at variable T_p_ values could correspond to a real physical process, claiming instead that the compensations were merely linear fitting conveniences. Additionally, the classical explanation of the physical meaning of the compensations of the relaxation modes by the protagonists of the thermal sampling compensations, Lacabanne and collaborators [6], has been challenged by our dual-phase explanation [5,16], as will be developed further in this review.-The network of multi-compensations and super-compensations found for the organization of the enthalpy and entropy of rheomolded specimens (Figure 45, Figure 46, Figure 50, and Figure 57 in Section 2.3) may represent the greatest challenge to any theory of polymer physics, including our own model: why do all the relaxation modes describing interactive coupling in the amorphous phase, above and below Tg, structure into a super-compensation network of positive and/or negative compensations?-In summary, the T_g,ρ_ T_LL_, and multi-compensation network described in Section 2.3 are experimental facts that theories cannot ignore.

The dual-phase model of dissipative interactions is introduced in Section 1.2 and in [7]. Section 1.2 is a simplified quantitative presentation of the terms and assumptions of the dual-split kinetic models (limited to the vertical structuring of the dissipative energy [7] that we will refer to in the following Section 2.2, Section 2.3 and Section 4). As explained in Section 1.2, the dual-conformers interact pursuant to a cross-duality: they belong to a given macromolecular chain that partially controls its intra-chain conformational energy (cis–gauche–trans or (c,g,t)) and are also coupled to the collective set of dual-conformers belonging to other macromolecules via intermolecular interactions. We symbolize this cross-duality of the dual-conformers by [(b/F) ↔ (c, g, t)], where b and F represent two conjugated states of the inter–intra duality, and (c, g, t) refers to three conformational states of the dual-conformers. The distribution function between the different states available is described by the dual-split kinetics (Equations (6)–(8)). The cross-duality of the dual-conformers forces the existence of a local heterogeneity, i.e., the presence of b-grains surrounded by F-conformers. Furthermore, the coupling by compensation between the horizontal and vertical solutions of the minimization of the dissipative function results in the formation of a dissipative elastic wave that propagates through the material (above its T_g_) and delocalizes the local heterogeneity of the density due to the F/b duality. As explained below, these assumptions are “experimentally verifiable” by the presence of the T_g,ρ_ peak observed in TSD (linked to the b-grains surrounded by the F-conformers’ structure) and by the manifestations of the T_LL_ transition in the upper melt (linked to the stability of the dissipative elastic wave that dissipates at T_LL_). In a certain way, the presence of the T_g,ρ_ peak and T_LL_ may be considered an indirect validation of the assumptions of the local duality (b/F) in our model (at T_g,ρ_) and its collective aspect (at T_LL_).

### 3.2. Dual-Phase Understanding of T_g_, T_g,ρ_ and the T_LL_ Manifestations

The characteristics of T_g,ρ_ and T_LL_ were discussed in Section 2.2. The presence of the T_g,ρ_ peak is universal in the characterization of polymers by TSD. We found the peak just above T_g_ for all the polymers tested. The position of T_g,ρ_ with respect to T_g_ depends, to a large extent, on the choice of the polarization temperature T_p_, which enhances the respective magnitude and the resolution of the peaks by the effect of polarization selectivity mentioned in Section 2.2. TSD, which works at a very low-frequency equivalent (∼10^−5^ to 10^−3^ Hz, see II.1 in [5]), is capable of resolving the T_g_/T_g,ρ_ kinetic presence with far higher resolution and sensitivity than other techniques. However, it is not uncommon to observe that the T_g_ and T_g,ρ_ peaks merge into a broad intense peak for certain polarization temperatures, thus complicating their fine analysis. The transition T_LL_ should not be confused with T_g,ρ_, as has many times been the case [21]. Its manifestation by a peak in TSD, located at a higher temperature than T_g,ρ_, is not systematic and depends on the thermo-mechanical treatment of the specimen. In some instances, the mechanical history of the specimen and the temperature of polarization are such that T_g,ρ_ and T_LL_ merge or overlap (Figure 35), which may be the reason for the confusion. TWD normally allows for a good separation of all the peaks, which can be isolated and deconvoluted individually. As mentioned above, the dual-phase model assumptions are responsible for the common dynamic origin to T_g_, T_g,ρ_, and T_LL_, and even to T_β_ [25]. The apparent complexity of the depolarization relaxation behavior above T_g_ arises for two reasons: (1) The kinetics are due to the interactive coupling between horizontal structuring and vertical structuring that combine to create the structure of the free energy. A dissipative wave, active only above T_g_, generates the coherence between the two structuring modes: the result is a split of the peaks observed at the occurrence of the main relaxation, like in Figure 35. Therefore, one could say that T_g,ρ_ is inherently reflecting the duality of the b/F-conformers; the presence of a shoulder on the main TSD or DSC peak (Figure 35, Figure 36 and Figure 37) might just be generated by kinetic effects. However, even if the T_g_ and T_g,ρ_ are kinetically related, their dielectric origin is different most of the time, because the effect of the voltage field on the b and F-conformers is different: The T_g_ peak occurs due to the relaxation of dipole moments attached to the b-conformers, and the kinetic complexities might reveal some modulations on the peak itself. The dipolar moment for T_g,ρ_ is probably associated with ionic dipoles, or perhaps the space charges attached to the interfacial tissue created by the F-conformers. There is plenty of evidence (II.2 in [5]) that suggests that the T_g,ρ_ peak is related to the free volume in the sample, which the dual-phase interpretation redefines as the volume around the F-conformers. Either local unstable ionic dipoles or space charges get trapped in the free volume characteristic of the surroundings of the F-conformers. These charges are injected by the voltage field, or enhanced by it, and relax at T_g,ρ_. In some instances, the T_g,ρ_ peak is broad and contains a combined effect of molecular dipoles and free volume relaxations (Figure 2.10 of II. 2 in [5]). In such cases, two or more additional compensations are observed in the (T > T_g_) region up to T_LL_. The existence of this special F-conformer tissue in the structure is also revealed by the response of polymers during the Thermally Stimulated Polarization Current (TSPC) experiments, either upon heating or cooling; this type of test is not covered in this review but elsewhere (II.1 of [5]): during a TSPC run, the T_g,ρ_ peak is not observed, but volume or surface conductivity starts at that same onset temperature. According to studies on the dual-phase viewpoint [10,17], the local density difference between the b-grains and the F-conformers is “time-averaged” by the constant sweeping (above T_g_) of a “dissipative elastic wave” with a frequency ω_o_ that is a function of temperature and molecular weight and thus is different from the Brownian dissipation, i.e., the thermal fluctuation characteristic of the Boltzmann mean field (the classical kT/h term). As a consequence of the dissipative elastic wave in the rubbery and melt states, the F-conformers become the source of formation of a tree-type channel of nano-conduction for the injected charges from the electrodes across the non-conductor. At about 20 °C below T_g_, the F-conformers are frozen, which makes the boundaries around the b-grains become localizable (and the source of crazes when submitted to axial stresses). T_LL_ is the end of the existence of the dissipative elastic wave, which turns into a Brownian dissipation. It is the end of the network of compensation. T_g,ρ_ is revealed or reveals the existence of the nano-conductive charges partially filling the F-conformers in the spatial area around the b-grains.

The value of T_LL_ can be calculated (I.2 of [7]) from the negative compensation in a relaxation map, by extrapolating the entropy vs. T_p_ results for ΔS_p_ = 0 (or toward a minimum asymptotic value). In this sense, we can consider T_LL_ as the temperature marking the end of the relaxation behavior due to the modulation by the dissipative term in the dual-phase dynamics. This is seen in Figure 62 below when the rate of dN_b_(t)/dt becomes 0, ending the modulation of n_tb_(t) by N_b_ (t). This figure is understood from the explanations given in Section 1.2 regarding Figure 14, Figure 15 and Figure 16 related to vertical Dual-Split kinetics.

In split structuring, N_b_(t) is the sum of the population n_tb_ and n_cgb_, and Figure 62 plots the derivative of N_b_(t) and n_tb_(t) to show the modulation that starts upon cooling (q = −1 K/s) at T_LL_. It is clear that T_LL_ is rate-dependent for either heating or cooling ramps. We propose that the mechanism of relaxation in polymers is due to the dynamic coupling of two types of splitting processes of the total statistical population of conformers in interactions: the creation of N_s_(t) energetic kinetic systems (horizontal splitting) and the modulation of the conformational structure of these systems by the dissipative function (vertical splitting). This applies to the coupling of cooperatively interactive dipolar motions and the modulation of both the sub-T_g_ and (T > T_g_) kinetics. It might be more appropriate to categorize T_LL_ as one of the kinetic manifestations resulting from the cooperative kinetic process already giving rise to T_β_, T_g_, and T_g,ρ_. Beyond T_p_ = T_LL_, the organization of the inter–intramolecular interactions between the various dipoles as a dissipative network is kinetically inefficient and hence has ended. As we said earlier, a description of the properties of the polymer by invoking the properties of the individual macromolecules embedded in a mean field is acceptable from this point on. T_β_ is the temperature for the onset of energetic kinetic coupling. It should be a function of thermal history. The temperature of infinite viscosity defined by the WLF equation, fitted to the T_g,ρ_ Arrhenius transform, should characterize the lower limit for T_β_. The intercept of the (T > T_g_) Z lines, either from multi-compensations or for specimens with variable thermal history, should also define T_βo_, the lowest T_β_ achievable.

### 3.3. Compensations, Multi-Compensations, and Super-Compensations in Dual-Phase Systems’ Simulation

In the analysis of the TWD depolarization results, we make use of compensations as a quantitative characterization tool to evaluate the interactive coupling between the Debye relaxation modes. The compensation of the Debye characteristics (ΔH, Ln τ_o_) means that their modes of relaxation, stimulated at various temperatures (T_p_), are not independent but coupled. The interactive coupling is clearly visible graphically: a series of plots of log τ(T) vs. 1/T in the Arrhenius plane, or of ΔG(T) vs. T in the ΔG plane [16], when T_p_ varies, displays a set of spectral lines that converge to a single point, the compensation point. Hence, the compensation point coordinates appear to define the interactive coupling between the relaxation modes. When the Debye characteristics of the relaxation modes are their thermo-kinetics variables, ΔS_p_ and ΔH_p_, their interactive coupling is characterized by the linearity between ΔS_p_ and ΔH_p_ when T_p_ varies. This line is the compensation line.

Most of the scientists working with TSD/TWD apply the concept of compensation to the analysis of complex TWD results when the amorphous phase is made up of cooperative relaxation modes of different origins, namely semi-crystalline polymers, copolymers, and blends of polymers, as well as when additives are added. In every such case, the compensation characteristics of the amorphous state(s) are modified, and this modification can be quantified by comparing its compensation point coordinates with a reference value. This is why the TWD characterization technique offers a unique value in the spectrum of thermal analysis instruments available to determine the properties of non-conductive materials.

Yet, beyond the practical usefulness of the use of compensations to characterize the amorphous state of non-conductive matter, the theoretical value of compensation laws is a full debate on its own. We refer to the appropriate literature for more details [42,43,44,45]. In this section, we examine the possible analogy between the compensation results obtained by TWD when T_p_ varies and the compensations observed in dual-phase systems when the variable Δ_e_ is varied in Equations (6)–(8).

The fundamental constants of the dual-split kinetic model (Equations (6)–(8) of Section 1.2) are B_o_, Δ_m_, υ_m_, and Δ_e_. When these constants are known, we can characterize the dynamics of this system of B_o_ dual-conformers by their statistical population: their cis-, gauche-, and trans-spatial conformation, and their b/F dissipative state, which controls the free volume in the system. The interactive coupling between the conformational states and the b/F state is described by Δ_x_, υ_x_, which are functions of Δ_m_, υ_m_, and Δ_e_. At equilibrium, the system of interactions is stable, and the b and F populations are “transparent”, i.e., undistinguishable, statistically speaking. As soon as non-equilibrium conditions occur, such as by cooling or heating, the b/F population becomes asymmetric, which is governed by dissipative kinetics, the dual-split kinetics, or EKNETICS as we have called it. The EKNETICS generates a dissipative term that can be quantitatively expressed by a kinetic-like constant, k_x_, which varies only with T, for a given set of Δ_m_, υ_m_, and Δ_e_ (Figure 27 of Section 1.2). As explained in Section 1.2, for a given value of Δ_e_, we run simulated annealing experiments after quenching from the same temperature down to a series of five lower temperatures at which the value of (k_x_) is determined from a phase plot of dN_b_/dt vs. N_b_ at t → 0. Figure 28 shows that the temperature dependence of k_x_ follows an Arrhenius variation, which corresponds to the linearity of Ln(k_x_(T)) with 1/T. From the intercept and slope of such an Arrhenius plot, we determine Ln υ_x_ and Δ_x_, respectively, which we could hypothetically associate with the entropy and enthalpy of activation for the (b/F ↔ (c, g, t)) interactive coupling kinetics, the way we did to convert the current of depolarization into a relaxation time and determine Ln(τ_o_) and ΔH from the Arrhenius temperature dependence of τ(T). We repeat the simulations for a series of Δ_e_ values at constant Δ_m_ and υ_m_. This procedure establishes the value of Ln υ_x_ and Δ_x_ for various Δ_e_ for a given set of Δ_m_ and υ_m_. Finally, we change the values of Δ_m_ and υ_m_ and repeat the entire procedure (note that in the TWD experiments, the variables are τ_o_ and ΔH, i.e., a relaxation time and enthalpy, whereas in the simulation, υ_x_ and Δ_x_ refer to a frequency and enthalpy. This difference results in a change in sign for the log axis variable, since 1/υ_x_ = τ_x_). Figure 63 displays the influence of Δ_e_ on the k_x_ statistics: for a given set of Δ_m_ and υ_m_ (Δ_m_ = 9250 and υ_m_ = 10^11^ in Figure 63), a compensation law is observed between Ln υ_x_ and Δ_x_ when Δ_e_ varies. The compensation between Ln υ_x_ and Δ_x_ when Δ_e_ varies is similar to the compensation between -log τ_o_ and ΔH when T_p_ varies in a TWD experiment (the compensation line in a compensation search). The compensation means that the Arrhenius lines of Ln(k_x_) vs. 1/T converge to a single point, the compensation point, when Δ_e_ varies. This is similar to the situation observed in a relaxation map showing the compensation of the spectral Arrhenius lines when T_p_ varies. As we said earlier, and is also explained in Section I.2.5 of [5] or in [16], the coordinates of the compensation point of Ln(k_x_) vs. 1/T can be determined by the slope and intercept of the corresponding compensation line in a compensation search, in our case, Ln υ_x_ vs. Δ_x_ obtained by regression.

Figure 64 shows the influence of changing the value of Δ_m_ and υ_m_ on the compensation line for various Δ_e_. Note that the variables in Figure 64 are “normalized”, i.e., become (Δ_x_ − Δ_m_) and (Ln(υ_x_) − Ln(υ_m_)). When Δ_e_ varies, for a given pair of (Δ_m_, υ_m_), this is Δ_x_ and υ_x_, which vary. This will be important to remember when we simulate the behavior of (ΔH, Ln(τ_o_)) with T_p_ using the behavior of (Δ_x_, Ln(1/υ_x_)) with Δ_e_. In thi figure, it should also be noted that the Δ_e_ values are rather small (ranging from 5 to 150) with respect to Δ_m_ and that Δ_e_ decreases from left to right on the compensation lines that all pass through the origin. The origin corresponds to (Δ_x_ − Δ_m_) = 0 and (Ln(υ_x_) − Ln(υ_m_)) = 0, obtained for Δ_e_ = 0. All of these particular features are important to note to be able to correlate the effect of T_p_ on the Ln(τ) vs. 1/T kinetics with the effect of Δ_e_ on the Ln(k_x_) vs. 1/T kinetics. In particular, we need to know if the dual-split model of the interactions can simulate the switch from a positive compensation law to a negative compensation law when T_p_ crosses the T_g_ of the polymer or if this results from another cause. This necessitates understanding how Δ_e_ and T_p_ are related.

It is clear in Figure 64 that changing the values of Δ_m_ and/or υ_m_ in this low-value range of Δ_e_ modifies the slope only. The inverse of the slope gives the value of T_c_, the temperature of compensation for the Ln(k_x_) vs. 1/T lines when Δ_e_ varies (since we equate the gas constant to 1 in the simulations, we need to divide (1/slope) by 1.987 cal/mole if ΔH_x_ is expressed in such units). In Figure 64, one sees that T_c_ for Δ_m_ = 9250 (square symbol) is greater than for Δ_m_ = 9500 (the “+” symbol), at a constant υ_m_ (10^11^). The same slope can be obtained if we change Δ_m_ and υ_m_ simultaneously. Compare the dots (8750, 10^12^) and the “+” symbols (9500, 10^11^). Figure 65, Figure 66 and Figure 67 show the effect of increasing Δ_e_ for these three sets of Δ_m_ and υ_m_ parameters. Figure 65 applies to Δ_m_ = 9500, υ_m_ = 10^11^. Δ_e_ is now extended to 400 (4.21% of Δ_m_). We observe two compensation lines, one passing through the origin, point O, at low values of Δ_e_, corresponding to the “+” symbols in Figure 64, and another straight line covering the greater values of Δ_e_, ending at point G in the figure. We have written the equation of the compensation lines on the graph y = α_o_ + α_1_ x for the larger values of Δ_e_ and y = α′_1_ x for the low Δ_e_ values. We can express the coordinates of the compensation points of [Ln(k_x_), T_c_] for these two compensation lines:
Figure 65Normalized compensation search of Ln(υ_x_/υ_m_) vs. (Δ_x_ − Δ_m_) for Δ_m_ = 9500, υ_m_ = 10^11^. Effect of varying Δ_e_. Here, Δ_e_ decreases from Δ_e_ = 400, at the left, to Δ_e_ = 5, at the far right. Notice that Δ_e_ =0 corresponds to Δ_x_ = Δ_m_ and υ_x_ = υ_m_. Reproduced with permission from [4], SLP Press, 1993.
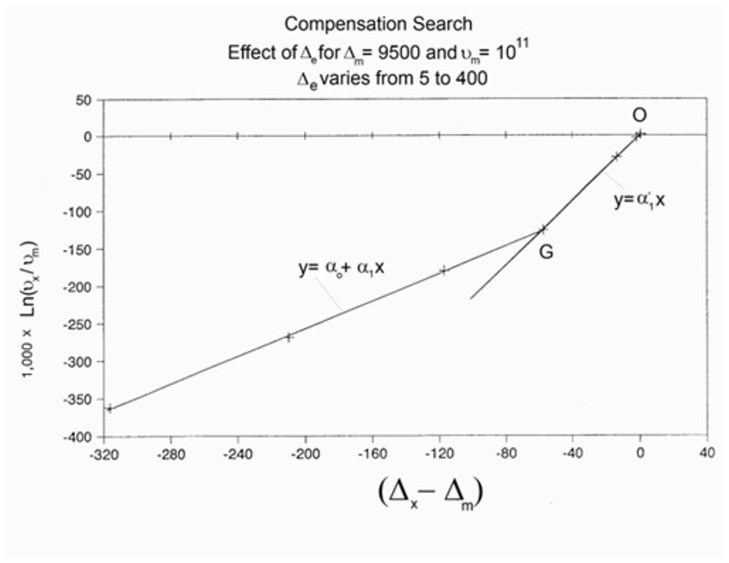

T_c1_ = (1/1.987α_1_)(14)Ln(k_x_)_c1_ = α_o_ + Ln(υ_m_) − α_1_Δ_m_
with Δ_e_ > Δ_eg_, where Δ_eg_ is the value of Δ_e_ at point G.T_c2_ = (1/1.987α′_1_)(15)Ln(k_x_)c_2_ = Ln(υ_m_) − α′_1_ Δ_m_
with Δ_e_ < Δ_eg_.

Could the two compensation lines of Figure 65 simulate the change in behavior at T_g_ for an amorphous phase analyzed by TWD? Let us consider the experimental evidence first: for a Debye relaxation map, scanning T_p_ from a low value below T_g_, crossing T_g_, and above T_g_, we go from a positive compensation to a negative compensation, meaning that the value of ΔH_p_, the slope of the Arrhenius line, increases first with T_p_, passes through a maximum value at T_g_, and then decreases toward a minimum value. This maximum of ΔH_p_ describes the typical response of an amorphous phase submitted to a voltage field stimulation according to the thermal-windowing protocol (Figure 3). Now, assuming that we can simulate the effect of T_p_ on the depolarization kinetics by a change in Δ_e_ on the k_x_ relaxation kinetics, we can formulate and test the hypothesis that the compensation line below point G in Figure 65, corresponding to the higher values of Δ_e_, simulates the T_p_ < T_g_ positive compensation and that the compensation line above point G, corresponding to lower and vanishing values of Δ_e_, simulates the negative compensation. This would only occur ifT_c2_ < T_c1_(16a)
andLn(k_x_)c_2_ > Ln(k_x_)c_1_(16b)

The first requirement is met since α′_1_ > α_1_ in Figure 65, which, plugged into Equations (14) and (15), means that (16a) is true. The second requirement, (16b), will be met if and only if the following applies:Ln(k_x_)c_2_ = {Ln(υ_m_) − α′_1_ Δ_m_} > Ln(k_x_)c_1_ = {α_o_ + Ln(υ_m_) − α_1_Δ_m_}(17)
where the expressions of Ln(k_x_)c_1_ and Ln(k_x_)c_2_ are brought in from (14) and (15), respectively. Equation (17) simplifies to the following:Ln(υ_m_/υ_xg_) + α_1_ (Δ_xg_ − Δ_m_) < (α′_1_ − α_1_)Δ_m_(18)
where υ_xg_ and Δ_xg_ are the values of υ_x_ and Δ_x_ at point G, at which Δ_e_ = Δ_eg_. Hence, we must demonstrate thatLn(υ_m_/υ_xg_) < α′_1_ Δ_m_ − α_1_Δ_xg_(19)

However, at point G, we also haveLn(υ_xg_/υ_m_) = α′_1_ (Δ_xg_ − Δ_m_) (20)
which is the transposition of y = α′_1_ x.

Finally, the second requirement in Equation (16b) resumes to test if Equation (21) is true:α′_1_ (Δ_m_ − Δ_xg_) < α′_1_ Δ_m_ − α_1_Δ_xg_(21)

Equation (21) simplifies to the following:−α′_1_ Δ_xg_ < −α_1_Δ_xg_(21a)
orα′_1_ > α_1_(21b)
which is always true when T_c2_ < T_c1_.

In summary, the two conditions (16a) and (16b) are verified, and we can conclude that Figure 65 is the correct representation of the positive and negative compensations observed for describing interactive coupling in the amorphous phase of a homopolymer across its T_g_. This analogy implies that Δ_e_ decreases as T_p_ increases, which is observed in Figure 65 since Δ_e_ decreases from left to right.

However, there is one remaining question to consider: Even if this solution seems to work, is it the only plausible solution? In other words, is the solution offered by Figure 65 the only possible explanation for the switchover at T_g_ from a positive to a negative compensation? Here are a few good reasons to remain cautious:

The simulation in Figure 65 is not exactly a simulation of the TWD experiment (we will return to this point). In particular, the depolarization stage in a TWD experiment is obtained under a thermal temperature ramp at a constant rate. This is not the case for the dual-split model simulations which are performed under isothermal conditions.If Equations (16)–(21) were truly simulating the TWD two-compensation split at T_g_, point G in Figure 65 would correspond to the T_g_ of the polymer system of interactions. At this point, Δ_e_ = Δ_eg_, which can be obtained from the simulation data, and the values of Δ_x_ and Ln υ_x_ are those at point G. The crossing of the positive and negative compensations in an amorphous polymer occurs at T_p_ = T_g_, defining the characteristics of the Z line, the line joining the two compensation points. However, the existence and the position of the break in Figure 65 are determined by the choice of the values of Δ_m_ and υ_m_. As we will see, for certain pairs of Δ_m_ and υ_m_, the break observed at point G in Figure 65 is not observed (see Figure 67 discussed later). This could, of course, simply mean that the choice of Δ_m_ and υ_m_ to simulate polymers should be restricted to those values that display a break conform to Figure 65 in their compensation search when Δ_e_ varies. The fact that the choice of Δ_m_ and υ_m_ is not unlimited but rather should correspond to some criteria to meet the requirements that they simulate polymers is, indeed, a reasonable proposition that we believe is true, regardless of the issue considered here.However, the fact that we need to assign a certain value of Δ_e_ equal to Δ_eg_, the value of Δ_e_ when T_p_ = T_g_, faces various objections. The first objection is that T_g_ is a kinetic phenomenon observed whether a voltage field is applied or not. As indicated in Section 1.2, we attribute the freezing of the statistical populations at low temperatures to kinetic reasons. This occurs for all values of Δ_e_, not just a specific one. The second objection is that we attribute the break in Figure 65 to the interference on the kinetics of k_x_ of the coupling between Δ_e_ and (Δ_m_, υ_m_): this is what we mean by interactive coupling between the conformational states and free volume (the b/F state). The coupling is more prominent at large values of Δ_e_, i.e., for Δ_e_ > Δ_eg_ in Figure 65, and simplifies to a pure influence of Δ_e_ alone at lower values of Δ_e_, vanishing to 0. This has nothing to do with the T_g_ transition that is observed for all values of Δ_e_. Moreover, in the simulations, the value of Δ_e_ is constant during cooling or heating; it is not a function of T. Even if, in our explanation of the compensation of ΔH and ln τ_o_, it has been said that the value of T_p_ during the polarization stage fixes the value of Δ_e_, this value remains constant during the simulation of the cooling and depolarization stages. Obviously, it is possible to change this assumption in the simulations and make Δ_e_ a function of T, but this was not the case in the simulations that resulted in Figure 65.

Our conclusion is that despite the remarkable and simple explanation provided by the break in Figure 65, we need to find another explanation for the sharp change in the compensation slope and intercept at T_g_ that results in a positive followed by negative compensation across T_g_. We have found such an explanation, and it is not based on the change in slope in Figure 65; rather, it has to do with “grid-shifting”, which is not detailed in this review but in I.3.4.5 in [5]. Grid-shifting essentially indicates that there are three rheological temperature “scaling ranges” for the internal mobility in amorphous polymers [11,12] and that the transfer function of Equation (22) below is affected when the polarization temperature, T_p_, is located in either one of these three ranges. For T_g_ < T_p_ < T_g_ + 25, the kinetics of molecular motion induces the elastic dissipative wave that starts to un-freeze, move around, and delocalize the (b/F) pockets of free volume around the b-grains, which remain frozen for T_p_ below T_g_. This reorganization of the pockets of free volume is responsible for a change in the polarization nature in this temperature range above T_g_, changing from electronic and atomic polarization to orientation polarization: this modifies the polarization efficiency, i.e., the ability to modify Δ_e_ at T_p_. As we will see with respect to Figure 67, Figure 68, Figure 69 and Figure 70, a large initial value of Δ_e_ (E, T_p_), relative to its equilibrium value, may be present in the sample at T_p_, during the polarizing stage, as frozen-in internal stress was induced by mechanical treatment during the processing of the TWD sample, and such an initial Δ_e_ value present at T_p_, could drastically change the depolarization behavior such as the slope and intercept of the compensation in Figure 66.

In the equations of the dual-split model (EKNETICS), the dynamics of the interactive coupling between the dual-conformers is characterized by the dissipative thermo-kinetic values, Δ_x_ and ln υ_x_, which, as we saw, depends on the value of Δ_e_. If we want to apply this model to the dynamics of the polarization and depolarization of the dipoles, that is, the activated or deactivated dual-conformers, we need to determine or assume the coupling equation between the voltage field, E; the polarization temperature, T_p_; and Δ_e_: Δ_e_(T_p_, E). Equation (22) below is formulated from considerations of both the dual-phase model and the thermo-kinetics of the dipoles by TSD/TWD:

Point O in Figure 65 can be associated with a temperature T_x_ of the polymer system of interactions for which Δ_e_ = 0. We assume that T_x_ is the thermodynamic value of T_LL_. In other words, T_x_ is the T_p_ value corresponding to Δ_e_ = 0. We still need to understand the physical meaning of T_x_. At the origin O, x = y = 0, and thus υ_x_ = υ_m_ and Δ_x_ = Δ_m_. Therefore, this state does not generate asymmetric b/F statistics, which is our definition of T_LL_. We view T_x_ as the thermodynamic (rate-independent) transposition of T_LL_ that can only be experimentally measured by extrapolation of T_LL_ at the zero rate. Regarding the T_LL_ transition, we previously stated that it was the temperature of “superposition” of the b and F states; borrowing an expression from quantum mechanics, we called it “the temperature at which the b and F states become ‘transparent’ or ‘undistinguishable’”. Yet, in the dual-split statistics, a system of interactions at equilibrium is defined by three fundamental constants, namely Δ_m_, υ_m_, and Δ_eo_, not simply Δ_m_ and υ_m_. We, therefore, suggest that T_LL_ is the temperature at which Δ_e_ (T_p_) = Δ_eo_, which provides a dynamic character to the “superposition” transition. By combining the statements above regarding the value of Δ_e_ at T_g_, T_LL_, and T_x_, we can tentatively conclude that Δ_e_ varies with T_p_ and the voltage field E as follows:Δ_e_ (E, T_p_) = Δ_eo_ + kF(α, μ,N) E (T_LL_ − T_p_) T_LL_(22)
where E is the normalized voltage field (V/m); F is a transfer function affecting the efficiency of the modification of Δ_e_; α, μ, and N are the polarization Debye parameters expressed in a modified generalization of the Clausius–Mosetti formulation of the dielectric constant (Equation 3.20 p. 245 of [5]); and k is a scaling constant.At T_p_ = T_LL_; Δ_e_ = Δ_eo_(23)Δ_e_ = 0 for T_p_ = T_x_

Here, T_x_ is the thermodynamic value of T_LL._

Δ_eo_ is also the value of Δ_e_ when the material is not dielectrically stimulated (E = 0). Note that F(α, μ,N) in Equation (22) is a transfer function between the macroscopic and the molecular world, which, for the present purpose, does not need to be discussed but can be derived from the expression of the dielectric constant, ε, as a function of the total dipole moment, permanent and induced, and the number of dipoles, N (Eq. 3.20, p. 245 in [7]). Equation (22) can be rewritten as a function of Δ_eg_:Δ_e_ (E, T_p_) = Δ_eo_ + (Δ_eg_ − Δ_eo_) (T_LL_ − T_p_) (T_LL_ − T_g_) (24)
with:Δ_eg_ = Δ_e_ (E, Tg) = Δ_eo_ + kF(α, μ,N) E (T_LL_ − T_g_) T_LL_(25)

Equation (22) is assumed to couple the electrical field effect and the temperature effect, using the framework of the dual-phase concept to define interactions in dielectric materials: in other words, it is assumed that these two effects are not separable but coupled; furthermore, since T_LL_ and T_g_ are rate-dependent and thus subject to thermal history, Δ_e_(E,T_p_) is also a function of time for samples out of equilibrium and a function of the annealing parameters. The true simulation of the TWD polarization and depolarization stages using the Bucci equations has been covered elsewhere (Equations (1.21)–(1.24) of I.1 in [5]) but is not what is simulated in Section 1.2 and in Figure 61, Figure 62, Figure 63, Figure 64, Figure 65, Figure 66, Figure 67, Figure 68, Figure 69 and Figure 70, which report the effect of Δ_e_ on Δ_x_ and υ_x_ for an isothermal relaxation process taking place to determine Δ_x_ and Ln(υ_x_) at various Δ_e_ values. In other words, the TWD relaxation protocol is different from the dual-conformers relaxation protocol.

When Δ_m_ and υ_m_ take another set of constant values, the aspect of the normalized compensation search shown in Figure 65 presents notable changes, as evidenced by Figure 66 and Figure 67. Figure 66 applies to the system comprising Δ_m_ = 8750 and υ_m_ = 10^12^ and Figure 67 to the system comprising Δ_m_ = 9250 and υ_m_ = 10^11^, already analyzed in Figure 63 but now presented with an extended range of Δ_e_ values (from 5 to 900).

**Figure 66 polymers-17-00239-f066:**
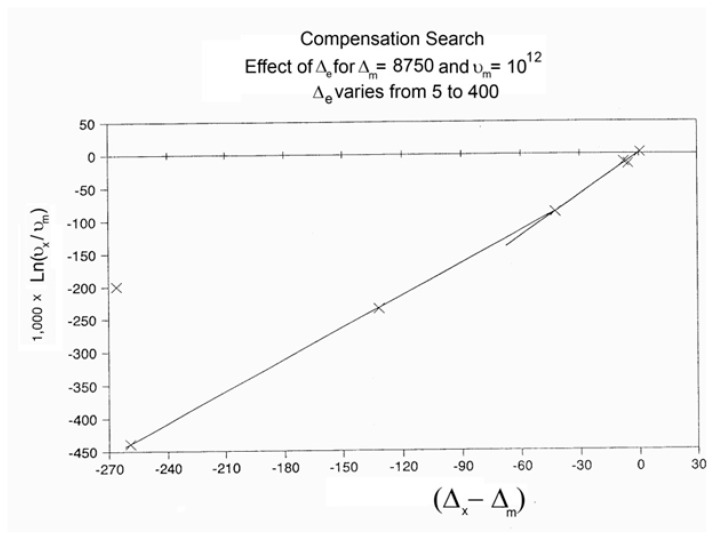
Influence of Δ_e_ on Δ_x_ and Ln υ_x_ using the normalized variables. This is the same graph as in Figure 65, except that Δ_m_ = 8,750 and υ_m_ = 10^12^. The point (x) for (Δ_x_ − Δ_m_) = −268 appears to be off the line. See text. Reproduced with permission from [4], SLP Press, 1993.

**Figure 67 polymers-17-00239-f067:**
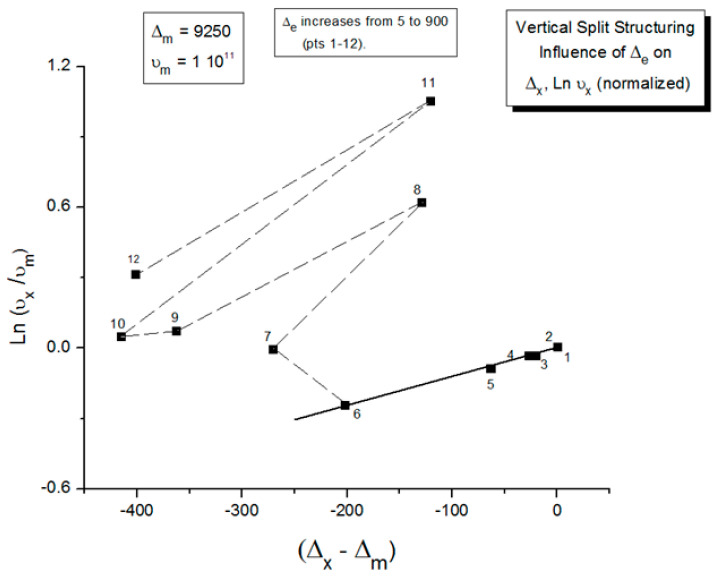
Similar to Figure 65, except that Δ_m_ = 9250 and Δ_e_ ranges from 900 to 5 as the index of the points continuously decreases from point #12 to point #1 in Table 6. What may appear chaotic can be considered a network of interlaced positive and negative compensation lines (Figure 68) for Ln(υ_x_) vs. Δ_x_. Reproduced with permission from [4], SLP Press, 1993.

Figure 66 provides the same information as Figure 65: the existence of two compensation lines defined by points G and O and the slopes α_1_ and α′_1_ such that α′_1_ > α_1_. Again, the simulation results for a different set of Δ_m_ and υ_m_ values correspond to positive and negative compensations for Ln(k_x_) vs. 1/T. The differences bear on the relative positions of the compensation lines, their intercept defining Δ_eg_, and the relative magnitude of T_c1_ and T_c2_. The other observable difference is the last point on the left corresponding to Δ_e_ = 400 and (Δ_x_ − Δ_m_) = −260, on the higher Δ_e_ side of the y = α_o_ + α_1_ x compensation line. This point is totally off the compensation line, as if it was due to an error. However, the simulation for this particular Δ_e_ is entirely similar to the other simulations at lower values of Δ_e_, with the same regression accuracy in the determination of the values of k_x_(T) to find the corresponding Δ_x_, Ln υ_x_. Thus, this point is not due to a simulation error; it is real. We further analyze this apparently erratic phenomenon in Figure 67. As we said, this figure explores the same Δ_m_, υ_m_ settings as in Figure 63 but with a much broader range of Δ_e_ values: 12 values ranging from Δ_e_ = 5 to Δ_e_ = 900. Table 6 provides the results of the simulation.

**Table 6 polymers-17-00239-t006:** Results of a simulation by Equations (6)–(8) of Section 1.2 with Δ_m_ = 9250, υ_m_ = 10^11^, and Δ_e_ constant varying between 5 and 900. Δ_x_ and Ln(υ_x_) are the dissipative dual-phase kinetic parameters corresponding to Ln(k_x_) vs. 1/T, where k_x_ is determined from N_b_(t).

	Δ_e_	Δ_x_	Ln(υ_x_)	Δ_x_ − Δ_m_	Ln(υ_x_) − Ln(υ_m_)
1	5	9250.75	25.33095	0.75	0.00251
2	25	9250.5	25.33285	0.5	0.00442
3	75	9230.38	25.29485	−19.62	−0.03359
4	100	9223.42	25.29295	−26.58	−0.03548
5	150	9187.16	25.23795	−62.84	−0.09048
6	300	9048.21	25.08195	−201.79	−0.24649
7	500	8979.68	25.32185	−270.32	−0.00658
8	600	9121.68	25.94725	−128.32	061882
9	650	8887.69	25.40105	−362.31	0.07262
10	700	8835.45	25.37655	−414.55	0.04812
11	800	9129.59	26.38035	−120.41	1.05191
12	900	8849.11	25.64155	−400.89	0.31312

Figure 67 is a normalized compensation search apparently showing one compensation line only: for points 1–6 (the other points are connected by dashed lines). The points are tagged by their row number listed in the first column of Table 6. They are sorted by increasing Δ_e_ values. The overall view of the effect of an increase in Δ_e_ for these Δ_m_, υ_m_ settings is that, beyond a certain value, say corresponding to the end of the compensation line (1–6), the dissipative system of Δ_x_, Ln(υ_x_) values becomes chaotic in appearance. We drew dashed lines linking the successive points, following a continuous increase in Δ_e_, and we observed the typical response of scattered disorder. For this particular set of values for Δ_m_ and υ_m,_ and for Δ_e_ between 0 and say 400, we observed a compensation line (1–6) susceptible to simulate a positive TWD compensation line in a compensation search of −Ln τ_o_ vs. ΔH. Beyond that value of Δ_e_, which is a function of the choice of Δ_m_ and υ_m_, in Figure 68, we assumed a certain order of the chaos by drawing a series of alternative positive and negative compensations, identified by the direction of the arrow connecting adjacent points. For instance, in Figure 68, the first positive compensation line (1–6) would not end at point 6 but at the intersection of line (1–6) and line (7–8), with the direction of the arrow going down for (1–6) and up for (7–8). Figure 68 uses the same data as in Figure 67 (Table 6), replotted as Ln υ_x_ vs. Δ_x_ (without the normalization of the axes by υ_m_, Δ_m_) to show the analogy with the compensation search of −Ln τ_o_ vs. ΔH (or ΔS_p_ vs. ΔH_p_ in the Eyring plane. In Figure 68, we have drawn three black lines with an arrow pointing down and two red lines with arrows pointing up. The arrows follow the tendency for the y variable, Δ_x_, to decrease or increase within a range. The ranges are determined arbitrarily since the lines only join two points (with the exception of the first range which has six points). It would, indeed, be quite useful, in a complementary research project, to increase the number of Δ_e_ values to resolve the definition of the ranges in Figure 68 (for instance, Δ_e_ = 472.5, 550, 625, 675, 733, 766, 833, and 866). However, the main idea in this section of this review is to suggest that there is an analogy between the compensation of the −Ln(τ_o_) vs. 1/T Arrhenius lines seen for the TWD depolarization behavior when T_p_ varies and the Ln(k_x_) vs. 1/T relaxation results of the EKNETICS when Δ_e_ varies at Δ_m_ and υ_m_ constant. The analogy associates the roles played by T_p_ and Δ_e_ in their respective relaxation processes, which may have consequences on our understanding of what is actually deconvoluting in a thermal-windowing procedure (see Section 3.4).

**Figure 68 polymers-17-00239-f068:**
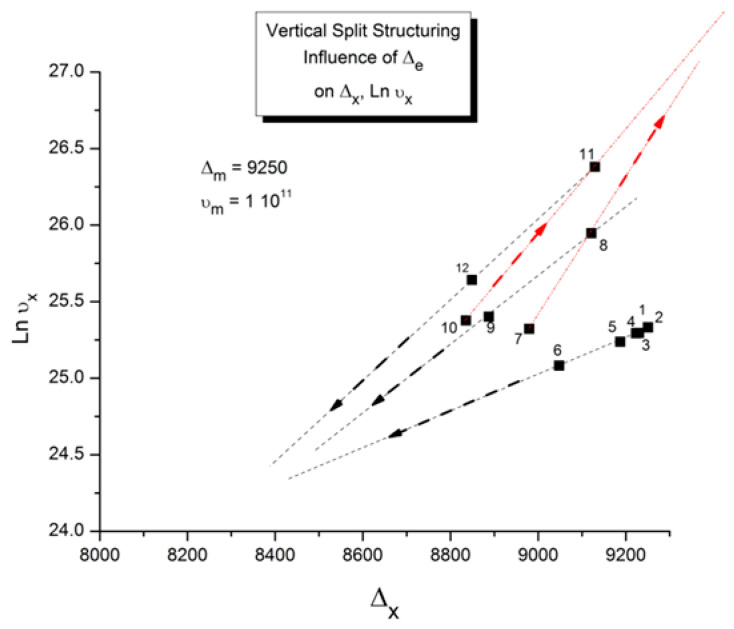
Same compensation search as in Figure 67 but without the normalization of the axes by Δ_m_, υ_m._ This presentation of the results makes the analogy of the compensation at Δ_e_ variable in Figure 68 with that of −Ln τ_o_ vs. ΔH at T_p_ variable in depolarization compensation searches more apparent.

The apparent multi-compensation structure assumed in Figure 68 to explain the chaos observed for large values of Δ_e_ in Figure 67 raises another analogy with a TWD behavior seen for samples put out of equilibrium by “strong” processing conditions (rheomolding treatments) before testing them by TWD. For instance, in Figure 68, as Δ_e_ decreases continuously from a high value, the “positive” lines (11–12, 8–9, and 1–6,) appear to compensate into a super-compensation, while the interlaced “negative” lines (10–11 and 3–8) compensate into another super-compensation; the analogy is with Figure 45 and Figure 46, respectively, for rheomolded sample PS_RL.

Figure 69 and Figure 70 present another angle to reveal the possible structure of Δ_x_ and Ln υ_x_ at higher values of Δ_e_. These figures plot the variation in the normalized data, (Ln υ_x_/υ_m_) and (Δ_x_ − Δ_m_), against Δ_e_, shown in Figure 69 and Figure 70, respectively. The first observation is that Δ_x_ or Ln(υ_x_) decreases when Δ_e_ increases in range 1 [points 1–6]; this correlation is the same for the other positive compensations but is the opposite for the negative compensations: (Δ_x_ or Ln(υ_x_) increases when Δ_e_ increases. A second observation is that in range 1, a quadratic equation can be used to fit the Δ_e_ dependence of either Ln υ_x_ or Δ_x_. This allows us to evaluate the value of Δ_e_ that ends the first range or starts the second range: we found Δ_e_ = 445 with Δ_x_= 8835 and Ln υ_x_ = 24.6. The third observation is that there seems to be a periodic oscillation of the ups and downs for adjacent ranges, with the down values possibly aligned on the straight horizontal line in the case of (Δ_x_ – Δ_m_) in Figure 70. These observations are possibly analogous to what was expressed in Section 2.3, Figure 41, Figure 45, and Figure 46, for instance, regarding the relaxation map for a classical amorphous polymer, polystyrene, which could be fragmented into separate groups, each with a characteristic compensation point; the number and sign of these multi-compensations depended on the thermo-mechanical history on cooling. The network of compensations (Figure 50, Figure 51 and Figure 52) revealed a simple relationship between the various compensation lines in the compensation search plane, from 50 °C below to approximately 60 °C above T_g_. We concluded that the relaxation process described by the individual relaxation modes, i.e., occurring below T_g_, at T_g_, and above T_g_, was correlated as a structure of compensations and super-compensations.

**Figure 69 polymers-17-00239-f069:**
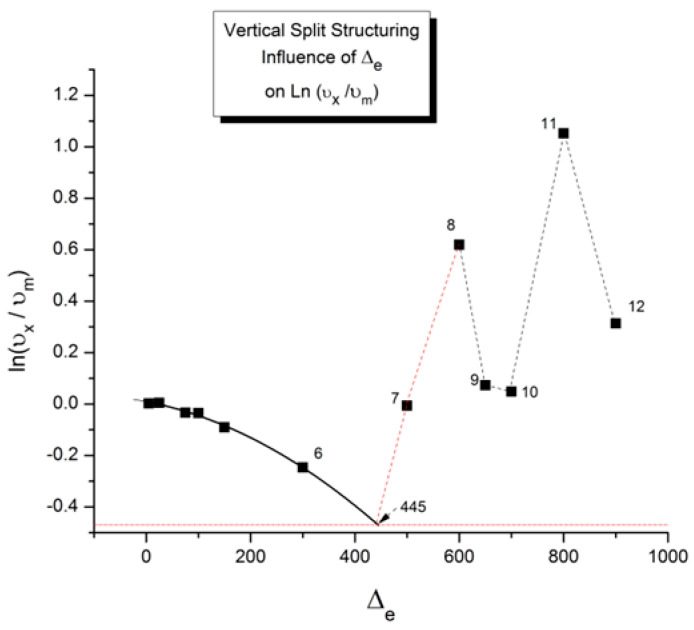
Variation of Ln(υ_x_/υ_m_) against Δ_e_ at υ_m_ and Δ_m_ constant, for the data of Figure 67 (Table 6). Reproduced with permission from [4], SLP Press, 1993.

**Figure 70 polymers-17-00239-f070:**
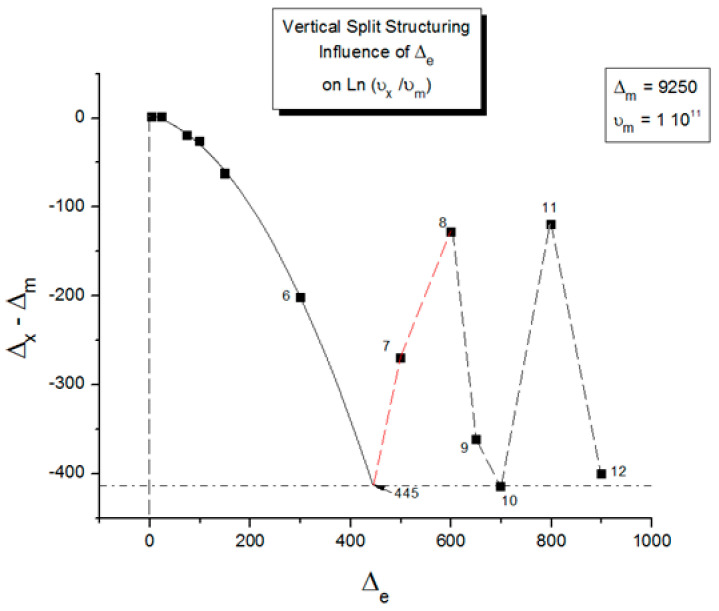
Variation of (Δ_x_ − Δ_m_) against Δ_e_ at Δ_m_ and υ_m_ constant, for the data of Figure 67 (Table 6). Reproduced with permission from [4], SLP Press, 1993.

Now, the analogy between these two types of relaxations, both complex and difficult to analyze, namely the dual-phase relaxation when Δ_e_ varies “widely” and the TWD relaxation of “widely” out of equilibrium samples when T_p_ varies, appears to be strikingly confirmed. We just need to qualify what “widely” means for the TWD results and the dual-phase dynamics simulations.

Consider Figure 68, Figure 69 and Figure 70, in particular Figure 68, on the one end, and Figure 45, Figure 46, and Figure 57 on the other end. Could it be suggested that the network of compensations observed for the TWD thermo-kinetics of these out-of-equilibrium samples is generated by a combination of two effects on the value of Δ_e_ of the dual-phase system simulating the dynamics of the interactions when the sample is relaxing to return to its equilibrium state? The two effects that modify Δ_e_ are as follows:The thermo-mechanical history during the processing of the sample;The thermal–dielectric history during its TWD characterization.

The coupling of both of these affects the value of Δ_e_(E, T_p_, t) of Equation (22) in a way that brings its value to the range that appears chaotic but is in fact structured, as suggested in Figure 68. As the TWD sample anneals at different stages of the TWD protocol, for instance, as T_p_ increases or during depolarization at T_d_ during a time t_d_, and the heating stage at a constant rate, the value of Δ_e_(E, T_p_) decreases, passing through the different values of Δ_x_, Ln υ_x_ from left to right in Figure 68, revealing the multi-compensation structure of Ln υ_x_ vs. Δ_x_ along the way.

Our tentative conclusion for Section 3.3, provided our analogy assumption is founded, is that the TWD characterization technique reveals the dissipative mechanisms of the structuration of the interactions that we have assumed are the basic assumptions controlling the behavior of polymers. In such a case, the TSD/TWD method should somehow lead to the determination of the fundamental parameters of the statistical model of polymer interactions introduced in Section 1.2. The value of Δ_e_ is equal to Δ_eo_ at equilibrium, but we cannot measure it directly, except at T = T_LL_. The values of υ_m_ and Δ_m_ are unknown, especially since what we are measuring are υ_x_ and Δ_x_, except at T_x_, the thermodynamic value of T_LL_, where Δ_e_ = 0 and υ_m_ = υ_x_ and Δ_m_ = Δ_m_. In order to find T_x_, we must modify the state of non-equilibrium of the amorphous phase by mechanical means and/or by dielectric means in order to access it by extrapolation.

### 3.4. Effect of the Voltage Field on the Dielectric Medium; on the Dual-Split Statistics of Interactive Coupling; New Interpretation of Thermal-Windowing and of Compensations; The Nature of What TWD Is Actually Deconvoluting; and on the Potentiality of the TWD Methodology to Find the Fundamental Parameters of the Dual-Phase Model

#### 3.4.1. Effect of the Voltage Field on the Dielectric Medium

The voltage field (E) and the polarizing temperature (T_p_) are a priori two independent variables: one can arbitrarily choose the value of E and T_p_. Yet, this does not consider the interactions induced by the dielectric nature of the material exposed to such a voltage at such a temperature. In fact, the external field, E_o_, triggers the presence of a different field, E, in the material and their ratio is the dielectric constant, ε = E_o_/E, expressing the “polarizability”, i.e., the tendency of a material to polarize in response to an applied electrical field. The field inside the dielectric, E, is the vector sum, E = E_o_ + E_p_, where E_p_ is generated by the polarization P of the dipoles in the dielectric: E_p_ = −P/ε_o_, where ε_o_ is the permittivity of the vacuum. The polarization is usually proportional to the macroscopic field: E_p_ = −χE, where χ is called the macroscopic susceptibility of the dielectric. The susceptibility, χ, and the polarizability P = ε_o_ (ε − 1)E_o_/ε are equivalent constants to express the polar nature of the chemical structure of the polymer. The Clausius–Mossotti expression of the dielectric constant (Eq. (3.20) p. 245 in [5]) correlates the molar mass of the dual conformer, its density, the total number of activated dipoles, the normalized induced moment, the permanent moment, and the Boltzmann thermal energy. In other words, the dielectric constant (ε) primarily depends on the chemical nature of the conformer (which controls the type and the magnitude of polarizability) but also varies with the physical state of the polymer, i.e., whether it is a solid, paste, liquid, or liquid crystal, in equilibrium or out of equilibrium, etc., all of which are controlled by the temperature at polarization and by the thermo-mechanical history of the sample, which also determines its non-equilibrium state. In summary, T_p_, E, and ε are interactively coupled in a circular way, and the best way to describe it is with an implicit function of the type H(ε, T_p_, E) = 0. This expression would greatly simplify if we could find a simple expression for the temperature dependence of ε. However, ε is the sum of various contributions: the electronic permittivity, the atomic permittivity (both quasi-time-independent, i.e., rapidly in steady state), and the dipolar (orientation) permittivity, with its characteristic spectrum of relaxation and its sluggish time dependence, characterized by τ(T). The main idea behind TSD and TWD is precisely to characterize the temperature dependence of all the relaxation times of the electronic, atomic, and molecular motions and their interdependence to create the cohesion of the bonds and relate it to the dielectric response as temperature varies. We can, therefore, rewrite the implicit equation by inserting the two constants that define the temperature dependence of τ(T): I(τ_oc_, ΔH_c_, T_p_, E) = 0. The subscript “c” in τ_oc_ and ΔH_c_ stands for “compensation”, which this paper has shown to describe “ranges of similar dielectric behavior”. In the newly written implicit equation, the emphasis has shifted from the expression of ε to describe the behavior of dielectrics, the classical approach, to the compensation approach, i.e., the description of the interactive couplings between the enthalpy and entropy at different “scales”, where scales refer to the ranges of dielectric behavior. By enthalpy and entropy, we, of course, refer to the physical sense attributed to the slope and intercept in the Arrhenius plane of Ln(τ(T)) vs. 1/T but also to the x and y coordinates of compensation plots such as Ln(τ_o_) vs. ΔH when T_p_ varies. By “different scales”, we symbolize the observed changes in the mechanism of interaction between the voltage field, temperature, and the dielectric material, in a way similar to, but fundamentally different from, the classical school description of the various contributions to the dielectric constants (electronic, atomic, and dipolar). In our dual-phase dissipative model of interactive coupling, the normalized variables of the master plot have two similar terms, (Δ_m_ − Δ_x_) for the x-axis and (Ln(υ_m_) − Ln(υ_x_)) for the y-axis. It is shown elsewhere [7] that the “m” and “x” terms do not play the same role: only Δ_x_ and Ln(υ_x_) vary with Δ_e_; Δ_m_ and Ln(υ_m_) remain constants within the “scales”, i.e., within the “three ranges of T_p_” that create a master curve by “grid-shifting”. Δ_m_, υ_m_ define the scale, and Δ_x_, Ln(υ_x_) define the dissipation character of the dual-phase dynamics. In summary, as detailed in another publication [Figure 3.56 p. 217 of [7]), we need both the dissipative contribution and the scale contribution to complete a one-line master plot of the network of compensation, which goes through the origin for the three scales (corresponding to the different nature of the dielectric behavior below T_g_, between T_g_ and T_g_ + 25, and T_g_ + 25 to T_LL_. We have mentioned several times that above T_LL_ (range 4), there is no compensation and no interactive coupling, and the Debye relaxations are parallel to each other as T_p_ increases, only shifted by the Boltzmann thermal energy, marking the end of EKNETICS.

#### 3.4.2. Effect of the Voltage Field on the Statistics of Interactive Coupling

The dual-phase model claims that a polymer that is not disturbed by an external field (mechanical, electrical field, etc.) but is just simply submitted to thermal activation alone (cooling, heating, and annealing) behaves like a self-dissipative statistical system working at Δ_eo_ constant that generates specific EKNETICS non-equilibrium characteristics to the amorphous phase: upon cooling, a T_LL_ temperature characteristic of the start of the EKNETICS is observed, followed by the T_g_ transition freezing the changes in the population dual conformation states; upon heating, its own dissipative statistics reactivates the changes in the dual conformational states at T_g_ until T_LL_ is reached, marking the end of the EKNETICS dissipative statistics, followed by a classical kinetic return to equilibrium. Yet, in order to determine the values of Δ_eo_, Δ_m_, and υ_m_, we need to modify the original value of Δ_eo_ by imposing either a voltage field, E, in the instances of this review (Equation (22)) or a mechanical stress field (shear and/or pressure) in other instances (Part I, Chapter 4, [13]), before freezing the sample into a glass. Then, we need to follow and characterize the changes occurring during the return to equilibrium of the sample induced by thermal activation means (DSC, TSD/TWD, DMA, etc.). Pursuant to the application of the voltage field or/and the mechanical field, the statistical dual-phase system has the same Δ_m_, υ_m_ set of values but a different value of Δ_e_, a “thermally dielectrically stimulated value of Δ_eo_” (Equation (22)). Polarizing the sample at various T_p_ values precisely allows for changing the value of Δ_e_ in the simulation before quenching and thermally depolarizing the material. By repeating the process at various T_p_ values, we can practice what the dual-phase simulation does: change the value of various Δ_e_ to follow its impact on the EKNETICS. Finally, the value of Δ_e_ = Δ_eo_ is obtained by determining T_LL_ either from TSD or DSC (Figure 36 and Figure 37) or TWD (T_p_ for ΔS_p_ = 0 in the negative compensation region). The variation with Δ_eo_ in the dissipative dual-split pair (Δ_x_, Ln(1/υ_x_)) simulates the variation with T_p_ in the experimental TWD pair (ΔH, Ln(τ_o_)). By sequentially scanning at various T_p_ values from a low temperature below T_g_ (range 1) to higher temperatures in the rubbery (range 2) or liquid state (range 3 below T_LL_ and range 4 in the Boltzmann state), the TWD procedure essentially reveals the interactive coupling between the dual-conformers; the coupling between the conformational interactions and the free volume; and the organization of the collective set of all the interactions as a network, i.e., the existence of a global dissipative network of the interactions. We acknowledge that this interpretation of the thermally stimulated depolarization results is new and unconventional.

#### 3.4.3. New Interpretations of “Thermal-Windowing” and “Compensations”

As far as the traditional interpretations of compensations observed by TWD are concerned [43,44,45], some reservations should be raised: these models stipulate that the entropy and enthalpy calculated from the Arrhenius spectral lines are equal to N times the value of a basic elementary unit with entropy and enthalpy ΔS #, ΔH #. In our opinion, this interpretation arising from trying to understand the “unreasonable” large values found for the magnitude of ΔH and Ln(τ_o_) at T_g_ does not explain the asymmetry for ΔS_p_(T_p_) and ΔH_p_(T_p_) on the positive and negative sides of the T_p_ = T_g_.peak. Moreover, these authors do not recognize the interactive coupling characteristic of the relaxation modes for T_p_ > T_g_ and thus ignore the differences between positive and negative compensations across T_g_. The distinction between a positive and negative compensation at the crossing of T_g_ is crucial to understanding the reason why the entropy and enthalpy pass through a maximum at transitions, such as at T_g_, and explaining why the behavior is asymmetrical crossing T_g_. In this Section 3.4.3, we argue that our understanding of compensations in TWD experiments by analogy with compensations in the dual-phase simulations really comes down to questioning the “classical” physical meaning of thermal-windowing itself (sometimes designated “thermal sampling”): does TWD really deconvolute elementary Debye peaks, the traditional interpretation, or does it deconvolute the dual-split complexity of the interactions, as we suggest?

#### 3.4.4. The Nature of What TWD Is Actually Deconvoluting

The interpretation of thermal-windowing as a filtering process takes its source in the description of the relaxation process of a viscoelastic material with the concept of a spectrum of relaxations. In this context, thermal-windowing, also called thermal sampling, consists of a physical deconvolution of the spectrum of relaxations to isolate its elementary Debye components. We still agree with the general statement that something needs to deconvolute, yet we now offer a different approach to what is deconvoluting. Our dual-split interpretation of the effect of the voltage field at temperature T_p_ is to modify the value of Δ_eo_ to become Δ_e_ according to Equation (22), which perturbs the statistical state of the dual-conformers, namely their (b/F) free volume state and their (c,g,s) spatial conformer state, resulting in a sample out of its original state. The mechanism of perturbation and the magnitude of this perturbation essentially depend on the original state of the material being polarized at T_p_, whether it is a frozen glass below its T_2_ temperature or above its T_2_ temperature or a rubbery or molten polymer above its T_g_ (T_2_ is the value of the temperature that makes the Newtonian viscosity infinite in the Vogel–Fulcher equation (sometimes designated T_∞_). It is assumed that the free volume is zero at T_2_, removing any possibility of motion left to single conformers’ internal rotation. Hence, in order to evaluate the amount of disturbance by the application of the voltage at T_p_, we need to position T_p_ with respect to T_2_; T_g_; and, as we have seen, T_LL_. The other steps of the TWD procedure are carried out without the presence of an electric field: the return to the equilibrium state of the system, Δ_e_ = Δ_eo_, is activated by temperature and time only, at T_d_, t_d_, and during the ramp-up at constant ramp rate. We are able to observe the return to equilibrium because it involves the depolarization of what was polarized. It should be clear that the polarization step is not the only cause of the non-equilibrium state of the material. The material processing history also created a non-equilibrium state for the interactions between the dual-conformers. This is true for all amorphous matter cooled below its T_g_. In summary, the return to equilibrium as temperature activates the relaxation to this stable state does not simply involve the return of what was disturbed dielectrically, but also, on top of that and/or besides that, the return to the stable state of the dual-conformers. The analysis of the depolarization stage is, therefore, the combination of the relaxation of the dielectric disturbance due to the voltage and the relaxation of the disturbance due to the effect of the temperature history on the initial dual-phase state. This relaxation occurs as soon as the temperature and/or the voltage field changes for the sample inserted in the TSD equipment. This happens at all the stages of the TWD procedure, starting with the polarizing step at T_p_, the partial depolarizing at T_d_, and the depolarizing during the ramp-up to create the depolarization current–temperature curve. This represents a cycle. We could have stopped the procedure at the end of this first cycle and replaced the sample with a new one before repeating the procedure, but this is not what was conducted in this TWD procedure. After the first cycle, we changed T_p_ and repeated the same operation several times. It is important to realize that this is the same sample that had already been depolarized and annealed previously, perhaps many times already, depending on which thermal-windowing cycle we were starting. It appears that changing T_p_ is like changing the initial value of Δ_e_ of the sample. We are not talking, in this dual-phase explanation, about the successive relaxation of the discrete relaxations of a spectrum of relaxations, starting from the shortest relaxation times to the longest ones; we are talking of the evolution of Δ_e_(t) for the same sample with a different value of Δ_e_ and a different value of T_p_ in Equation (22) at the beginning of each cycle of depolarization. In conclusion, TWD does not consist of isolating elementary peaks from other elementary peaks; it involves deconvoluting the dual-split complex nature of the interactions resulting from the kinetic and energetic factors working in duality (Equations (6)–(8)). TWD reveals the gears of the Grain-Field Statistics. This is a major difference in the theoretical interpretation of TWD.

#### 3.4.5. On the Potentiality of the TWD Methodology to Find the Fundamental Parameters of the Dual-Phase Model

If the Grain-Field Statistics theory of the interactions (vertical and horizontal structuring) can successfully predict the properties of polymers by simulation, it becomes essential to know how to find the fundamental constants of the theory, Δ_m_, υ_m_, and Δ_eo_. Is the TWD protocol or even a modified TWD protocol (a quench/isothermal annealing experiment) capable of extracting the fundamental constants of the Grain-Field Statistics from its own characteristic parameters, i.e., the free energy ΔG_p_, its structure ΔH_p_ and ΔS_p_, the coordinates of the compensation points, the value of these parameters at T_g_ and T_LL_, etc.? For instance, Equation (22) proposes a physical correlation between the value of Δ_e_, T_p_, the voltage field E, and the dipole moment μ. Equation (23) incorporates the effect of the voltage in the expression of Δ_e_ via Δ_eg_, so the unknown constant k can disappear from the expression. Δ_eg_ itself can be expressed in terms of T_g_ and Δ_eo_ (Δ_e_ at T = T_LL_), and the equation of the Z line can be deducted from equations similar to Equations (14) and (15). The slopes and intercepts of the Ln(k_x_) vs. 1/T Arrhenius plots at various Δ_e_ (i.e., various T_p_) can also be converted into enthalpy–entropy variables using the same equations used to convert the Ln(τ_o_), ΔH plots into ΔH_p_ and ΔS_p_, allowing us to define the free energy ΔG_e_ (i.e., ΔG_p_) and write the compensation line equations in terms that are the same as those used to characterize the relaxation modes in the ΔG planes (ΔG = ΔH_p_ − TΔS_p_). We seem to be ready to find the fundamental constants of the dual-split model: Δ_m_, υ_m_, and Δ_eo_. Actually, the task is more complex than that, and it is impossible at this stage to achieve such a goal. The reason must be reemphasized again: the Ln(k_x_) vs. 1/T Arrhenius plots at various Δ_e_ do not provide the kinetic constants that could simulate the dynamic depolarization stage of the TWD, which is carried out at a constant heating rate and not under isothermal annealing conditions. The TWD thermal history is not a two-stage straightforward polarize/anneal process. As mentioned previously, it involves an intermediary stage, at T_d_, where the state of the melt is allowed to relax partially. In effect, the TWD procedure includes a short isothermal annealing step, the kind that we use in the simulation. This extra step makes the TWD procedure a three-step process, with this annealing step not recorded presently in the TSC/RMA instrument of Solomat, although it is straightforward to implement such an operation. Another way to match the depolarization history in the TWD protocol with an identical relaxation history for the dual-split simulations is to work from the simulation side and rewrite the algorithm to achieve this goal. However, we are still confronted with the choice of the parameters of the EKNETICS to implement to be able to perform such a similar protocol simulation. Moreover, we face another challenge: the variables that behave like Ln τ_o_ and ΔH are Ln(1/υ_x_) and Δ_x_, the dissipative values, not the Ln (1//υ_m_) and Δ_m_ values. The only possibility to assess the values of υ_m_ and Δ_m_ is when T_p_ = T_x_, the thermodynamic value of T_LL_, which can be obtained by extrapolation of T_LL_ at the zero rate. We believe that this can be achieved experimentally for any given polymer, but in the simulation, it requires knowing the value of υ_m_ and Δ_m_ to perform cooling or heating simulations at various rates. The simulations can be repeated, indeed, by changing the values of υ_m_ and Δ_m_, separately, and the cooling and heating simulations can be repeated at various rates to obtain the value of ln k_x_ vs. 1/T from which Ln(υ_x_), Δ_x_ and and the value of T_x_ matching the one found from the experiments can be extracted. This will necessitate a very large number of simulations. In summary, practically speaking, it appears extremely tedious, perhaps even unrealistic, to easily find the parameters of the dual-phase model by comparing results obtained by a modified TSD/TWD protocol and simulating the same protocol pursuant to the dual-split equations.

## 4. Conclusions

This communication reviews the comprehension of “interactive coupling” between molecular motions in amorphous polymers using data generated by thermally stimulated depolarization (TSD) and thermal-windowing deconvolution (TWD), analyzed either “classically”, pursuant to work published on the subject [1,2,3,6] or “unconventionally”, using the language of the dual-phase model of polymer interactions [10,11,12,13]. TSD and TWD are two characterization methods that involve thermally inducing polarization in dielectric samples using a voltage field followed by depolarization by heating the sample at a constant rate. TSD and TWD provide a powerful way to quantify the “thermo-kinetic” state of amorphous matter by studying the local and cooperative relaxations occurring during the depolarization stage. The understanding of the amorphous state of matter is, in our opinion, essential to understanding the glass transition, molecular motions in the rubbery and molten states, and even the fundamental mechanisms leading to crystallization from the amorphous state.

This review summarizes the fundamentals of this dielectric characterization technique using the “classical approach language”:-The polarization and depolarization stages, the description of the current of discharge using Bucci’s equations [14,15]; the filtering of the spectrum of relaxations to isolate various elementary Debye relaxation peaks, and the compensation of the Debye relaxation times deconvoluted by “thermal-windowing”;-The origin of the dipole formation, induced or permanent dipoles, and the origin of the Wagner space charges;-The observation of the T_g,ρ_ peak just above T_g_ and of the T_LL_ manifestation, the spectroscopic nature in a relaxation map, and the description of their compensation and, for certain specimens (rheomolded), their multi-compensations and super-compensations (the compensation of the compensation lines themselves).

This review also introduces the “language” of a new physics of polymer interactions [10,11,12,13], the dual-phase theory, and applies it to simulate the relaxation of a closed system of dissipative dual-conformers that are assumed to be the statistical basic units that interactively couple to explain the viscoelastic properties of polymers. We use the language of this “unconventional” statistics (with the introduction of a dissipative term in the energy equation) to describe and understand the dielectric properties of amorphous polymers. It is assumed that the dual-conformers are polarized to generate the dipoles and that the space charges localize at the space around the b-grains (F-conformers). The simulations explain the dynamics of the dissipative system of dipoles upon cooling, heating, and relaxation and show, for instance, how cooling at a constant rate brings the system out of equilibrium in a way different from a non-dissipative system. The temperature at which the system starts to become dissipative (upon cooling) or ceases to be dissipative (upon heating) is assimilated to the T_LL_ transition easily detectable by TSD and TWD manifestations. Additionally, the presence of the F-conformers (attributed to the free volume around the b-grains) attracts and localizes the Wagner space charges in the amorphous structure and explains the presence of the T_g,ρ_ peak in TSD experiments. The dissipative thermo-kinetics parameters, υ_x_ and Δ_x_, which are self-generated by the inducement of the non-equilibrium states, are fundamental parameters of this new kinetics (the EKNETICS), which are associated by analogy to the thermo-kinetic parameters 1/τ_o_ and ΔH, extracted from the analysis of the relaxation map in TWD experiments. The analogy between the simulated relaxation dynamics of the dual conformer and the relaxation dynamics of the dipoles during the depolarization stage explains the structuring of their interactive coupling as positive and negative compensations across the T_g_ transition. The super-network of compensation structures observed for the rheomolded PS samples (brought out of equilibrium by mechanical means) can also be simulated. Interestingly, a relaxation map that appeared complex and incomprehensible at first in Figure 45 and Figure 46 turned out to have a very simple dual-phase explanation to comprehend it (Figure 68). This apparent successful theoretical breakthrough to explain compensations and super-compensations may be regarded as a sort of validation of the power of the TSD/TWD thermal analysis technique to measure up the amorphous state of electrets [5]; additionally, on the theoretical side, it may also provide the proof of the benefits of using the dual-phase statistics to simulate the properties of polymers, including to provide a coherent interpretation of the T_LL_ and T_gρ_ manifestations.

There is also another dimension to this review: we argued in Section 3.4.3 that our understanding of compensations in the TWD experiments based on the analogy with compensations in the dual-phase simulations challenged the “classical” physical meaning of thermal-windowing itself (sometimes designated “thermal sampling”); the dual-phase approach suggests that thermal-windowing reflects the deconvolution of the dual-split complexity of the interactions, with Δ_e_ decreasing toward Δ_eo_ as T_p_ varies (Figure 68), which contradicts the classical interpretation of thermal-windowing as the deconvolution of global peaks into elementary Debye peaks. Likewise, the classical “local order” explanation of the T_LL_ manifestations (TSD in Figure 36, TWD in Figure 11 (ΔS_p_ = 0)) and the T_g,ρ_ peak (Figure 30 and Figure 31) is not consensual among the protagonist users of the TSD/TWD technology [1,2,3,6]. The interpretation of Lacabanne’s school regarding the compensation of ΔH and Ln(τ_o_) [6] was challenged by Sauer and Moura Ramos [42], who raised the issue of whether compensation had a real physical meaning. The dual-phase response to that question validates the question of these authors by providing a new interpretation of the compensations than the classical answer. However, the questioning of the core interpretation of thermal-windowing was the good bargain used by the adepts of the current paradigm to discredit the reality of T_LL_, a transition notably more visible by TSD/TWD than by other technologies [40,41]. There is no theoretical resonance in the framework of the current paradigm of polymer physics for T_LL_ and T_g,ρ_ [12,13]. The backers of the molecular dynamic models of polymer physics (the gatekeepers of the current paradigm) rightfully recognize the existential threat to their paradigm of the existence of the T_LL_ transitions and claim that it is an experimental “artifact” [40,41]. Finally, the question of the negative compensations found for T_p_ > T_g_ and of the Z structure at T_g_ (Figure 11) is never mentioned by any of the protagonists of the TSD/TWD, (as if it was not a reality), nor by the gatekeepers of the current paradigm, who prefer to ignore all the facts that they cannot designate “artifacts”. Hence, in this context of theoretical questioning [40,41,42], it seems clear why the TSD/TWD methodology has only been partially adopted as a powerful characterization technique. 

Yet, the recognition of T_LL_ is essential, in our view [10,11,12,13,14], for the understanding of the dissipative nature of polymers pursuant to the dual-phase theory (Figure 62) and also to comprehend the logic behind the “disentanglement” processing technologies to improve the fluidity of melts (rheo-fluidification) and pellets (“sustained orientation”) benefitting their final properties [10].

In summary, we believe that a broader recognition of the merits of TSD/TWD in thermal analysis requires the correct interpretation of the T_LL_ transition and of T_g,ρ_, and, moreover, the reality of compensations [42]. This implies a clear and straightforward presentation of the facts regarding their experimental manifestations and a solid theoretical explanation of how their physical presence can be integrated with the rest of the properties of polymers. This requires, in our opinion, making advances on two fronts: 1. presenting a new understanding of the TSD/TWD results (this is the objective of this review) and 2. making the current paradigm of polymer physics (molecular dynamics) obsolete by providing a different explanation of the experimental facts by the dual-phase theoretical language. The molecular dynamic models, despite their many successes, are presently facing too many deficiencies, which continue to be ignored [12,13]. The experimental evidence for which the current paradigm of polymer physics is called deficient is explained in several papers [10,11,12,13], as well as, implicitly, in this review of the TSD/TWD results. The deficiency of the molecular dynamic models regards various aspects of the viscoelasticity of the melt: in rheology, the non-linearity of shear-thinning [10,11] and the misconception of entanglements [12]; in processing, the incapability to comprehend “sustained orientation” by rheo-fluidification [10,11,12,13]; and, in this review and in [5], failing to consider the existence of fundamental transitional behavior observed by TSD/TWD: the T_LL_ transition and the T_g,ρ_ peak. The alternative dual-phase interpretation for those same experimental facts has been published for the last 10 years [5,8,10,11,12,17] and yet has not received any contradiction/interest.

In our view, “dual-conformers”, the constituents of macromolecules, gather into statistical systems that go beyond belonging to individual macromolecules. A conformer is shown in Figure 13, duplicated from reference [19]. The macromolecules themselves represent a chain of “covalent conformers” put together as an entity. The problem is to determine whether the chain properties, derived from its statistics, entirely control the dynamics of the collection of chains making up a polymer. This is what has been assumed by all the other theories, and this is what the dual-split kinetics and the Grain-Field Statistics challenge. In our opinion, this is a key issue: the currently established theoretical models of the interactions in polymers are based on “chain dynamics” statistics. In rheology, for instance, the Rouse and reptation models are dominant for M < M_c_ and M > M_c_, respectively [11]. The conclusion obtained from the present work, based on the Grain-Field Statistics of the interactions, is that the application of macromolecular (chain) dynamic models could only be justified for conditions of use that position its temperature above the T_LL_ transition (T > T_LL_). The significant problem is that the existence of T_LL_ is not even recognized by these macromolecular dynamic models. Below T_LL_, the free energy of the collection of chains assembled as a polymer is not equal to the scaled-up free energy of a macromolecule embedded in a mean field created by the influence of the other macromolecules. Moreover, the temperature T_LL_ is itself a function of the dynamics of the experiment and the chain characteristics. The dual-phase model of polymer interactions does not require, in its hypotheses and derivations, a description of the changes that occur in the individual macromolecules. The dynamic statistical systems dealt with in this model, which are used to determine the free energy and its structure (enthalpy and entropy), are not the macromolecules. However, the fact that macromolecules compose the basic structure is essential to understanding the basis of our new dual-phase statistics and explaining “entanglements”, for which the dual-phase model provides a completely different interpretation than the ones offered by the conventional macromolecular dynamic models [11,19]. A “dual conformer” is not the same as a “free conformer” (Figure 13), defined from the monomer repeat unit involved in the polymerization process. Its interaction with other conformers by covalent bonding modifies the conformational potential energy of a free conformer, and this governs the statistical properties of a free chain. When dealing with a collection of chains put together, our approach differs from the classical one. Dual-conformers belong to two types of sets: they belong to macromolecules, which link them via covalent forces, as we just said, and they belong to the grand ensemble of conformers, which are linked by inter–intramolecular forces, van der Waals forces, dipole–dipole forces, and electrostatic interactions which affect and define the viscous medium. That duality is intrinsic to conformers, which we call the “dual-conformers” to mark this specificity. The potential energy of a dual conformer is different from the potential energy of a conformer part of a free chain. To simplify, one could view the difference between our statistical model and the classical model to describe the properties of polymers as follows: According to the classical views, the statistical systems are the macromolecules, i.e., a network of chains; the properties of the chains are disturbed by the presence of other chains and by the external conditions (temperature, stress tensor, electric field, etc.). The classical definition of a statistical system as the macromolecule contrasts with our approach in which the statistical systems are the “dual-conformers”, not the macromolecules. The interactive coupling between the dual-conformers is defined by a new field of statistics, the Grain-Field Statistics, which explores the correlation between the local conformational property of the dual-conformers and their collective behavior as a dissipative network.

## Figures and Tables

**Figure 1 polymers-17-00239-f001:**
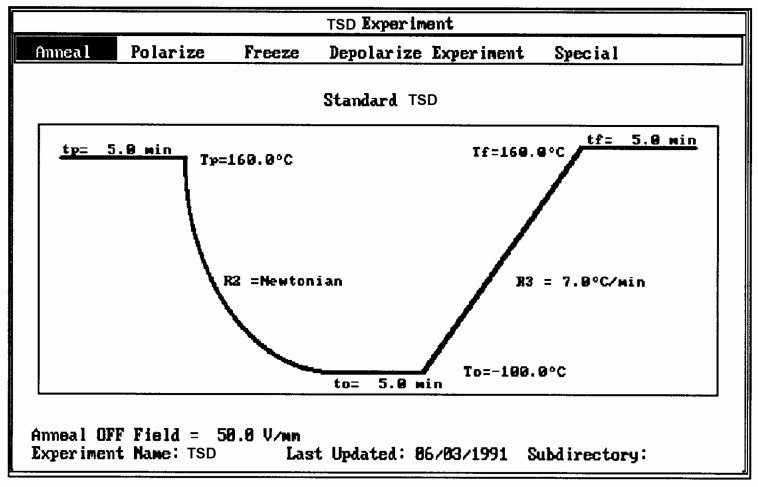
Description of the steps involved in a TSD experiment (polarization, cooling, annealing, and heating) resulting in the output (a depolarization current vs. temperature) by thermal stimulation. Reproduced with permission from [4], SLP Press, 1993.

**Figure 2 polymers-17-00239-f002:**
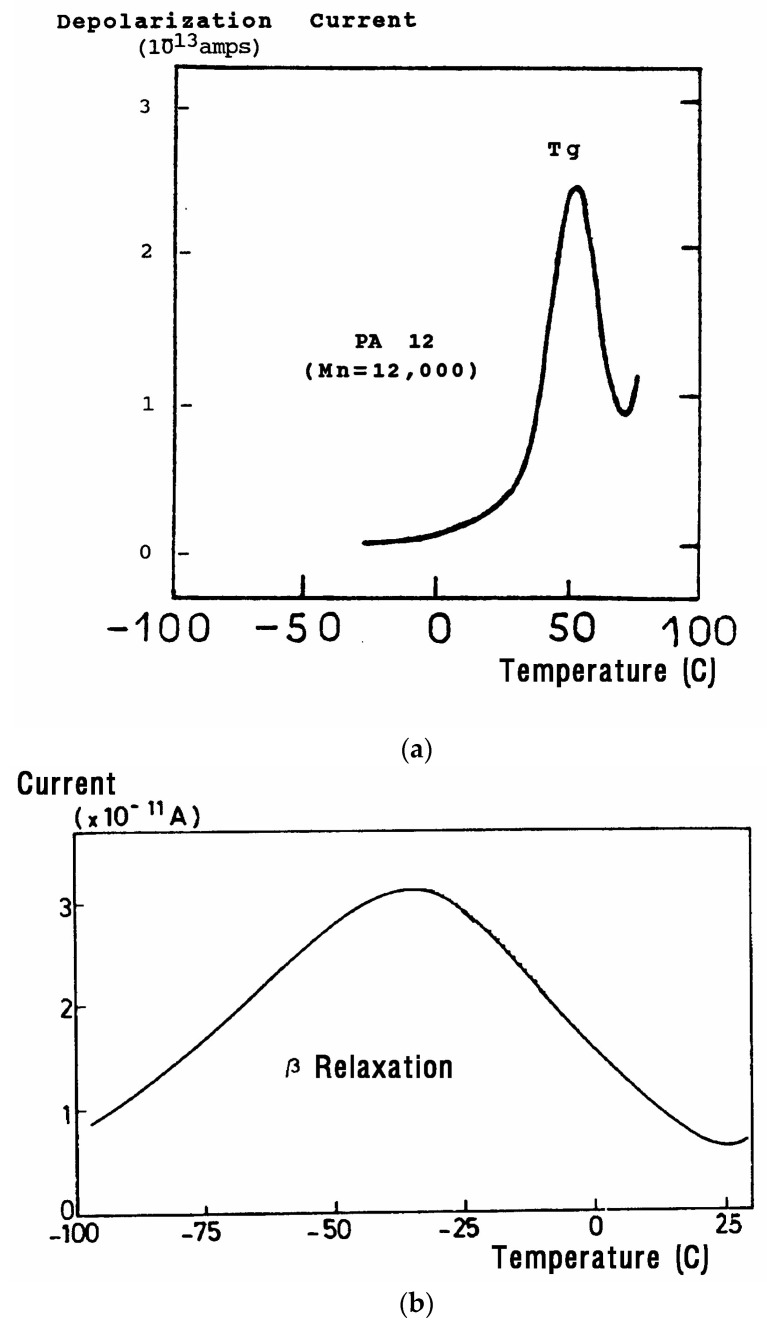
(**a**). Depolarization current vs. temperature during the thermal stimulation heating stage of Polyamide 12. (**b**). Depolarization current vs. temperature during the thermal stimulation heating stage for a polarization temperature near the β-transition of an amorphous polymer. Reproduced with permission from [4], SLP Press, 1993.

**Figure 3 polymers-17-00239-f003:**
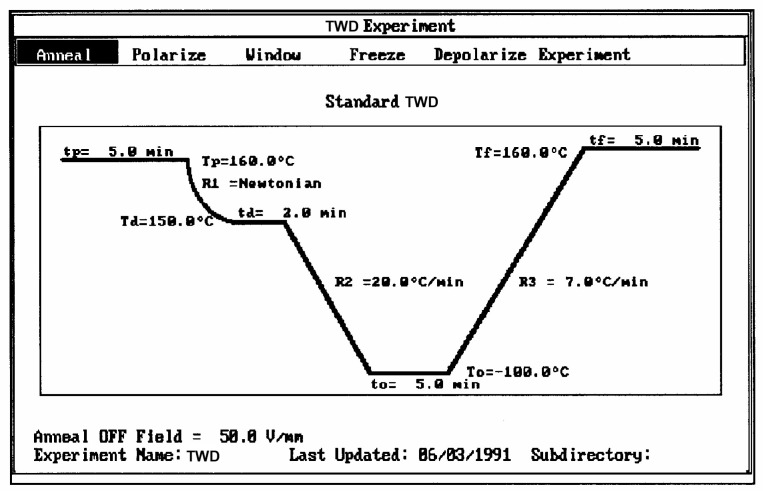
Description of the steps involved in a TWD experiment to thermally deconvolute a global TSD peak into its elementary Debye components and determine the interactive coupling between the relaxation modes. Reproduced with permission from [4], SLP Press, 1993.

**Figure 4 polymers-17-00239-f004:**
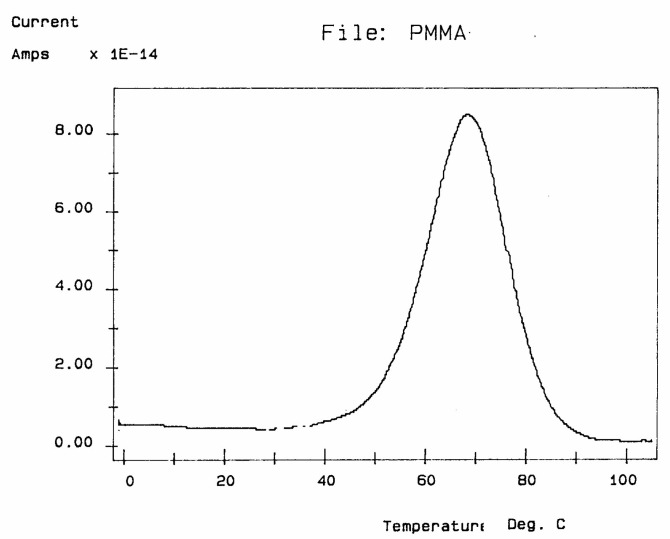
Current of depolarization vs. T for a TWD experiment. The polymer is PMMA. Reproduced with permission from [4], SLP Press, 1993.

**Figure 5 polymers-17-00239-f005:**
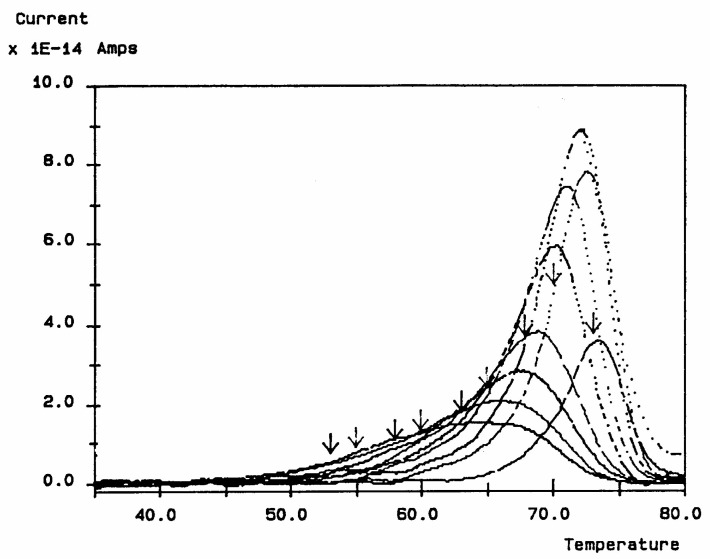
Effect of changing the polarization temperature T_p_ in a TWD experiment, indicated by the arrow, on the current of depolarization vs. T plot. Reproduced with permission from [4], SLP Press, 1993.

**Figure 6 polymers-17-00239-f006:**
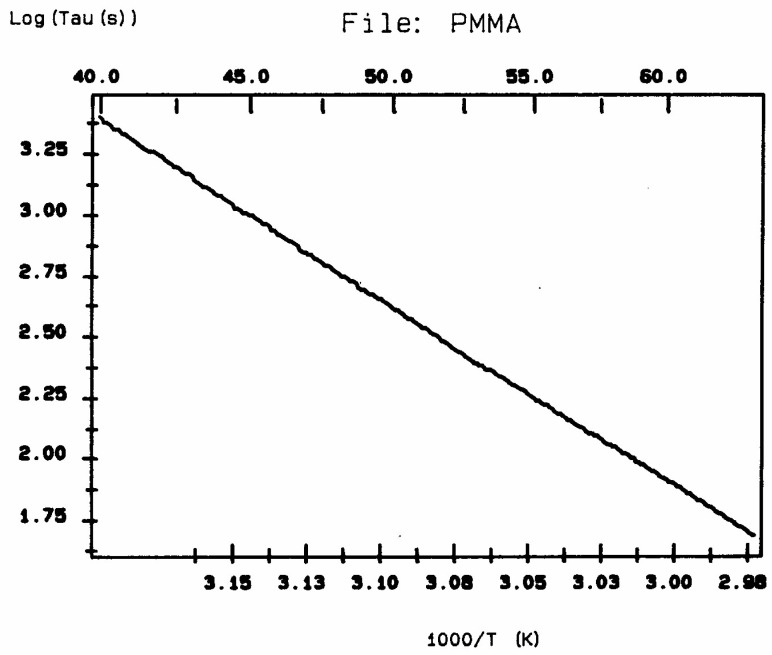
Conversion of the output in Figure 4 to an Arrhenius spectral line. The bottom axis is the Arrhenius scale, 1/T (K), in descending numbers; the top axis is the temperature in °C. The y-axis is the log of the relaxation time for the mode isolated by TWD at T_p_. Reproduced with permission from [4], SLP Press, 1993.

**Figure 7 polymers-17-00239-f007:**
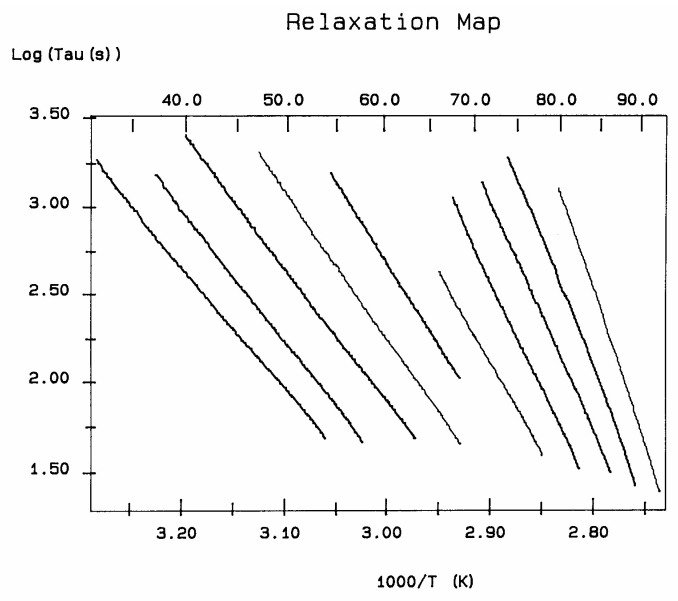
Relaxation map in the Arrhenius plane of all the relaxation modes isolated by TWD at various T_p_, as shown in Figure 5. Reproduced with permission from [4], SLP Press, 1993.

**Figure 8 polymers-17-00239-f008:**
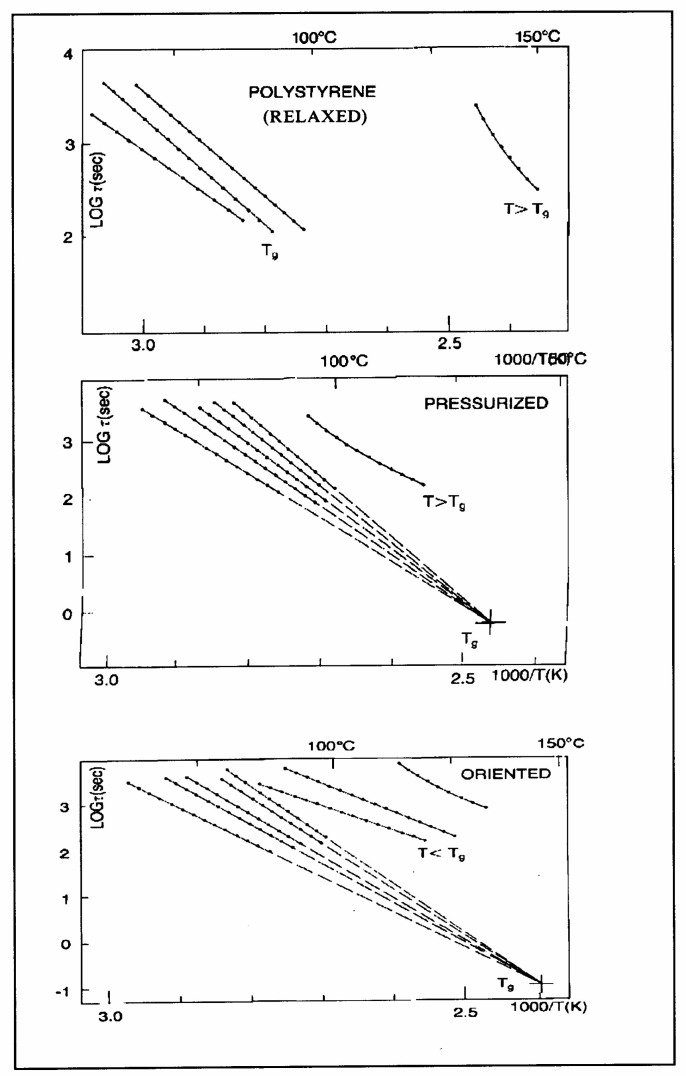
Illustration of the effect of various mechanical treatments of the melt during molding on the aspect of the relaxation map obtained by TWD of the glasses produced. Reproduced with permission from [4], SLP Press, 1993.

**Figure 9 polymers-17-00239-f009:**
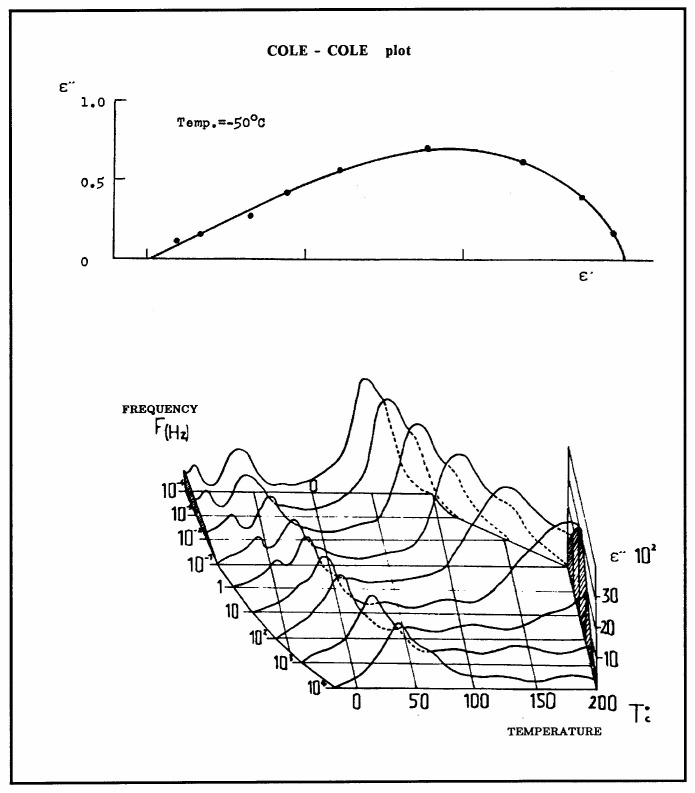
Response of a dielectric material to an AC voltage field. A Cole–Cole plot (top) consists of a plot of ε″(f) vs. ε′(f) at a given T; a frequency map is shown at the bottom for ε″(f,T). Such plots can be calculated from the TSD/TWD response. Reproduced with permission from [4], SLP Press, 1993.

**Figure 10 polymers-17-00239-f010:**
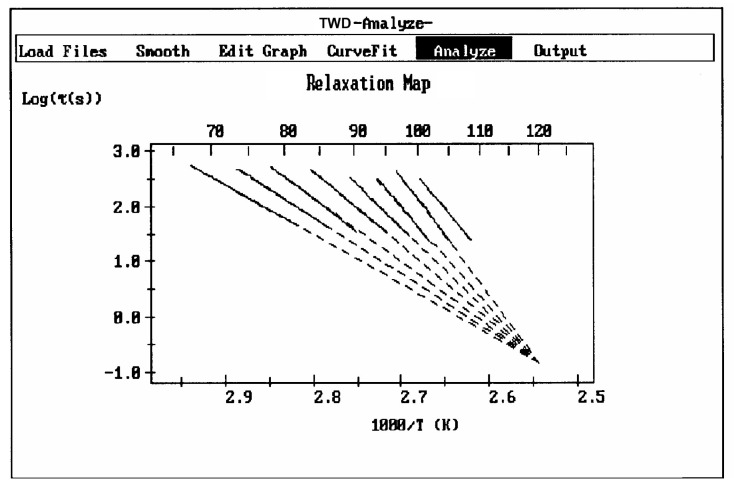
Relaxation map in the Arrhenius plane illustrated for PMMA (limited to T_p_ < T_g_). The spectral lines obtained at various T_p_ converge at a compensation point. The coordinates of the compensation point are assumed to reflect the state of the amorphous phase due to the interactive coupling between the relaxation modes. Reproduced with permission from [4], SLP Press, 1993.

**Figure 11 polymers-17-00239-f011:**
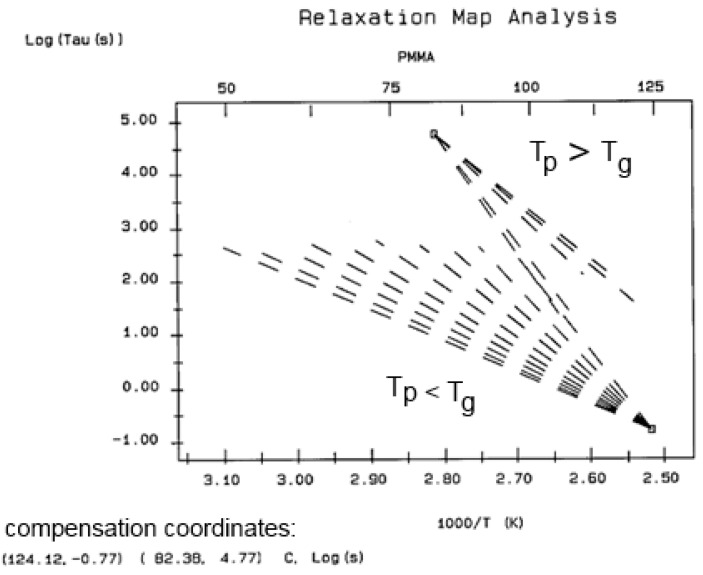
The “Z structure” of the T_g_ transition with a positive compensation of the Debye relaxations for T_p_ < T_g_ and a negative compensation for T_p_ > T_g_. The last relaxation of the interactive coupling network is the horizontal relaxation passing through the negative compensation point (log τ_c−_ = 4.77), which corresponds to T_p_ = T_LL_ (ΔS_p_ = 0), and the 1st relaxation of the interactive coupling network is the horizontal line passing through the positive compensation point (log τ_c+_ = −0.77), which corresponds to T_β_. Reproduced with permission from [4], SLP Press, 1993.

**Figure 12 polymers-17-00239-f012:**
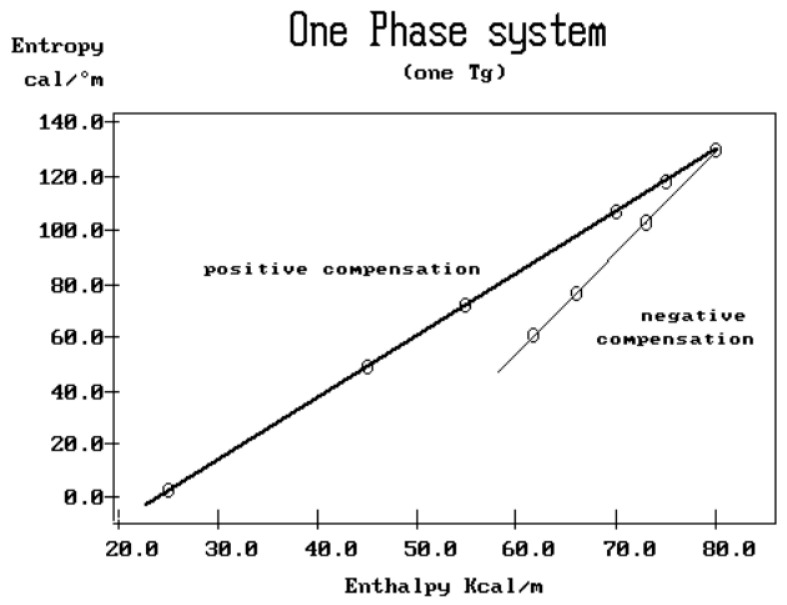
Compensation Search to determine the positive and negative compensation lines from a ΔG vs. T relaxation map. Reproduced with permission from [4], SLP Press, 1993.

**Figure 13 polymers-17-00239-f013:**
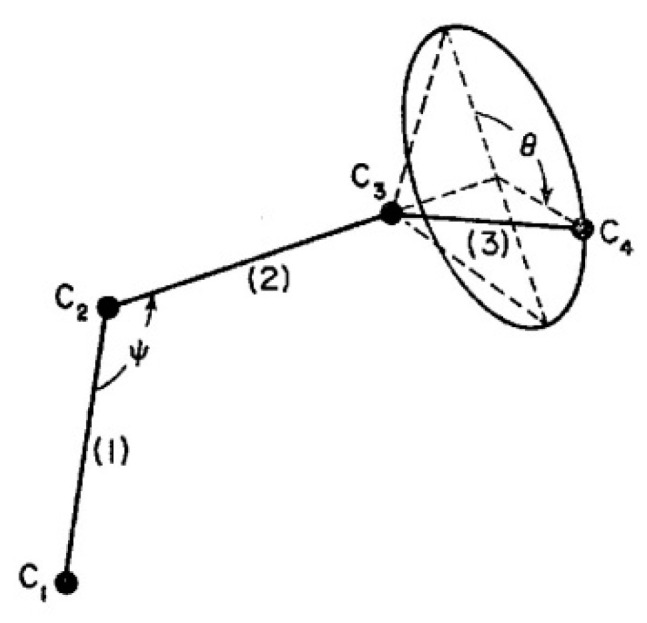
Sketch of a covalent conformer (Figure 1.2 of [10]), after Flory’s three-bond unit [19]. Reproduced with permission from [4], SLP Press, 1993.

**Figure 14 polymers-17-00239-f014:**
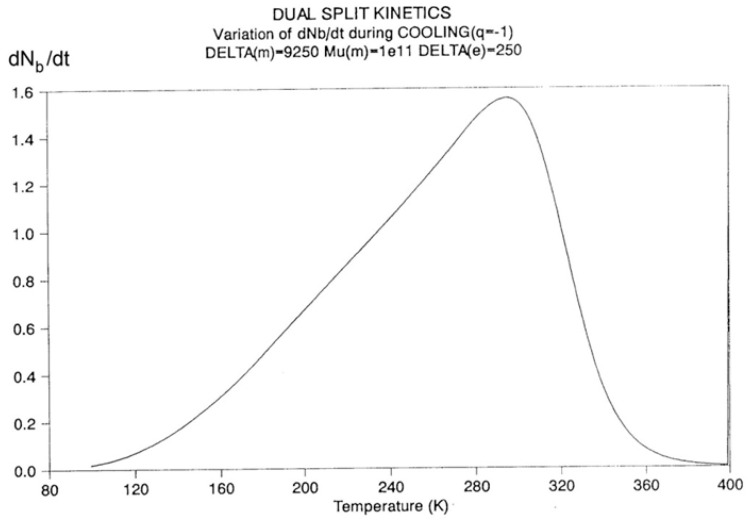
Variation in dN_b_/dt during cooling (q = −1) Δ_m_ = 9250, Δ_e_ = 250, υ_m_ = 10^11^, B_o_ = 1000, T_o_ = 400 K. The 1st peak observed (at ~ 300 K) is influenced by the value of the pair (Δ_m_, υ_m_), whereas the 2nd peak, only visible by a small hump at T~200 (K) in this Figure, is the reflection of the value of Δ_e_ on the kinetics. Reproduced with permission from [4], SLP Press, 1993.

**Figure 15 polymers-17-00239-f015:**
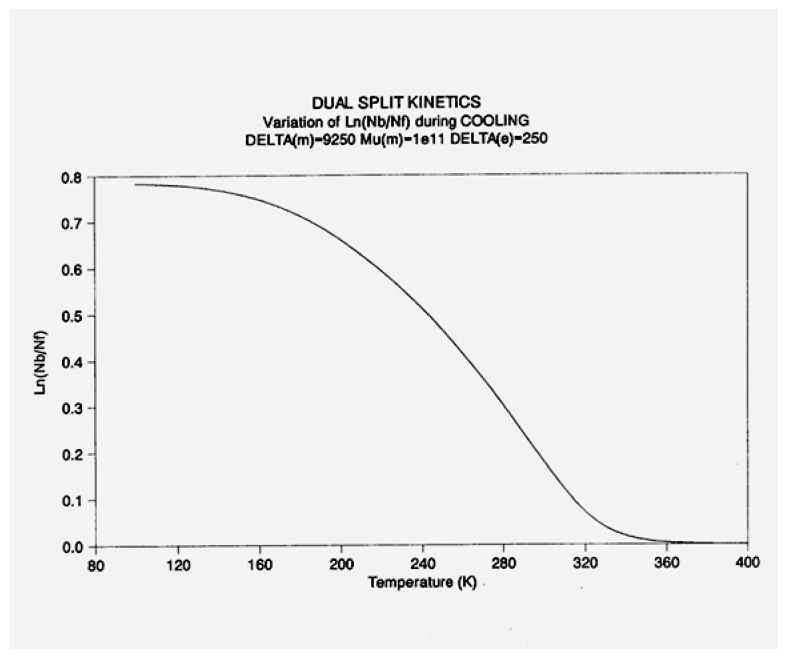
Dual-split kinetic simulation. Variation in dN_b_/dt during cooling (q = −1). Δ_m_ = 9250, Δ_e_ = 250, υ_m_= 10^11^, B_o_ = 1000, T = 400 (K). Reproduced with permission from [4], SLP Press, 1993.

**Figure 16 polymers-17-00239-f016:**
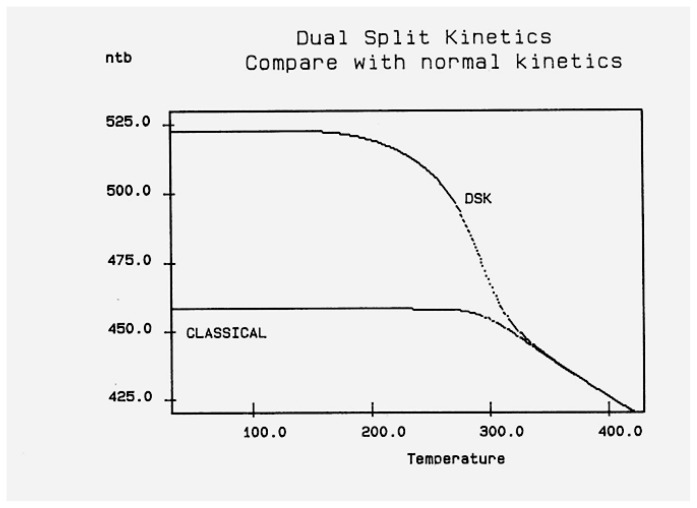
Dual-split kinetics (cooling at q = −1). Compare the simulations of the dual-split kinetics (DSK = EKNETICS) in Equations (6)–(8) and classical kinetics in Equations (1)–(3) using the same parameters (Δ_m_ = 9250, Δ_e_ = 250, υ_m_ = 10^11^, B_o_ = 1000, T = 400 (K)). Reproduced with permission from [4], SLP Press, 1993.

**Figure 17 polymers-17-00239-f017:**
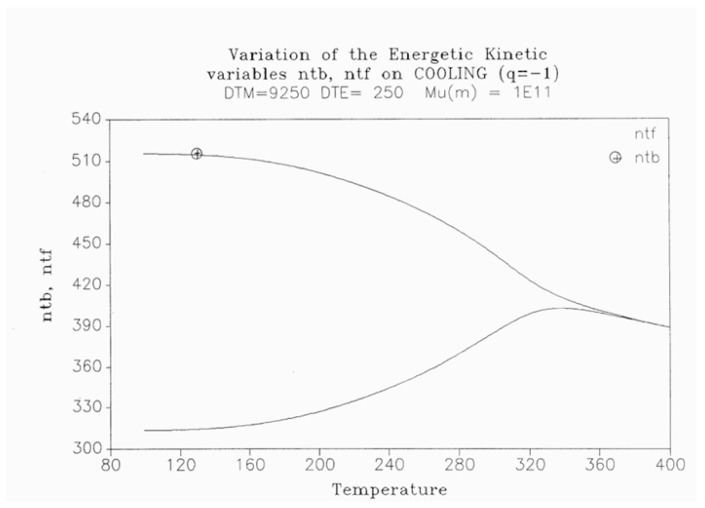
Variation in the energetic kinetic variables n_tb_ and n_tf_ during cooling (q = −1). Δ_m_ = 9250, Δ_e_ = 250, *υ*_m_ = 1011. Reproduced with permission from [4], SLP Press, 1993.

**Figure 18 polymers-17-00239-f018:**
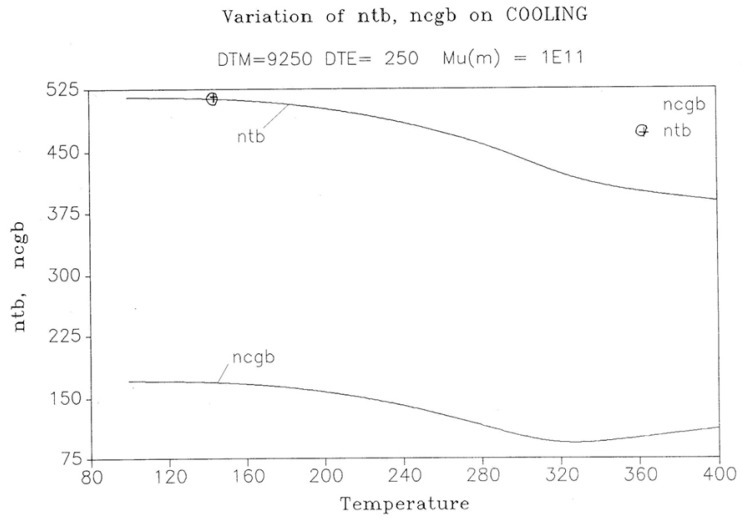
Variation in n_tb_ and n_cgb_ during cooling. Δ_m_ = 9250, Δ_e_ = 250, *υ*_m_ = 10^11^. Reproduced with permission from [4], SLP Press, 1993.

**Figure 19 polymers-17-00239-f019:**
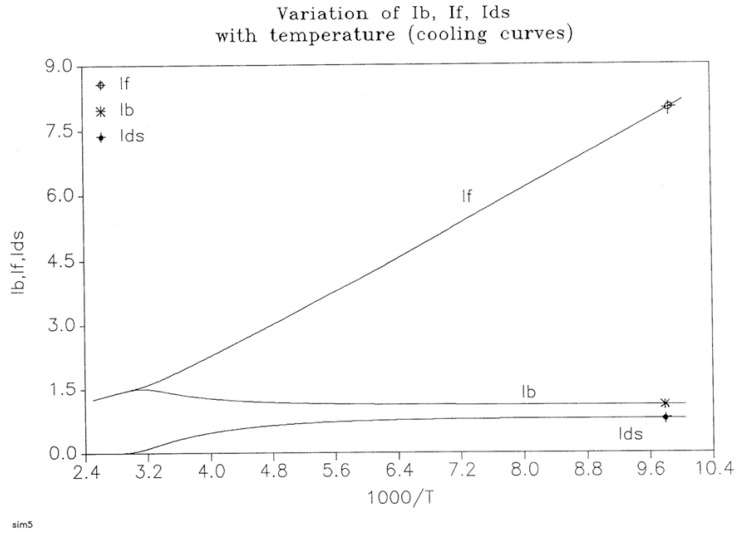
Variation in I_b_, I_f_, and I_ds_ with temperature (cooling curves). Reproduced with permission from [4], SLP Press, 1993.

**Figure 22 polymers-17-00239-f022:**
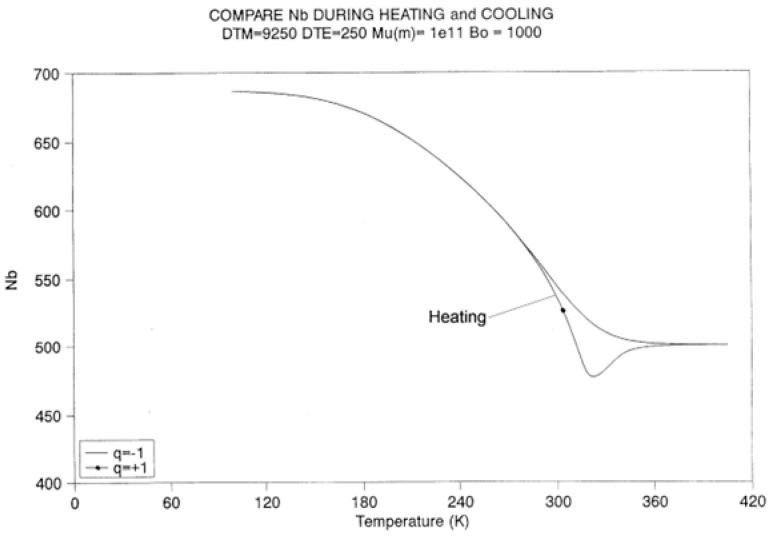
Compare N_b_ during heating and cooling. Δ_m_ = 9250, Δ_e_ = 250, *υ*_m_ = 10^11^, B_o_ = 1000. Reproduced with permission from [4], SLP Press, 1993.

**Figure 23 polymers-17-00239-f023:**
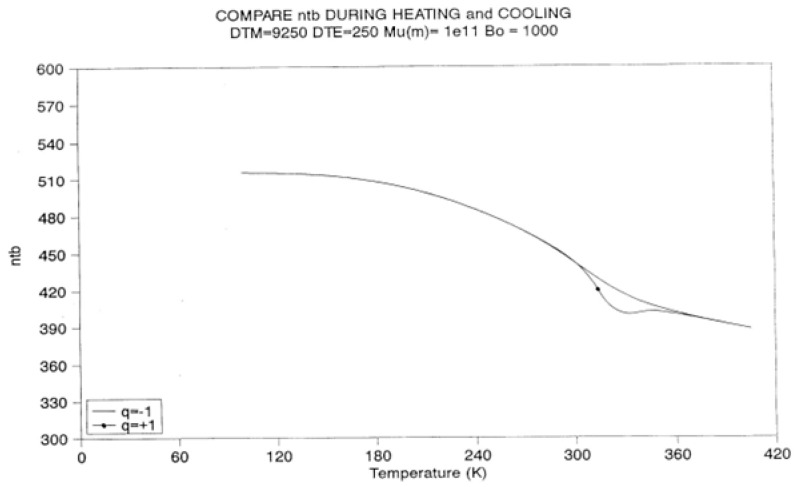
Compare n_tb_ during heating and cooling. Δ_m_ = 9250, Δ_e_ = 250, *υ*_m_ = 10^11^, B_o_ = 1000. Reproduced with permission from [4], SLP Press, 1993.

**Figure 24 polymers-17-00239-f024:**
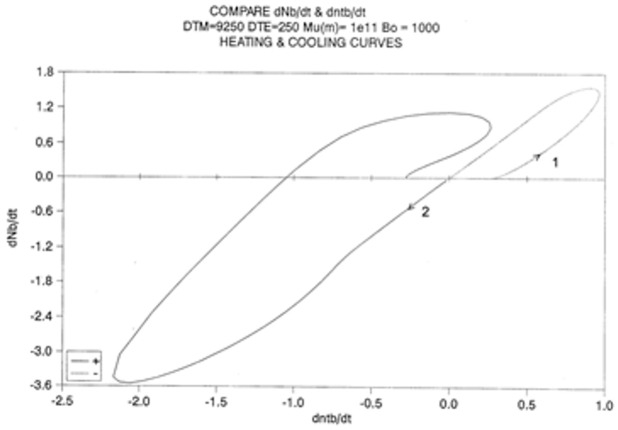
Phase plot of dN_b_/dt vs. dn_tb_/dt for a cooling step (q = −1) followed by a heating step (q = +1), both steps followed by arrows 1 and 2 on the curve, respectively. Δ_m_ = 9250, Δ_e_ = 250, *υ*_m_ = 10^11^, B_o_ = 1000, T = 400 K. Reproduced with permission from [4], SLP Press, 1993.

**Figure 25 polymers-17-00239-f025:**
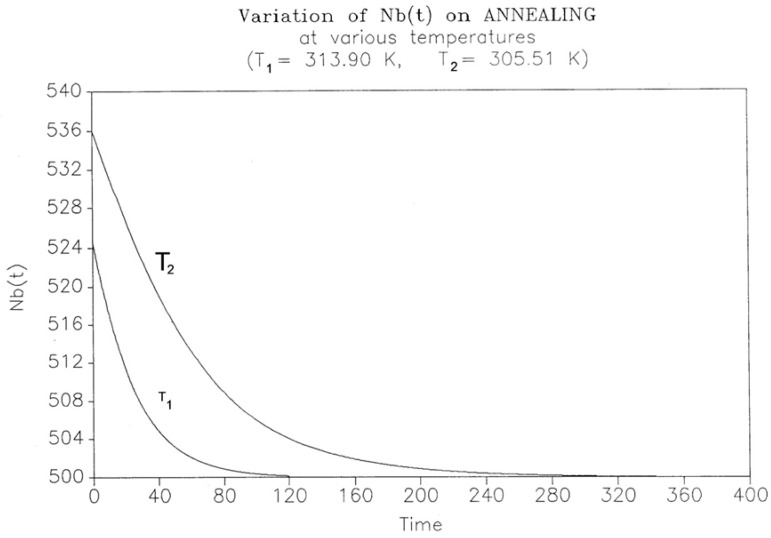
Variation in N_b_(t) on annealing at two temperatures (T_1_ = 313.90 K, T_2_ = 305.51 K). Reproduced with permission from [4], SLP Press, 1993.

**Figure 30 polymers-17-00239-f030:**
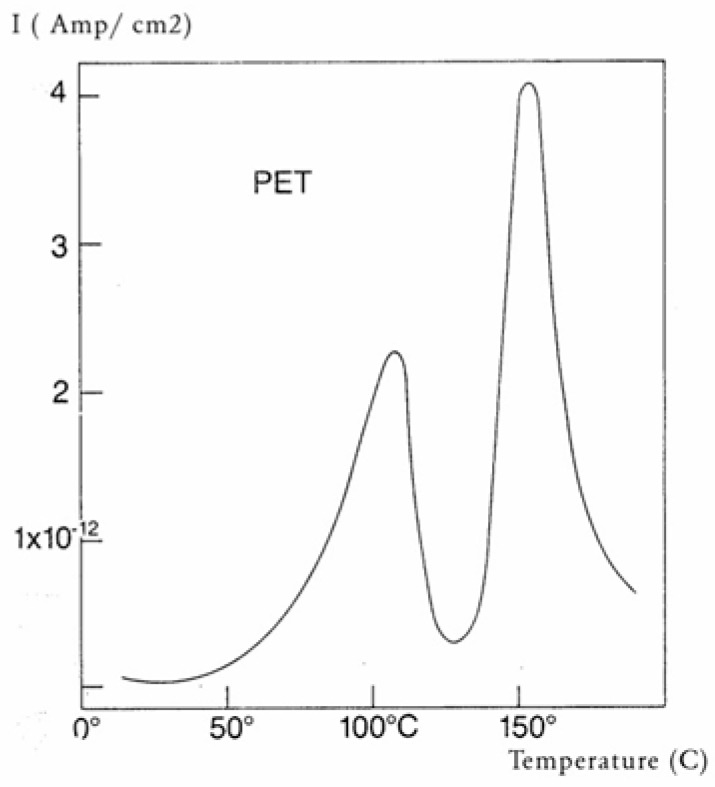
The TSD depolarization curve obtained for PET showing the T_g_ and T_gρ_ peaks. The T_gρ_ manifestation can be correlated to the free volume content in the material. Reproduced with permission from [4], SLP Press, 1993.

**Figure 31 polymers-17-00239-f031:**
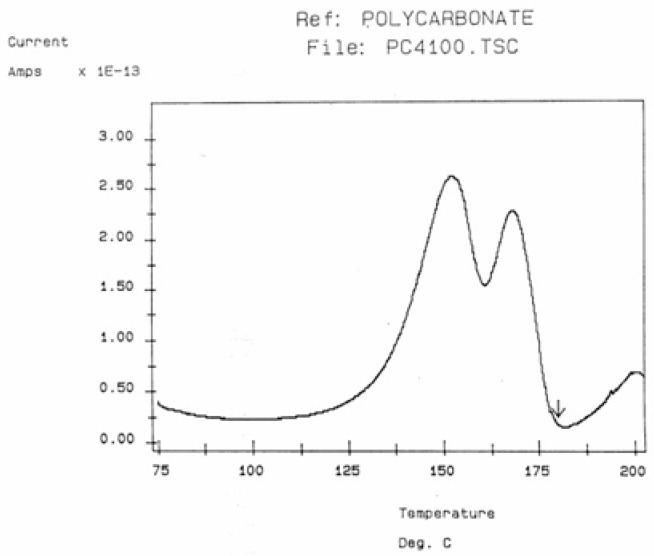
The TSD depolarization curve obtained for PC showing the T_g_ and T_gρ_ peaks. The arrow indicates the temperature of polarization. Reproduced with permission from [4], SLP Press, 1993.

**Figure 32 polymers-17-00239-f032:**
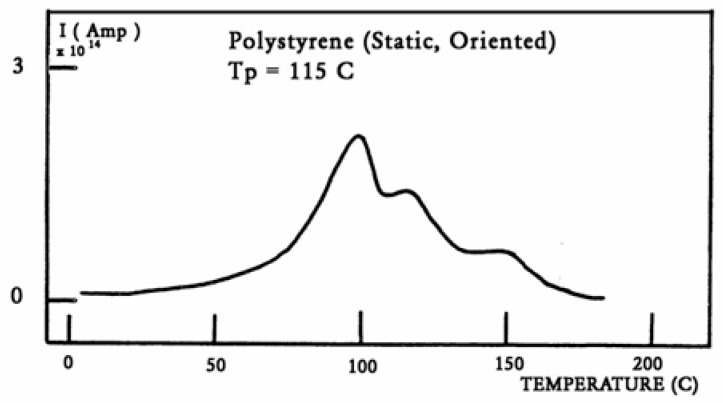
TSD depolarization curve obtained for an oriented compression-molded PS sample showing T_g_ and two other peaks above T_g_. Reproduced with permission from [4], SLP Press, 1993.

**Figure 33 polymers-17-00239-f033:**
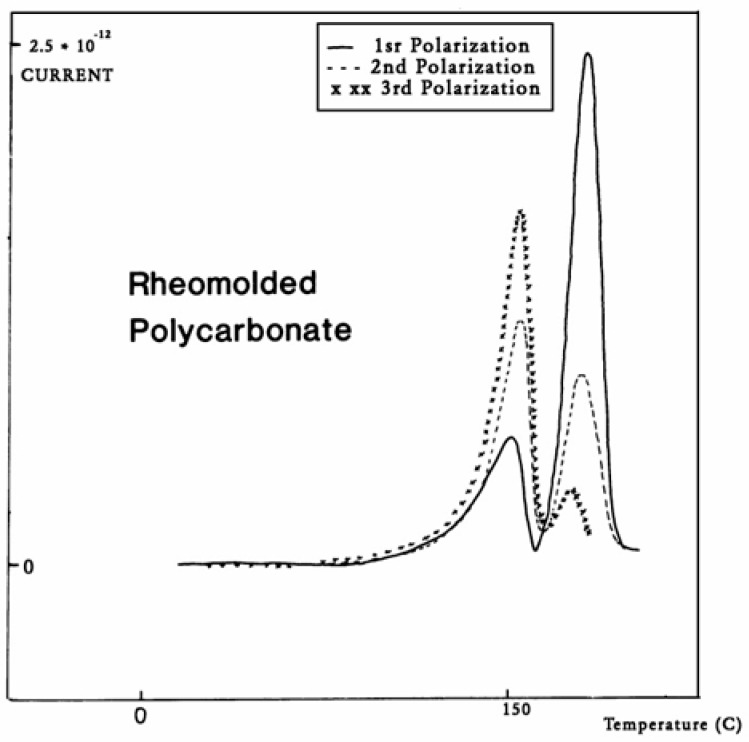
TSD depolarization curve obtained for a mechanically pressurized and vibrated PC sample during its compression molding showing the compensation between the T_g_ and T_gρ_ peak intensity when the sample is annealed due to the repeated polarizations performed above T_g_ on the same sample. Reproduced with permission from [4], SLP Press, 1993.

**Figure 34 polymers-17-00239-f034:**
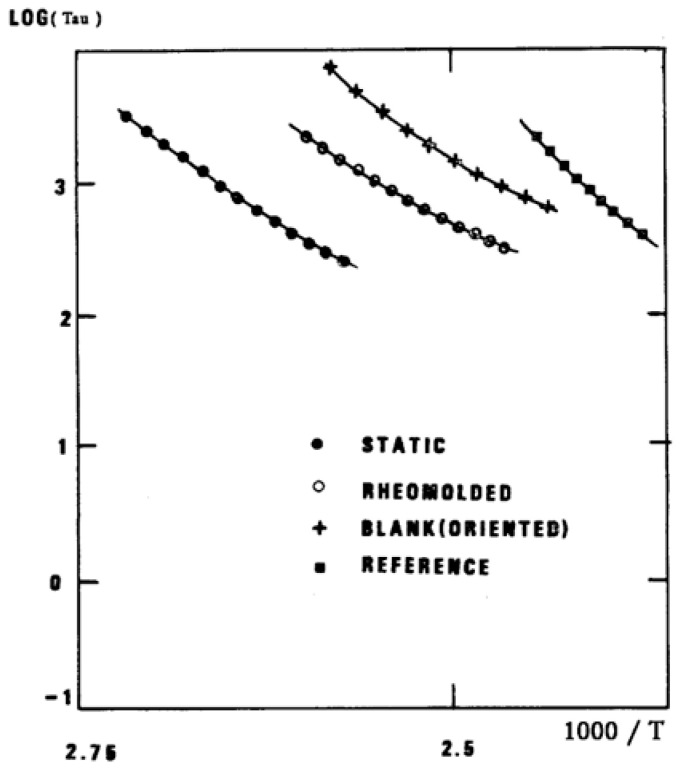
Variation with the thermo-mechanical history of the molded PS of the T_gρ_ peak perceived as a WLF curve in the Arrhenius plane after conversion using Bucci’s equation [14,15]. Reproduced with permission from [4], SLP Press, 1993.

**Figure 35 polymers-17-00239-f035:**
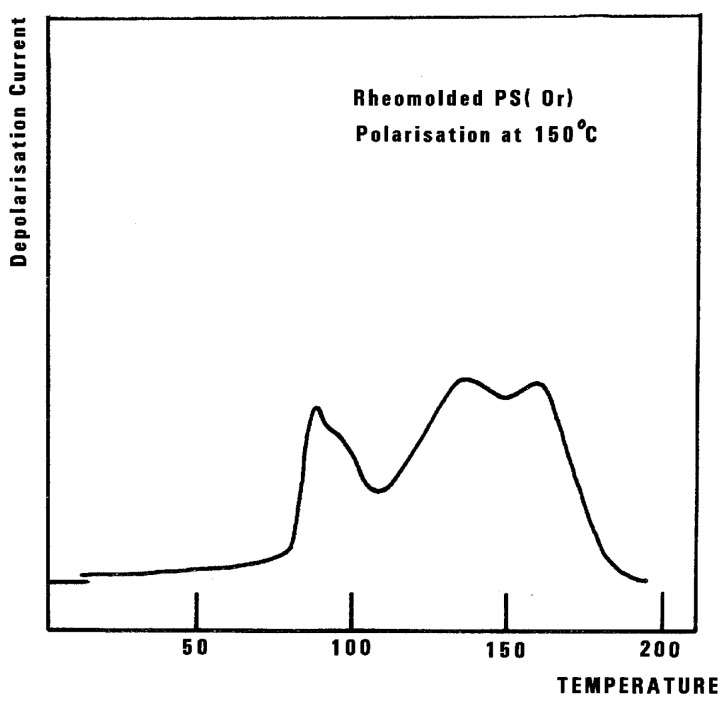
TSD depolarization curve obtained for a vibrated oriented compression-molded PS sample (rheomolded) after polarization at T_p_ = 160 °C showing T_g_ and two other peaks above T_g_. Reproduced with permission from [4], SLP Press, 1993.

**Figure 36 polymers-17-00239-f036:**
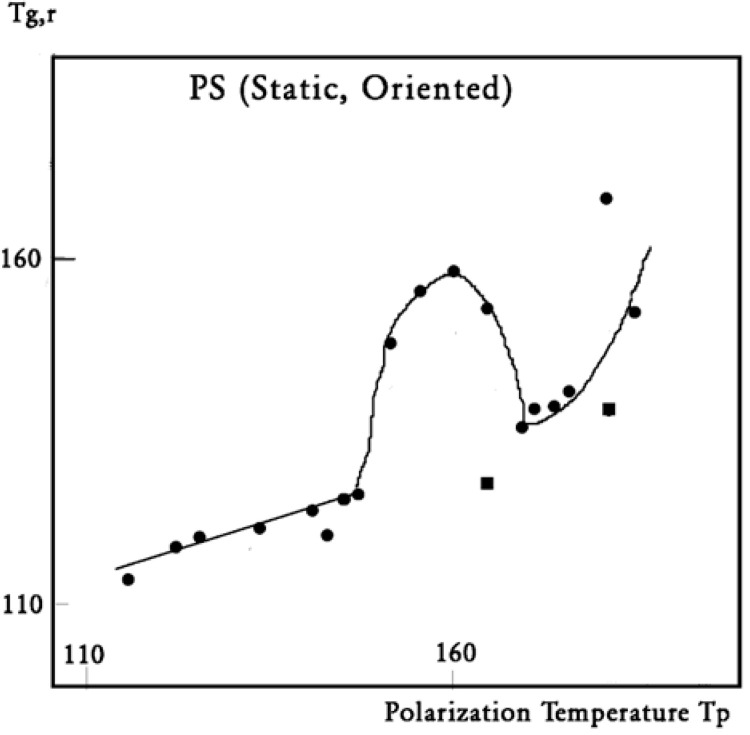
Variation in the position of the T_gρ_ peak with T_p_ for a static (no vibration) oriented compression-molded PS sample, exhibiting the T_LL_ transition at 160 °C. Reproduced with permission from [4], SLP Press, 1993.

**Figure 37 polymers-17-00239-f037:**
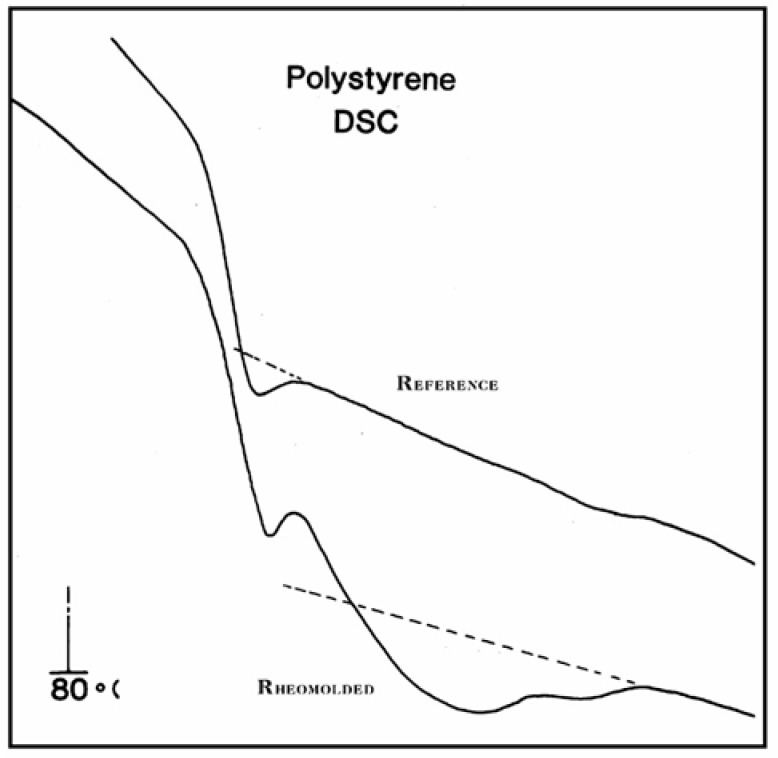
The DSC trace comparison of two PS samples: the reference at the top is a compression-molded general-purpose PS. The bottom trace gives the response for a rheomolded sample, compression-molded, and vibrated at the same time while cooled. The cooling conditions were the same for both samples. Reproduced with permission from [4], SLP Press, 1993.

**Figure 38 polymers-17-00239-f038:**
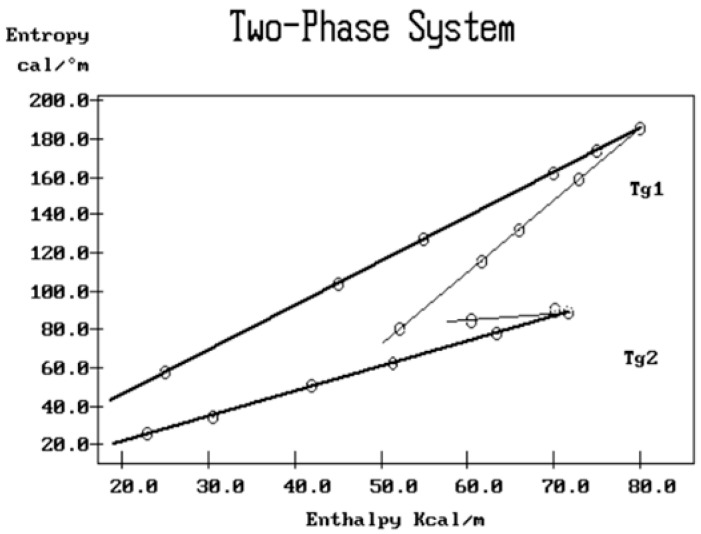
Schematic description of a compensation search in the EE plane to characterize the amorphous phases across their respective T_g_ for a two-phase system, typically a block polymer. Reproduced with permission from [4], SLP Press, 1993.

**Figure 39 polymers-17-00239-f039:**
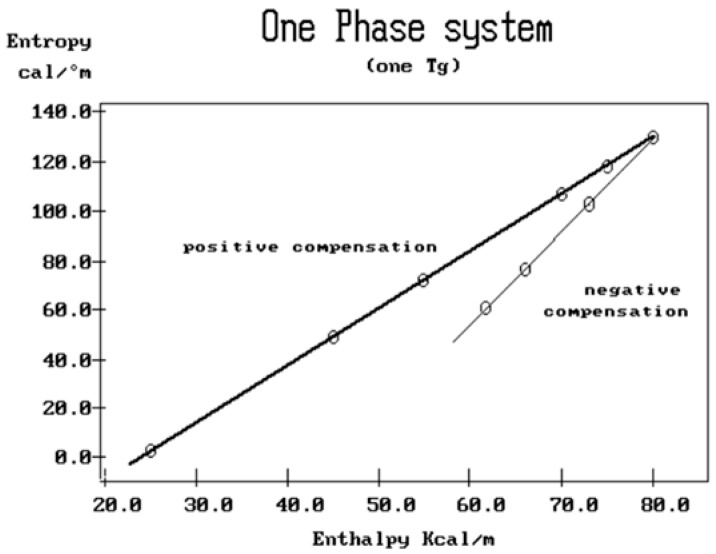
Schematic description of a compensation search in the EE plane to characterize the amorphous phase across T_g_ for a single-phase system, typically an amorphous homopolymer. Reproduced with permission from [4], SLP Press, 1993.

**Figure 42 polymers-17-00239-f042:**
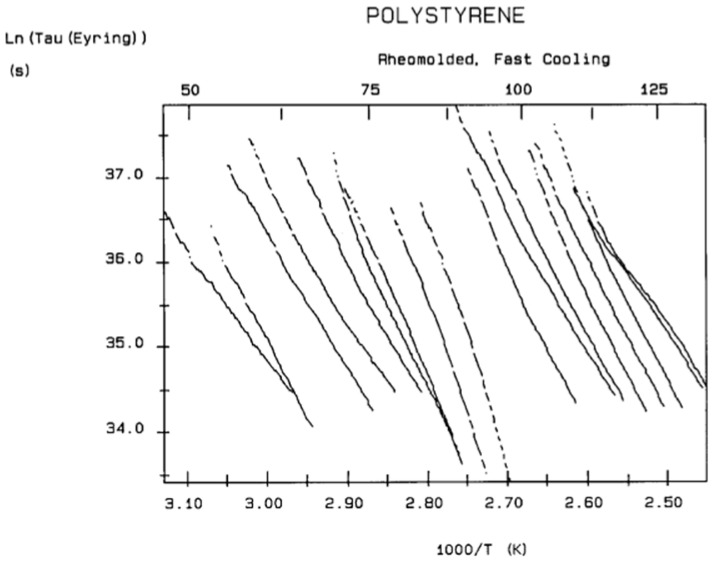
Relaxation map in the Eyring plane for a “rheomolded” compression-molded PS sample designated PS_RL. The sample is pressurized and vibrated during fast cooling in the mold. Reproduced with permission from [4], SLP Press, 1993.

**Figure 56 polymers-17-00239-f056:**
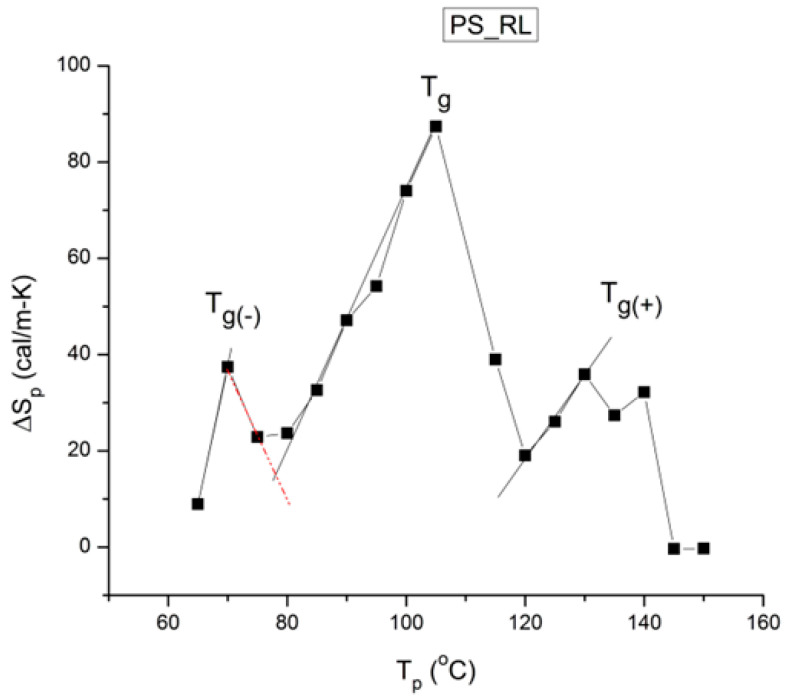
ΔS_p_ vs. T_p_ for PS_RL.

**Figure 57 polymers-17-00239-f057:**
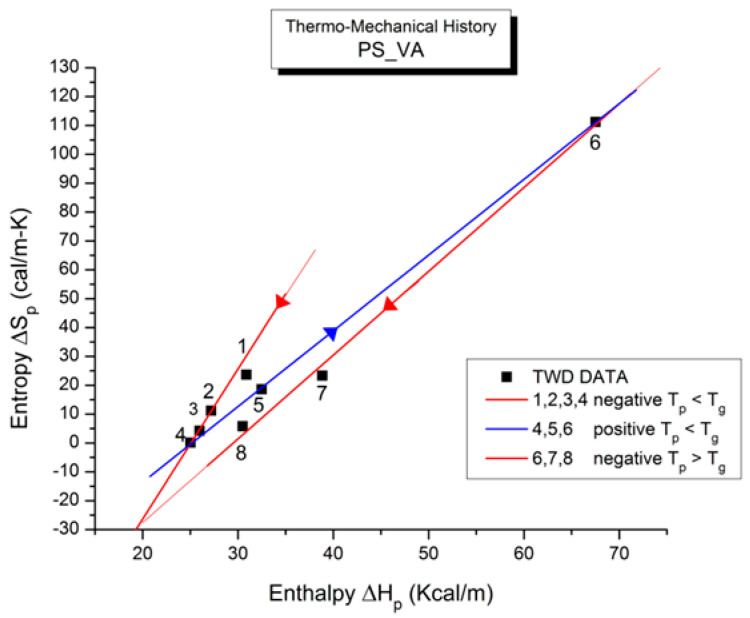
Revised Figure 41 considering points (4,5,6) forming a positive compensation. Additionally, note that the intercept of lines (1,2,3,4)^−^ and (4,5,6)^+^ is point 4, which is located on the ΔS = 0 horizontal line, perhaps indicating that the end of a negative compensation occurs for a T_p_ value for which ΔS = 0. Point 6 starts a new compensation for a T_p_ value that corresponds to ΔG_g_ = ΔG_c_. See text.

**Figure 58 polymers-17-00239-f058:**
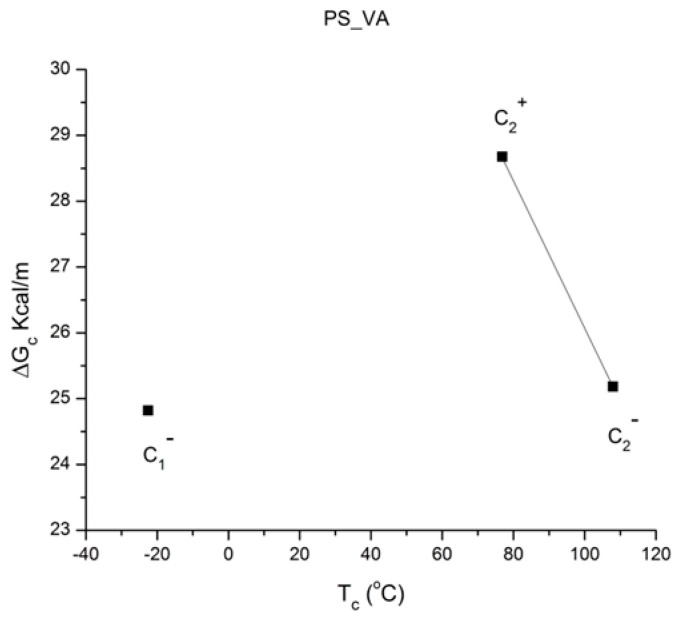
New understanding of the PS_VA thermo-kinetics results of Table 1. Compare to the network of compensations of PS_RL in Figure 48. The negative compensations are positioned here below the positive compensation, which is the opposite of what is seen in Figure 48.

**Figure 59 polymers-17-00239-f059:**
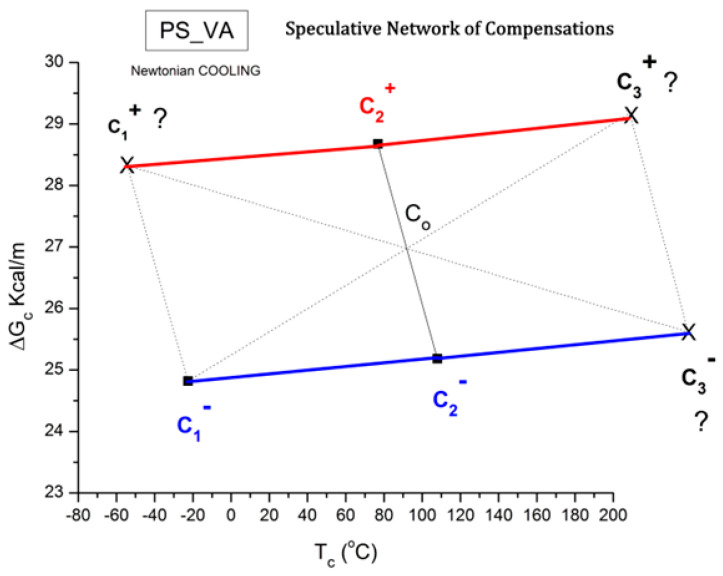
Speculative network of compensation using the geometrical criteria from the analysis of PS_RL to find the “missing” compensation points due to a lack of experimental points at lower and higher T_p_. The known values are in colored text, the extrapolated ones are in black text and with an interrogation point beside them. Compare to Figure 52 for PS_RL.

**Figure 60 polymers-17-00239-f060:**
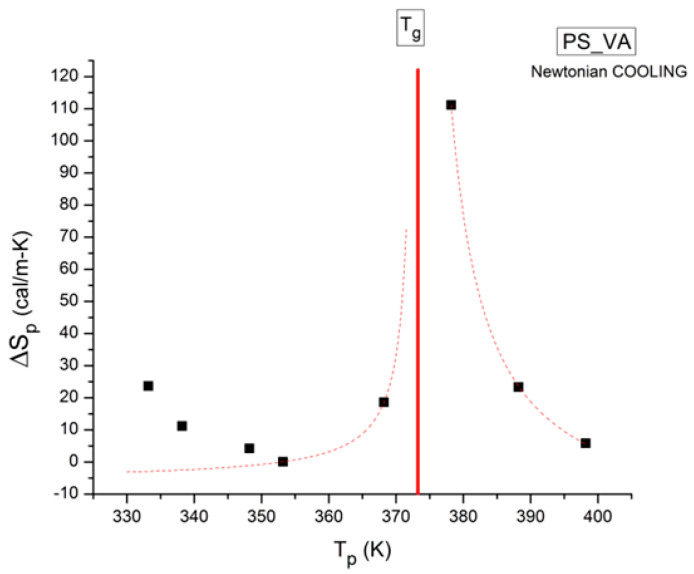
ΔS_p_ vs. T_p_ for PS_VA. Compare to Figure 56 for PS_RL. There is only Tg visible, and the peak of ΔS_p_ at T_g_ is the expected aspect for stable samples. The 4 points at the lower T_p_ end correspond to the negative compensation branch of the peak expected to be found for T_g_(−): a hyperbolic fit could be used to fit those points and determine the value of the T_p_ asymptote equal to T_g_(−).

**Figure 61 polymers-17-00239-f061:**
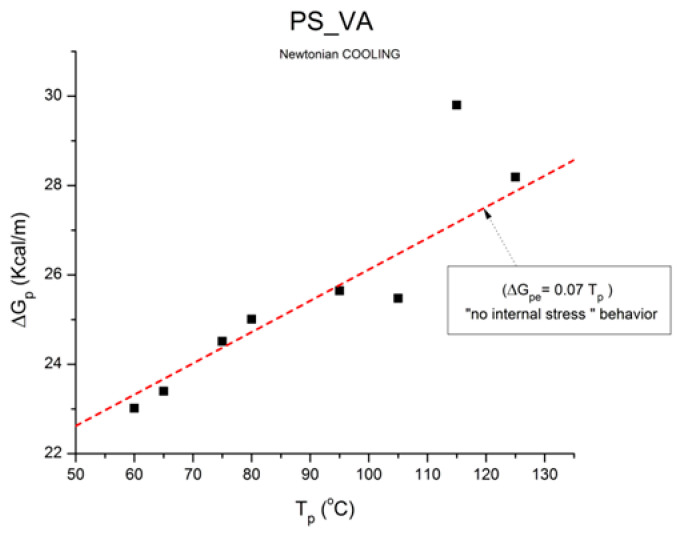
ΔG_p_ vs. T_p_ (C) for sample PS_VA. Compare to Figure 54 for PS_RL. For this slowly cooled sample, the mechanical history resulting in internal stress appears to have vanished since the ΔG_p_ vs. T_p_ returns to a straight line with a slope of 0.07 cal/m-K, as expected for stable samples. The explication of the deviation of 2 of the higher T_p_ points remains uncertain.

**Figure 62 polymers-17-00239-f062:**
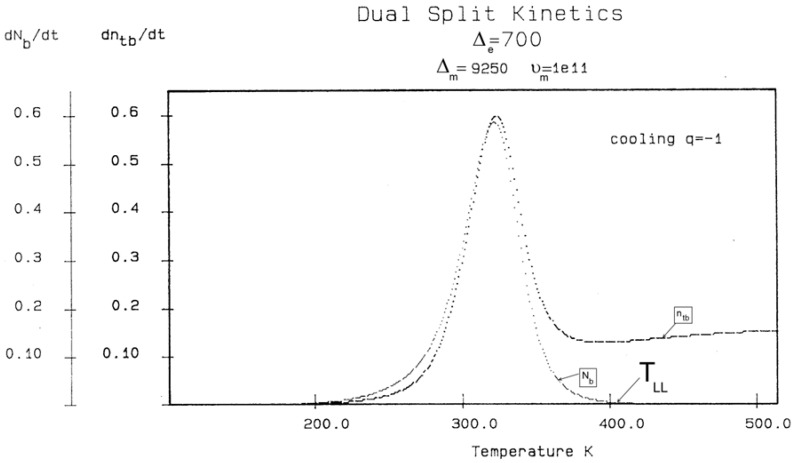
Dual-split kinetic simulation (solution of Equations (6)–(8) of Section 1.2) for Δ_m_ = 9250, υ_m_ = 10^11^, Δ_e_ = 700. Cooling simulation from T_o_ = 515 K with the cooling rate q = −1. The two curves designate the kinetic rates for the population of n_tb_ and N_b_. T_LL_ is the “dissipative“ temperature defined by the onset of an increase in dN_b_/dt. Upon cooling, T_LL_ is the temperature at which the classical kinetics convert to EKNETICS. Upon heating (Figure 22 and Figure 23, and also Figure 37), T_LL_ is the temperature ending the EKNETICS now returning to classical kinetics. Reproduced with permission from [4], SLP Press, 1993.

**Figure 63 polymers-17-00239-f063:**
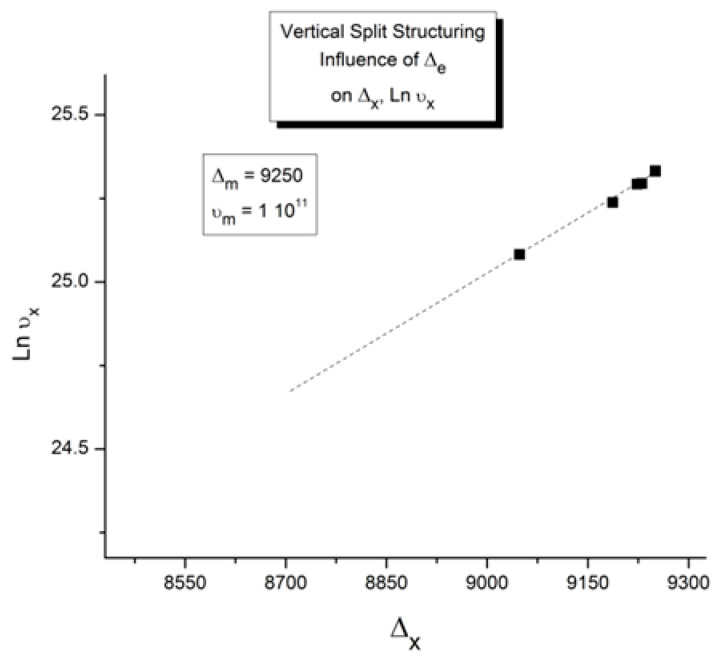
Ln υ_x_ vs. Δ_x_ for Δ_m_ = 9250, υ_m_ = 10^11^ and Δ_e_ variable. Reproduced with permission from [4], SLP Press, 1993.

**Figure 64 polymers-17-00239-f064:**
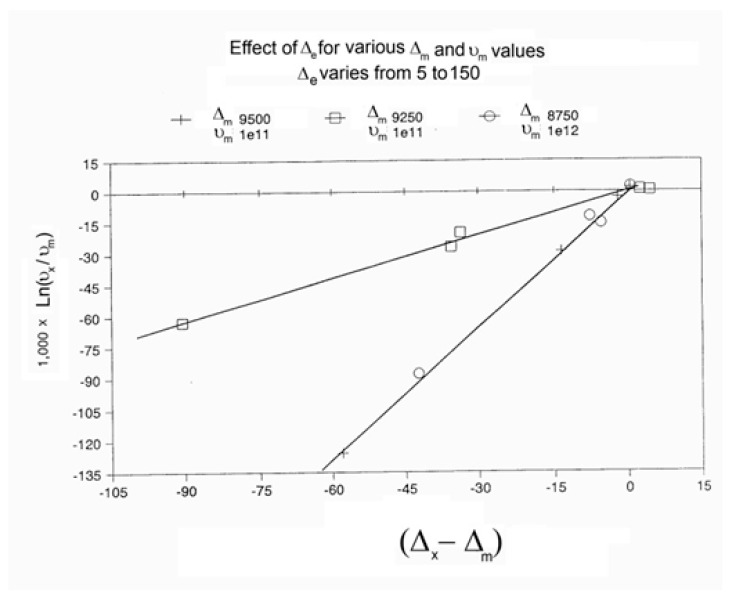
Effect of Δ_e_ for various Δ_m_ and υ_m_ values. Note that Δ_e_ decreases from left to right (from 150 to 5). Reproduced with permission from [4], SLP Press, 1993.

**Table 5 polymers-17-00239-t005:** Coordinates ΔG_c_ and T_c_ of the compensation points for the 3 compensations in Figure 57 for sample PS_VA.

	△G_c_ (kcal/m)	T_c_ (°C)
Comp(1)^−^	24.83	−22.37
Comp(2)^+^	25.23	108.08
Comp(3)^−^	28.67	75.47

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
