# Peer review of "Interactive Coupling Relaxation of Dipoles and Wagner Charges in the Amorphous State of Polymers Induced by Thermal and Electrical Stimulations: A Dual-Phase Open Dissipative System Perspective"

_polymers, 2025, doi:10.3390/polym17020239_

Round 1

Reviewer 1 Report

Comments and Suggestions for Authors

Dear author.

Thank you for interesting article. In total, the article has 89 pages, which I think is a lot. You are invited to view and inspire by the article with DOI: 10.1109/ACCESS.2024.3443462 , and you can see that both articles (your and this) explore how incorporating nanoparticles affects the dielectric properties of polymeric materials, although they use different polymers and experimental setups.

There is some suggestion for you.

  1. Could you elaborate on how the Dual-Phase model can be extended or adapted to different polymeric systems beyond those discussed, such as high-performance composites or bio-based polymers?
  2. What are the challenges or limitations in experimentally validating the Dual-Split Kinetics (EKNETICS) model with real-world polymer samples, and how do these influence the accuracy of the theoretical predictions?
  3. How does the Grain-Field Statistics approach specifically improve upon or diverge from existing theories such as the spaghetti model or traditional Boltzmann-based kinetic models in practical applications?
  4. Could you provide insights into the conditions under which TWD and TSD are most effective compared to conventional DSC or DMA techniques, particularly in materials with complex thermal histories?
  5.  How do you envision the application of TWD findings and the Dual-Phase model influencing future technological developments in material design or polymer processing techniques?

Author Response

Dear Reviewer.

First I would like to thank you very much to have spent the time reading my long review article and for your grading of 5 or 4 stars in each category of the evaluation report. I could not be more honored by your evaluation!

I searched for the DOI you suggested: DOI: 10.1109/ACCESS.2024.3443462 and found the following paper (in ResearchGate):

Fabrication and Broadband Dielectric Study of Properties of Nanocomposites Materials Based on Polyurethane.

The aim of that study addressed the influence of nanoparticles (NPs) on the dielectric properties of polyurethane. This interesting practical research appears to be far-fetched from the objectives of my theoretical article, but resonates well with Part II of my book (cited in Ref. 5 of my review), which deals with practical cases of TSD/TWD analysis, such as the incidence of the concentration of additives on the state of the amorphous phase of polymers, for instance in your case it would be Polyurethane and nanoparticles.

  1. Could you elaborate on how the Dual-Phase model can be extended or adapted to different polymeric systems beyond those discussed, such as high-performance composites or bio-based polymers?

 The Dual-Phase model (EKTOR) simulated in the Review article is the simplest of a family of simulation equations all based on the principle of introducing a dissipative factor in the energy equation of the EKNETICS. For all the simulations, only 3 fundamental parameters must be known, Dm, um, and Deo. In Ref. 7 of the article, I elaborate further what physical parameters influence which fundamental parameter to describe its impact on the state of the amorphous phase. I also say in the paper that the TSD/TWD thermal analysis method may be able to determine these fundamental parameters. Knowing that concentration of an additive does have an impact on the thermokinetic values of the amorphous state is an iindication of the relevance of your question.

  1. What are the challenges or limitations in experimentally validating the Dual-Split Kinetics (EKNETICS) model with real-world polymer samples, and how do these influence the accuracy of the theoretical predictions?

Well, Polystyrene is, indeed, “a real-world” polymer, and its TSD/TWD spectroscopic behavior could be simulated by the Dual-Phase model in the Review article. The simulation of the viscosity behavior can also be successfully addressed by another set of Grain-Field equations (EKTOR-1) which will be published very soon. Much work still needs to be done to address the case of other polymers, as well as the accuracy of the predictions which will require to have solved the issue of finding the fundamental parameters of the model (as covered in my answer to your question 1).

  1. How does the Grain-Field Statistics approach specifically improve upon or diverge from existing theories such as the spaghetti model or traditional Boltzmann-based kinetic models in practical applications?

What I can say at this stage of the development is that the existing theories do not explain a series of experimental facts that scientists systematically find and discard for lack of understanding, facts that the Grain-Field models expect to find and describe well (their rate effects in particular). This new Dual-Phase model explanation of the discarded experimental facts also explains the known viscoelastic behavior of polymers that are, indeed, apparently successfully described by the current models. The opening of the knowledge horizon to include an explanation of all the real facts is the objective of science and opens new practical applications (for example “Sustained Orientation” to decrease the viscosity of Rheo-fluidified pellets to greatly improve their processability without impact on the properties of the parts made from these pellets).  

  1. Could you provide insights into the conditions under which TWD and TSD are most effective compared to conventional DSC or DMA techniques, particularly in materials with complex thermal histories?

Part II, Ch. 1 of Ref.5: “A comparison of TSD/TWD, DSC, DEA, and DMA for the Characterization of the amorphous phase of polymers” might be the most helpful place for me to answer your question (I think you can find an earlier version of fit on the ResearGate platform).

  1. How do you envision the application of TWD findings and the Dual-Phase model influencing future technological developments in material design or polymer processing techniques.

Your question is actually stating the essence of my life’s professional objectives which resulted in the parallel development of the Rheomolding technology, the Rheo-Fluidification process of “disentanglement” to temporarily reduce viscosity of melt and sustain the viscosity reduction in pellets (Sustained-Orientation), the development of the TSD/TWD instruments (at Solomat Instrumentation (CT, USA), and on the theoretical side (starting with my PhD thesis at MIT in 1975), the development of a new theory of the amorphous state, which has become, after 50 years of progressive improvements, the Grain-Field Statistical model of Dissipative Open Systems, which I have only recently started to use to understand and disseminate the other practical ways to describe polymers. So, to respond to your question in one word: ESSENTIAL!  

Reviewer 2 Report

Comments and Suggestions for Authors

The authors review the comprehension of “interactive coupling” between molecular motions in amorphous polymers, pursuant to the published results of related research, or ‘non-traditional’ analyses using the language of Dual-Phase model of polymer interactions, using data generated by thermally stimulated depolarisation and thermal window deconvolution. As the authors state, the understanding of the amorphous state of matter is essential to understand the glass transition, molecular motions in the rubbery and molten states, and even the fundamental mechanisms leading to crystallization from the amorphous state. As a highlight, this review also introduces the “language” of a new physics of polymer interactions, the Dual-Phase theory, and applies it to simulate the relaxation of a closed system of dissipative Dual-conformers that are assumed to be the statistical basic units that interactively couple to explain the viscoelastic properties of polymers. The simulations explain the dynamics of the dissipative system of dipoles on cooling, heating and relaxation and show, for instance, how cooling at constant rate brings the system out of equilibrium , in a way different from a non-dissipative system. The apparent successful theoretical break through to explain compensations and super-compensations may be regarded as a sort of validation of the power of the TSD/TWD thermal analysis technique to measure-up the amorphous state of electrets; and , on the theoretical side, it may also provide the proof of the benefits of using the Dual-Phase statistics to simulate the properties of polymers, including to provide a coherent interpretation of the TLL and Tg manifestations. When dealing with a collection of chains put together, the author defines statistical systems as ‘dual-conformers’ rather than macromolecules. The interactive coupling between the dual-conformers is defined by Grain-Field Statistics that explores the correlation between the local conformational property of the dual-conformers and their collective behavior as a dissipative network. The methodology proposed by the authors is very interesting and inspiring! This review is of great value to our understanding and re-conceptualisation of the intermolecular interactions, thermally induced polarization and dielectric behavior of amorphous polymers. The work is well organized and written. In conclusion, the manuscript will be a good guidance for readers to understand the interactive relaxation of dipole and Wagner charges in polymer amorphous forms induced by thermal and electrical stimulations. So it could be accepted for publication in this journal.

Author Response

Dear Reviewer.

Thank you so much for your almost “perfect score” evaluation of my Review article (only one star missing). I am quite honored by your evaluation comments and your summary of the article.

I would also like to build from the fact that you are addressing your response to “the authors”, although I am the only author of this Review. I particularly like the plurals that you have added to “author” because my research achievements have only been possible thanks to the participation of hundreds of individuals: students, University colleagues, post-docs, technicians and engineers. See the Acknowledgment section of this paper. 

Reviewer 3 Report

Comments and Suggestions for Authors

This paper summarizes the author’s understanding of the physics of interactions in polymers under a voltage field excitation. I do not think it appropriate to review the paper in detail, as it is based on many published texts and a book, which were all undoubtedly reviewed positively before. In fact, as the author admits, it is an abstract of his book, published in 2022, with 600 pages. And it is quite a long abstract, with 90 pages and 70 figures. This is rather unusual for a journal paper and seems more suitable for a chapter in a book or a brochure. I personally doubt that many readers will follow all the route of reasoning and calculations in the text. If the paper were much shorter and concentrated on the physical side of the Dual-Phase model, it would appeal to a wider audience.

Author Response

Dear Reviewer,

Thank you so much for your positive evaluation of my Review article. I am quite honored by your evaluation of my paper.

As far as the length of the review is concerned, it is always a concern of mine: when exposing new concepts, is it better to write a great number of short papers or an entire book? I hesitate and change my mind often to answer this question. As I said, and you mentioned it too, this review was an attempt to abstract the content of a whole book. Actually, this “abstract” focused in summarizing one chapter of the book, Chapter 3 of Part I (my Ref. 7). Even then, I did not cover in the Review “grid-shifting”, nor horizontal EKNETICS (cloning), two essential aspects of this research on interactive coupling in dissipative open systems.

The problem in learning something new or unfamiliar is the time it takes to grasp if it is consequential or not, and often we do not have the time to read the details, as you said. I do not know how to solve this dilemma.

Reviewer 4 Report

Comments and Suggestions for Authors

SUMMARY

The manuscript reviews the understanding of interactive coupling in amorphous polymers by analyzing data from Thermally Stimulated Depolarization (TSD) and Thermal-Windowing Deconvolution (TWD) techniques, both of which involve thermally inducing polarization in dielectric samples and then tracking the depolarization process as the sample is heated. The manuscript compares traditional interpretations of these techniques, rooted in classical polymer physics, with a new, unconventional approach based on the Dual-Phase model. This model offers a fresh perspective by simulating the behavior of polymer units, referred to as “dual-conformers,” which interact within a dissipative network, providing a deeper understanding of viscoelastic properties and molecular dynamics. The review highlights the significance of observing the TLL transition and the Tg,ρpeak in TSD and TWD experiments, phenomena that are not well explained by traditional polymer physics models. The Dual-Phase theory offers a new framework for interpreting these transitions, challenging the conventional understanding of polymer dynamics. The review concludes by advocating for a shift toward recognizing the merits of TSD/TWD techniques and incorporating the Dual-Phase model into polymer physics.

POSITIVE ASPECTS

1. The author provides a detailed historical and technical overview of TSD and TWD techniques.
2. The author discusses the technical procedures and results of the TWD experiment, detailing the processes, applications, and analytical perspectives. Discusses advanced compensation search concepts, the Dual-Split Kinetics (EKNETICS) model, and an introduction to Grain-Field Statistics.
3. The author explains the complex dynamics of polymer kinetics under the EKNETICS (or Dual-Split Kinetics) model, including vertical and horizontal structuring, Grain-Field Theory, and the distinction between conventional and EKNETICS-based kinetics. The author explores the system's kinetics and thermodynamics in response to varying cooling and heating rates and the dynamics during isothermal annealing.
4. In the second section, the author emphasizes how TSD, through peak analysis, captures nuanced thermal and molecular dynamics in polymers, from the dielectric relaxations to transitions that may otherwise remain undetected in conventional thermal analysis methods. This section offers an in-depth exploration of the complexities involved in studying amorphous polymers, particularly focusing on how different processing techniques (e.g., cooling rates, pressure application, and vibration) can lead to super-compensations and reveal new material characteristics that traditional methods might overlook. The proposed models and detailed analysis of compensation lines aim to enhance understanding of polymer behavior, providing more precise tools for material characterization in polymer science. The use of compensation lines and super-compensation points provides a detailed framework to analyze the relaxation processes in materials. This section blends experimental results with theoretical models to deepen the understanding of thermal transitions in polymers, particularly the significance of Tg and the behavior of compensation networks in different thermal states. It paves the way for a more nuanced interpretation of thermal-mechanical processes, particularly in non-equilibrium polymer systems. Additionally, this section presents an in-depth analysis of the TWD results for the PS_VA sample compared to the PS_RL sample.
5. The discussion section highlights several observations that challenge classical polymer physics and introduces the Dual-Phase model as an alternative explanation for the complexities of polymer behavior. The author argues that new interpretations of the relaxation dynamics, based on the interactions between different types of conformers and their associated dissipative behaviors, can provide deeper insights into phenomena like Tg, Tg,ρ, and TLL. The analogy between compensation results obtained by TWD experiments and the compensations observed in Dual-Phase systems is examined. The section posits the hypothesis that the compensation lines observed in simulations might simulate the behavior change that occurs at the Tg (glass transition) point for polymers analyzed using TWD. This section provides further analysis of the relationship between temperature, electrical field, and the kinetic model used for simulating polarization and depolarization dynamics in polymers, particularly in the context of a compensation phenomenon observed during thermal and electric field treatment.
6. The conclusion effectively stresses the importance of embracing new interpretations of polymer dynamics, particularly through the Dual-Phase model. By challenging the classical understanding, the review highlights the potential for TSD and TWD techniques to provide a more accurate representation of polymer behavior, specifically in amorphous states. This work emphasizes the necessity for theoretical advancements to reconcile observed experimental phenomena with polymer physics, advocating for a paradigm shift that could have significant implications for material science and polymer processing.

CONCERNS

My comments are merely editorial (of minor type).

Minor concerns
1. The author should write out the meaning of the acronyms upon first usage.
2. The author uses different types of quotation marks in the text.
3. The text contains redundant spaces.
4. The physical quantity symbols are always written in italic (sloping) type, irrespective of the type used in the rest of the text (ISO 80000-1: 2009), ISO 80000-5: 2007).
5. The symbol ºC for the degree Celsius shall be preceded by a space when expressing a Celsius temperature.
6. Remove the degree sign when the temperature is in Kelvin.

CONCLUSION

I find this manuscript helpful. Regretfully, the paper cannot be accepted in its present form. The author of the present review article have to correct the issues.

Author Response

Dear Reviewer,

Thank you for taking the time to read my paper in great details, and also thank you for your “almost perfect” evaluation (I just missed 1 star!).  Your comments indicate how deep your knowledge is of the critical issues addressed in the paper and I appreciated reading your review of my proposed solutions.

As far as the “minor concerns”, I recognize in your remarks several problems which were the reason for my re-sending the submission, after corrections, of the pdf and docx files on October 16th after having sent it first on October 15th. In any case, I will check that problem of files with the editor and you can be re-assured that these minor editing issues will not appear in the final printed version.  Thank you for pointing out all of them.

Round 2

Reviewer 3 Report

Comments and Suggestions for Authors

Dear Author,

I appreciate the response and the efforts you made during compiling the manuscript. However, I do not quite agree that there is a dilemma, it is rather a trilemma. The third option is to perform the next iteration and make an abstract of the abstract of ca. 75% of the text, the rest going to Supplementary Material.